# CAMSAPs and nucleation-promoting factors control microtubule release from γ-TuRC

Dipti Rai [1], Yinlong Song[1], Shasha Hua[2,3], Kelly Stecker[4,5], Jooske L. Monster [1,6], Victor Yin [4,5], Riccardo Stucchi[1,4,5], Yixin Xu [7], Yaqian Zhang[2,3], Fangrui Chen[1], Eugene A. Katrukha [1], Maarten Altelaar [4,5], Albert J. R. Heck [4,5], Michal Wieczorek[7], Kai Jiang [2,3] ✉ & Anna Akhmanova [1] ✉

γ-Tubulin ring complex (γ-TuRC) is the major microtubule-nucleating factor. After nucleation, microtubules can be released from γ-TuRC and stabilized by other proteins, such as CAMSAPs, but the biochemical cross-talk between minus-end regulation pathways is poorly understood. Here we reconstituted this process in vitro using purified components. We found that all CAMSAPs could bind to the minus ends of γ-TuRC-attached microtubules. CAMSAP2 and CAMSAP3, which decorate and stabilize growing minus ends but not the minus-end tracking protein CAMSAP1, induced microtubule release from γ-TuRC. CDK5RAP2, a γ-TuRC-interactor, and CLASP2, a regulator of microtubule growth, strongly stimulated γ-TuRC-dependent microtubule nucleation, but only CDK5RAP2 suppressed CAMSAP binding to γ-TuRC-anchored minus ends and their release. CDK5RAP2 also improved selectivity of γ-tubulin-containing complexes for 13- rather than 14-protofilament microtubules in microtubule-capping assays. Knockout and overexpression experiments in cells showed that CDK5RAP2 inhibits the formation of CAMSAP2-bound microtubules detached from the microtubule-organizing centre. We conclude that CAMSAPs can release newly nucleated microtubules from γ-TuRC, whereas nucleation-promoting factors can differentially regulate this process.

Microtubule organization in animal cells is a major determinant of cell architecture and polarity[1]. This organization critically depends on the activity of microtubule-organizing centres (MTOCs)—structures that can nucleate microtubules and stabilize and anchor their minus ends[2–4]. The major microtubule-nucleating factor in cells is the γ-tubulin ring complex (γ-TuRC)[5,6]. γ-TuRC localization and activity are controlled by diverse factors, such as augmin, pericentrin, CDK5RAP2 and chTOG[6,7]. γ-TuRC can also cap microtubule minus ends[8] and participate in their anchoring, possibly with the aid of additional microtubule-organizing centre components[9]. An alternative well-studied pathway of minus-end stabilization and anchoring depends on the members of CAMSAP or Patronin family[10–13]. These proteins specifically recognize free,

[1]Cell Biology, Department of Biology, Faculty of Science, Utrecht University, Utrecht, the Netherlands. [2]State Key Laboratory of Oral and Maxillofacial Reconstruction and Regeneration, Key Laboratory of Oral Biomedicine Ministry of Education, Hubei Key Laboratory of Stomatology, School and Hospital of Stomatology, Medical Research Institute, Wuhan University, Wuhan, China. [3]Frontier Science Center for Immunology and Metabolism, Wuhan University, Wuhan, China. [4]Biomolecular Mass Spectrometry and Proteomics, Bijvoet Center for Biomolecular Research, Utrecht Institute for Pharmaceutical Sciences and the Netherlands Proteomics Center, Utrecht University, Utrecht, the Netherlands. [5]Netherlands Proteomics Center, Utrecht, the Netherlands. [6]Center for Molecular Medicine, University Medical Center Utrecht, Utrecht, the Netherlands. [7]Department of Biology, Institute of Molecular Biology & Biophysics, ETH Zürich, Zurich, Switzerland. ✉e-mail: jiangkai@whu.edu.cn; a.akhmanova@uu.nl

uncapped microtubule minus ends because their signature domain, CKK, binds to a minus-end-specific site between flared protofilaments[14]. It was also proposed that CAMSAP2 can nucleate microtubules independently from γ-TuRC[15].

In γ-TuRC-capped microtubules, the protofilaments at the minus ends are straight[16,17]; therefore, they should not be able to bind to CAMSAPs. However, recent studies revealed that γ-TuRCs are asymmetric, and their structure does not fully match that of a 13-protofilament microtubule[18–21]. This finding raises the possibility that γ-TuRC-nucleated microtubules may not be fully attached to their template, and some protofilaments might have a flared conformation that would permit CAMSAP binding. Furthermore, since the microtubule-nucleating activity of purified γ-TuRC is quite low[18,22,23], a potential mechanism of stimulating microtubule nucleation would be to alter γ-TuRC conformation to make it more similar to the microtubule structure[6,7,24,25].

In this Article, to explore these possibilities, we have set up in vitro reconstitution assays and confirmed that the activity of purified γ-TuRC was low but could be enhanced by microtubule polymerase chTOG and γ-TuRC-associated protein CDK5RAP2 (refs. 26–28). We also found that γ-TuRC was also activated by CLASP2, which enhances microtubule outgrowth from stabilized seeds[29]. Furthermore, while microtubules almost never detached from γ-TuRC when it was present alone or together with CDK5RAP2 or CLASP2, CAMSAPs could bind to a subset of γ-TuRC-anchored minus ends and trigger their release. This process was counteracted by the γ-TuRC-binding factor CDK5RAP2, which also suppressed formation of CAMSAP2-stabilized non-centrosomal microtubules in cells, but not by CLASP2. By controlling not only the nucleation but also microtubule release, γ-TuRC activators can thus regulate the relative abundance of different microtubule populations, such as centrosomal and non-centrosomal microtubules.

## Results

### CDK5RAP2, CLASP2 and chTOG enhance γ-TuRC activity

To obtain purified γ-TuRC, we have used clustered regularly interspaced short palindromic repeats (CRISPR)–Cas9-mediated gene editing to generate a homozygous knockin HEK293T cell line where the γ-tubulin complex protein (GCP)3-encoding gene was modified by a C-terminal insertion of the green fluorescent protein (GFP) and a twin-strep-tag (GFP–SII, Extended Data Fig. 1a,b). In these cells, the GFP signal was diffusely localized in the cytoplasm and concentrated at the centrosome, as expected (Extended Data Fig. 1c). Western blotting showed that the whole GCP3 pool was shifted up by ~30 kDa (Extended Data Fig. 1d). SII-based purification yielded protein complexes that, based on western blotting and quantitative mass spectrometry (MS), contained all expected γ-TuRC components with relative abundances quite close to expected, as shown by intensity-based absolute quantification

(iBAQ) ratios (Extended Data Fig. 1e–g and Supplementary Tables 1 and 2). In these γ-TuRC preparations, we detected two proteins known to co-purify with γ-TuRC, NEDD1 and NME7 (refs. 18–20,27,30,31) (Extended Data Fig. 1g), but no other known γ-TuRC-binding partners or microtubule nucleation-promoting factors (NPFs).

We characterized the purified γ-TuRCs by 5–40% sucrose density gradient centrifugation and mass photometry (MP)[32,33] (Extended Data Fig. 1h–j). The most prominent band of γ-tubulin was observed in sucrose gradient fractions 10 of 14, consistent with previous studies[20,27]. Both methods indicated the presence of complete and incomplete γ-TuRCs (collectively termed γ-tubulin-containing complexes, or γ-TuCs, in this study). Complexes corresponding to the mass of fully assembled γ-TuRC (molecular weight ~2.5 MDa) represented ~11% of the particles analysed by MP. However, since MP detects all low molecular weight contaminants, it does not provide an accurate estimate of the ratio between complete and incomplete γ-TuRCs. To circumvent this problem, we compared fluorescence intensity of purified complexes with that of single GFP molecules and GFP–EB3 dimers[34]. In these measurements, GFP–EB3 was ~1.7× brighter than GFP, whereas GCP3–GFP-containing fluorescent puncta displayed two peaks, with intensities corresponding to one to two GFPs (37%) and four to five GFPs (36%) (Fig. 1a and Extended Data Fig. 1k). The first peak most likely included γ-tubulin small complexes (γ-TuSC), which are expected to contain one GCP3–GFP subunit, whereas the second peak confirmed the presence of complete γ-TuRCs with five GCP3–GFP subunits[18–21,25]. However, GFP counting may underestimate the number of GCP3–GFP subunits within individual complexes due to improper GFP folding, photobleaching or GFP blinking. Three-dimensional (3D) reconstruction of negative-stain transmission electron microscopy (EM) micrographs of purified γ-TuRCs showed a cone-like complex (Fig. 1b and below), matching the recently published γ-TuRC structures[17–21].

We next immobilized purified GFP-tagged γ-TuCs on coverslips using a biotinylated anti-GFP nanobody, observed microtubule nucleation in the presence of Rhodamine-labelled tubulin by total internal reflection fluorescence (TIRF) microscopy (Extended Data Fig. 1l) and counted the percentage of γ-TuCs nucleating microtubules within three consecutive 10 min periods. In the presence of 17.5 μM tubulin, only ~1–3% of γ-TuC could nucleate microtubules within 10 min of observation (Fig. 1c,d), similar to γ-TuRCs obtained using other purification approaches[18,22]. We then investigated the impact of several microtubule- or γ-TuRC-binding proteins on γ-TuRC-mediated microtubule nucleation (Fig. 1d–h). The addition of mCherry–EB3 to the assays did not affect the nucleation efficiency although it increased microtubule growth rate and catastrophe frequency (Figs. 1d,e and 2a,b and Extended Data Fig. 2a). In contrast, three other proteins, CDK5RAP2, chTOG and CLASP2, could potentiate microtubule nucleation, both when added to γ-TuCs immobilized on coverslips ('No premix') or when

**Fig. 1 | Human CDK5RAP2, CLASP2 and chTOG promote microtubule nucleation by purified γ-TuCs. a**, Left: representative images of single molecules of indicated purified proteins absorbed on coverslips. Middle: a histogram of single-molecule fluorescence intensities. The numbers of analysed molecules are as follows: $n = 21,349$ (GFP), $n = 28,776$ (GFP–EB3) and $n = 24,216$ (GCP3–GFP), from three independent experiments. Right: the probability density of GCP3–GFP intensities (yellow) fitted to a weighted sum of $N$-mers of GFP (dashed line) (a representative experiment). Weighted probability densities of individual GFP $N$-mer intensities (×1, ×2, …) are plotted beneath. See also Extended Data Fig. 1k. **b**, A 3D reconstruction of γ-TuRC (12,851 particles) from negative-stain EM data and rigid body fit of repeating γ-tubulin/GCP2 subcomplexes (from PDB ID: 6V6S (ref. 20)) individually docked into the γ-TuRC density map. Fits for two subcomplexes at the γ-TuRC 'seam' were not reliable and are therefore omitted (Extended Data Fig. 5i). **c,e–h**, Left: maximum intensity projections of 10 min videos acquired after 20 min of incubation, showing microtubules (cyan) nucleated from γ-TuC (yellow) in the presence of either tubulin alone (**c**) or together with mCherry–EB3 (**e**) or mCherry–CDK5RAP2 (**f**) or mCherry–CLASP2 (**g**) or chTOG–mCherry (**h**) in the indicated conditions. In **f** and **g**, γ-TuC

was also pre-incubated with indicated proteins ('Premix'); experiments without pre-incubation are labelled as 'No premix'. The arrowheads in insets indicate colocalizing particles. Right: representative kymographs and schemes illustrating microtubule dynamics and re-nucleation events (thin white arrows). Minus and plus indicate the two microtubule ends. The black arrowheads on top of kymographs indicate γ-TuC position and yellow and magenta arrows indicate γ-TuC and other proteins. The magnification is the same in **c** and **e–h**. **d**, Efficiency (mean ± s.e.m.) of microtubule nucleation by γ-TuC in the presence of either tubulin alone ($n = 3$) or together with mCherry–EB3 ($n = 3$), mCherry–CDK5RAP2 ($n = 3$), mCherry–CDK5RAP2 pre-incubated with γ-TuC ($n = 3$), mCherry–CLASP2 ($n = 4$), mCherry–CLASP2 pre-incubated with γ-TuC ($n = 3$), chTOG–mCherry ($n = 4$) or chTOG–mCherry pre-incubated with γ-TuC ($n = 3$), where $n$ is the number of independent experiments. ND, could not be determined. Representative images are shown on the left of **c** and **e–h** and Extended Data Fig. 2c,d. Data points represent single fields of view for the given time point per experiment. Data points in cyan (0–10 min) were acquired from a smaller field of view; data points at 10–20 min and 20–30 min were acquired from a larger field of view shown in **c** and **e–h**.

pre-incubated with γ-TuCs in solution before immobilization ('Premix', Fig. 1d,f–h and Extended Data Fig. 2b–e, and for MS-based characterization of the purified proteins, see Supplementary Tables 3–5).

Full-length mCherry–CDK5RAP2 increased microtubule nucleation ~threefold when added to immobilized γ-TuCs and more than 20-fold (up to ~35% nucleation efficiency) when additionally

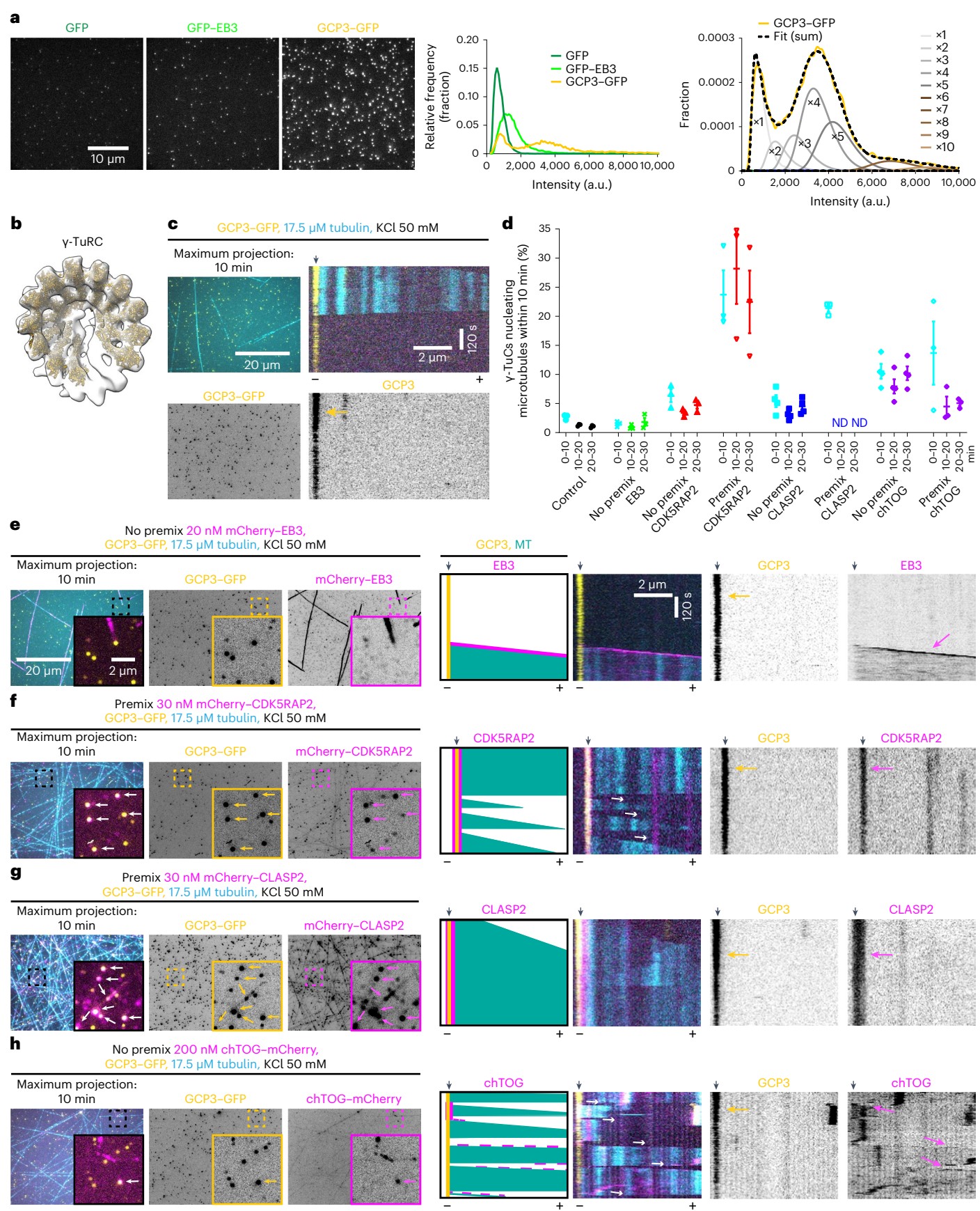

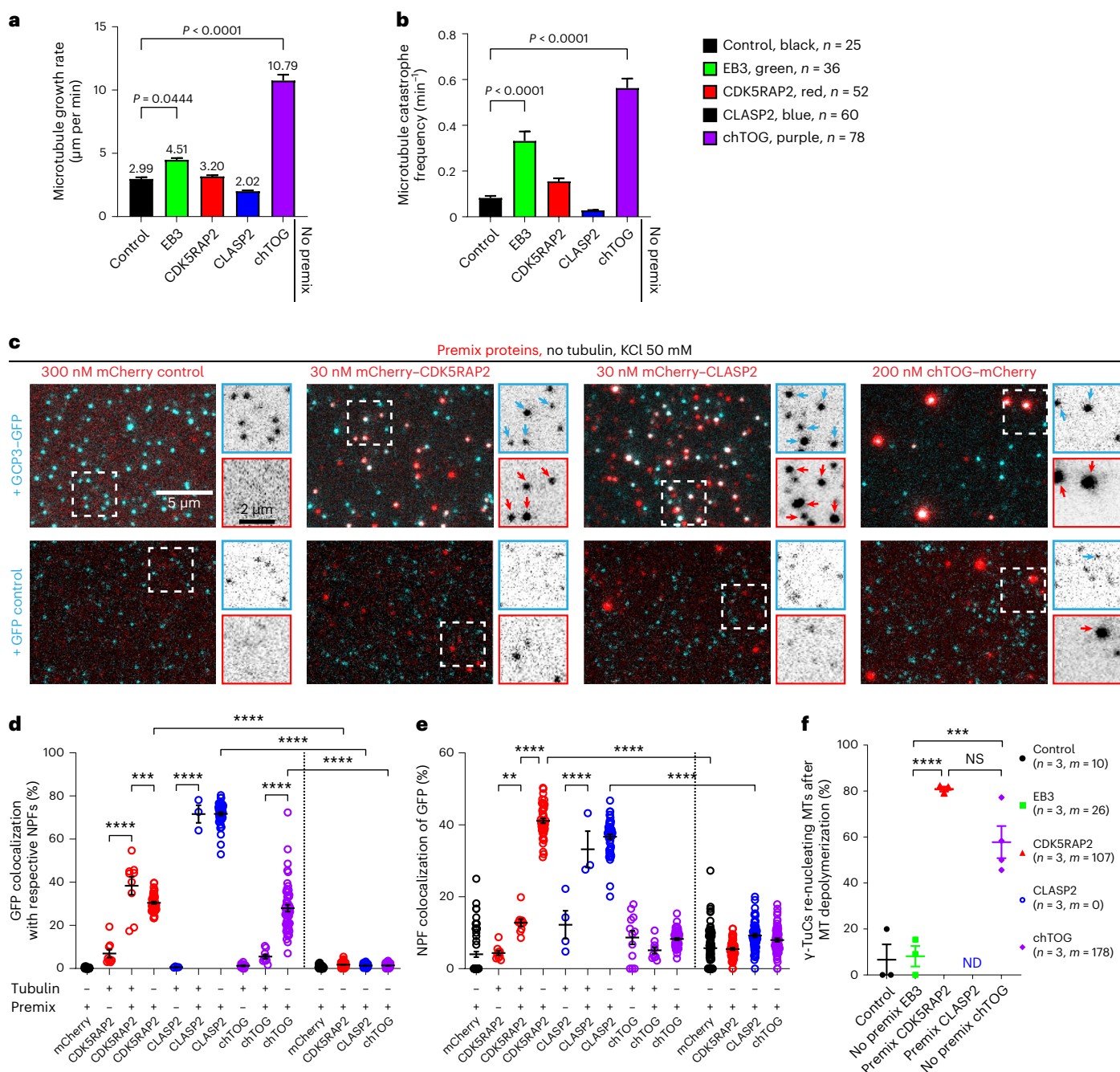

**Fig. 2 | Colocalization between γ-TuCs and NPFs and their effects on microtubule dynamics. a,b**, Plus-end growth rate (**a**) and catastrophe frequency (**b**) of microtubules grown in indicated conditions, shown in Fig. 1c,e,h and Extended Data Fig. 2c,d with representative kymographs on the right. The numbers of the growth events analysed, *n*, from three independent experiments, are indicated. One-way analysis of variance (ANOVA) tests with Dunnett's multiple comparisons corrected for multiple testing. **c**, Representative still images showing colocalization of the indicated premixed proteins. The enlargements show separate channels for γ-TuC/control (top) and NPFs (bottom). The arrows indicate colocalizing particles. **d,e**, Colocalization (mean ± s.e.m) of γ-TuC with indicated NPFs (**d**) and vice versa (**e**) under experimental conditions shown in **c** and in Fig. 1c,e–h and Extended Data Fig. 2c,d. Data points represent the percentage of γ-TuCs colocalizing with the indicated protein or vice versa in n fields of view, in *N* independent experiments. The plots show data for γ-TuC premixed with mCherry without tubulin (*n* = 63, *N* = 3), γ-TuC with mCherry–CDK5RAP2 and tubulin (*n* = 9, *N* = 3), γ-TuC premixed with mCherry–CDK5RAP2 with (*n* = 9, *N* = 3) and without tubulin (*n* = 47, *N* = 3), γ-TuC with mCherry–CLASP2

and tubulin (*n* = 4, *N* = 4), γ-TuC premixed with mCherry–CLASP2 with (*n* = 3, *N* = 3) and without tubulin (*n* = 48, *N* = 3), γ-TuC with chTOG–mCherry and tubulin (*n* = 12, *N* = 4), γ-TuC premixed with chTOG–mCherry with (*n* = 9, *N* = 3) and without tubulin (*n* = 51, *N* = 3), GFP control premixed with mCherry without tubulin (*n* = 80, *N* = 4), GFP control premixed with mCherry–CDK5RAP2 without tubulin (*n* = 60, *N* = 3), GFP control premixed with mCherry–CLASP2 without tubulin (*n* = 60, *N* = 3) and GFP control premixed with chTOG–mCherry without tubulin (*n* = 59, *N* = 3). **P* = 0.0068, ***P* = 0.0003 and *****P* < 0.0001, one-way ANOVA test with Tukey's multiple comparisons corrected for multiple testing. **f**, Microtubule (MT) re-nucleation efficiency (mean ± s.e.m) from experiments shown in Fig. 1c ,e–h. Data points represent percentage of γ-TuCs re-nucleating microtubules in a single experiment. *n*, the number of independent experiments analysed and *m*, the number of γ-TuCs which nucleated microtubules that underwent depolymerization, pooled from all three 10 min videos of all experiments. Not significant (NS) *P* = 0.0704, ****P* = 0.0007 and *****P* < 0.0001, one-way ANOVA test with Tukey's multiple comparisons corrected for multiple testing. ND, could not be determined.

pre-incubated with γ-TuCs (Fig. 1d,f and Extended Data Fig. 2b,c). This was probably because pre-incubation greatly increased the percentage of γ-TuCs colocalizing with CDK5RAP2 (4–6 fold), which in pre-incubated samples could reach 30–40%, both in the presence (~38%) and absence (~30%) of free tubulin (Figs. 1f and 2c–e and Extended Data Fig. 2c). In contrast, CDK5RAP2 did not show any specific colocalization with the purified GFP control even after pre-incubation (~2%, Fig. 2c,d). These data confirm that CDK5RAP2 directly interacts with γ-TuC[27] and suggest that in our assays, it can activate the majority of γ-TuCs to which it binds. The percentage of CDK5RAP2-positive puncta colocalizing with γ-TuCs was rather low (~13% with tubulin and ~41% without tubulin), probably because CDK5RAP2 was present in excess or due to the auto-inhibition of CDK5RAP2 that is controlled by phosphorylation[35,36]. Compared with the samples with tubulin alone or with mCherry–EB3, mCherry–CDK5RAP2 had no notable effect on microtubule growth rate or catastrophe frequency, but strongly increased the frequency of microtubule re-nucleation after depolymerization, indicating that CDK5RAP2 can maintain γ-TuC in an active state (Fig. 2a,b,f and Supplementary Video 1).

We also observed a strong increase in microtubule nucleation with mCherry–CLASP2 (Fig. 1d,g and Extended Data Fig. 2b), a protein known to promote microtubule outgrowth from stabilized seeds[29], but never tested for interaction with γ-TuRC. The activating effect of mCherry–CLASP2 was similar in magnitude to that of CDK5RAP2 and was again stronger after pre-incubation (Fig. 1d,g and Extended Data Fig. 2d). mCherry–CLASP2 also strongly and specifically colocalized with γ-TuC (~72% colocalization with γ-TuC in the presence or absence of free tubulin, ~1.5% colocalization with GFP control; Fig. 2c–e), suggesting that it binds to γ-TuC directly. Since CLASP2 strongly suppresses catastrophes[29], and therefore microtubules become very long (Figs. 1g and 2b), it was not possible to examine microtubule re-nucleation from the same γ-TuC or analyse premixed CLASP2–γ-TuRC samples incubated longer than 10 min due to high microtubule density.

Finally, we also examined the effect of chTOG, because it can enhance microtubule nucleation from free tubulin[37] and γ-TuRC[18], and its *Xenopus* homologue XMAP215 can synergize with γ-TuRC in egg extracts[26] and promote microtubule outgrowth from arrays of laterally associated γ-tubulins[38] and stabilized seeds[39]. We confirmed that chTOG enhanced γ-TuC-dependent microtubule nucleation, although unlike CDK5RAP2 and CLASP2, the effect was similar with and without pre-incubation (Fig. 1d,h and Extended Data Fig. 2b) and not as strong as previously published[18]. This could be caused by the differences in experimental conditions, but was unlikely to be due to the low activity of chTOG, as it strongly increased the growth rate and catastrophe frequency in our assays (Fig. 2a,b), in line with published data[40,41]. Colocalization of chTOG-positive puncta with γ-TuC was lower than that of CDK5RAP2 or CLASP2 (6% with free tubulin, to which

chTOG binds through its TOG domains[42]), but increased to 28% in the absence of free tubulin (Fig. 2d,e). When chTOG was pre-incubated with γ-TuC–GFP or GFP control, it formed clusters that specifically sequestered γ-TuC–GFP, but not GFP alone (Fig. 2c), and therefore we did not premix chTOG and γ-TuC in the experiments described from here onwards. Similar to CDK5RAP2, chTOG potentiated repeated microtubule nucleation from the same γ-TuC (Fig. 2f and Supplementary Video 1). Increasing the concentration of the tested proteins in conditions without premixing did not boost nucleation efficiency (Extended Data Fig. 2a,c–e), indicating that nucleation was limited by the activity or surface interactions of γ-TuCs rather than the availability of NPFs. We conclude that purified γ-TuC can nucleate microtubules in a manner dependent on various interactors.

## CAMSAP3 triggers microtubule release from γ-TuCs

In the assays described above, microtubule minus ends typically stayed attached to γ-TuCs that nucleated them, though the frequency of release events was slightly increased by chTOG (from ~1% to ~2.5–3%) (Extended Data Fig. 3a–c and Supplementary Video 2). To examine whether γ-TuCs prevent CAMSAPs from binding to minus ends, we initially used CAMSAP3, which has the highest minus-end affinity in vitro among mammalian CAMSAPs[43]. To recapitulate CAMSAP3 specificity for growing microtubule minus ends, the ionic strength of the buffer needs to be sufficiently high (MRB80 buffer supplemented with 80 mM KCl, instead of 50 mM KCl used above). Since high ionic strength suppresses microtubule assembly[44], we increased tubulin concentration to 25 μM and obtained ~1.5–3% nucleation efficiency with and without SNAP–AF647–CAMSAP3 (Fig. 3a,b and Extended Data Fig. 3d). Also in these conditions, very little microtubule release from γ-TuC was detected in the absence of CAMSAP3 (Fig. 3c,d). Strikingly, when SNAP–AF647–CAMSAP3 was added, microtubule nucleation events were frequently followed by specific binding of SNAP–AF647–CAMSAP3 to the γ-TuC-associated microtubule minus end and subsequent minus-end growth (Fig. 3e–g). In some cases, the microtubule remained attached to the glass surface near the γ-TuC that nucleated it (for example, Fig. 3e and Supplementary Video 3), while in other cases, the minus end detached and the microtubule floated away from the γ-TuC (Fig. 3f,g and Supplementary Video 4). Approximately 30% of all γ-TuC-nucleated microtubules acquired a CAMSAP3 signal at their minus end (Fig. 3h, Extended Data Fig. 3e and Supplementary Video 5), and approximately half of these microtubules initiated minus-end growth (Fig. 3i). As a result, the percentage of microtubules released from γ-TuCs increased more than tenfold, to 15% (Fig. 3d). The time interval between the initial CAMSAP3 binding and the onset of microtubule minus end elongation varied between 50 and 350 s (Fig. 3j). After microtubule release, the same γ-TuC could sometimes nucleate another microtubule (Extended Data Fig. 3f and Supplementary Video 6).

**Fig. 3 | CAMSAP3 triggers microtubule release from γ-TuC. a,c,** Maximum intensity projections (**a**) and single frame (top, arrow indicates γ-TuC-anchored microtubule) and representative kymograph (bottom) (**c**) of videos acquired after 20 min of incubation in indicated conditions. In all kymographs, the black arrowheads on top indicate the γ-TuC position and minus and plus indicate the two microtubule ends. **b,** The efficiency (mean ± s.e.m.) of microtubule nucleation by γ-TuC in the absence or presence of SNAP–AF647–CAMSAP3. *n*, the numbers of fields of view from *N* independent experiments. Data points in cyan were acquired from a smaller field of view and the data points in black are from a larger field of view shown in **a. d,** The frequency of microtubule (MT) release from active γ-TuCs in the absence or presence of SNAP–AF647–CAMSAP3. *n*, the numbers of active γ-TuCs from *N* independent experiments. Representative images are shown in **c, e, f** and **l. e–g,l,m,** Microtubule nucleation and release from γ-TuC in 10 min videos obtained in indicated conditions. Single frame and cropped images (**e**). Still frames and kymographs (**f,g,l,m**) (schematic representation in **g**). The insets in **f** (scale bar, 1 μm) show CAMSAP3 signal over time. In **g**, the magenta arrowheads and enlarged views of the kymograph

demonstrate the diffusion (double-sided wavy red arrow) of the released minus end. In **e** and **l**, separate channels for γ-TuC and CAMSAP3 are shown on the right and the magnification is the same as in merged images. In **e, f** and **l**, 0:00 min is the starting point of the video. The yellow arrows indicate γ-TuC and magenta arrows indicate CAMSAP3. **h,** The percentage (mean ± s.e.m.) of γ-TuC-anchored minus ends colocalizing with CAMSAP3, from experiments shown in **e** and **l**. **i,** The percentage (mean ± s.e.m) of microtubules released from γ-TuC colocalizing with CAMSAP3, shown in **h**. *n*, the number of independent experiments and *l*, the total number of active γ-TuCs in **h** and the number of active γ-TuCs colocalizing with CAMSAP3 in **i**, analysed over 10 min. Two-tailed unpaired *t*-tests. **j,** Distribution of time intervals between CAMSAP3 binding and the onset of minus-end elongation. Fourteen γ-TuC dissociation events were pooled from three independent experiments. **k,** Fluorescence intensities (mean ± s.e.m) of all active GCP3–GFP molecules engaged in the indicated events, from experiments represented in **e**, pooled from four independent experiments. One-way ANOVA test with Tukey's multiple comparisons.

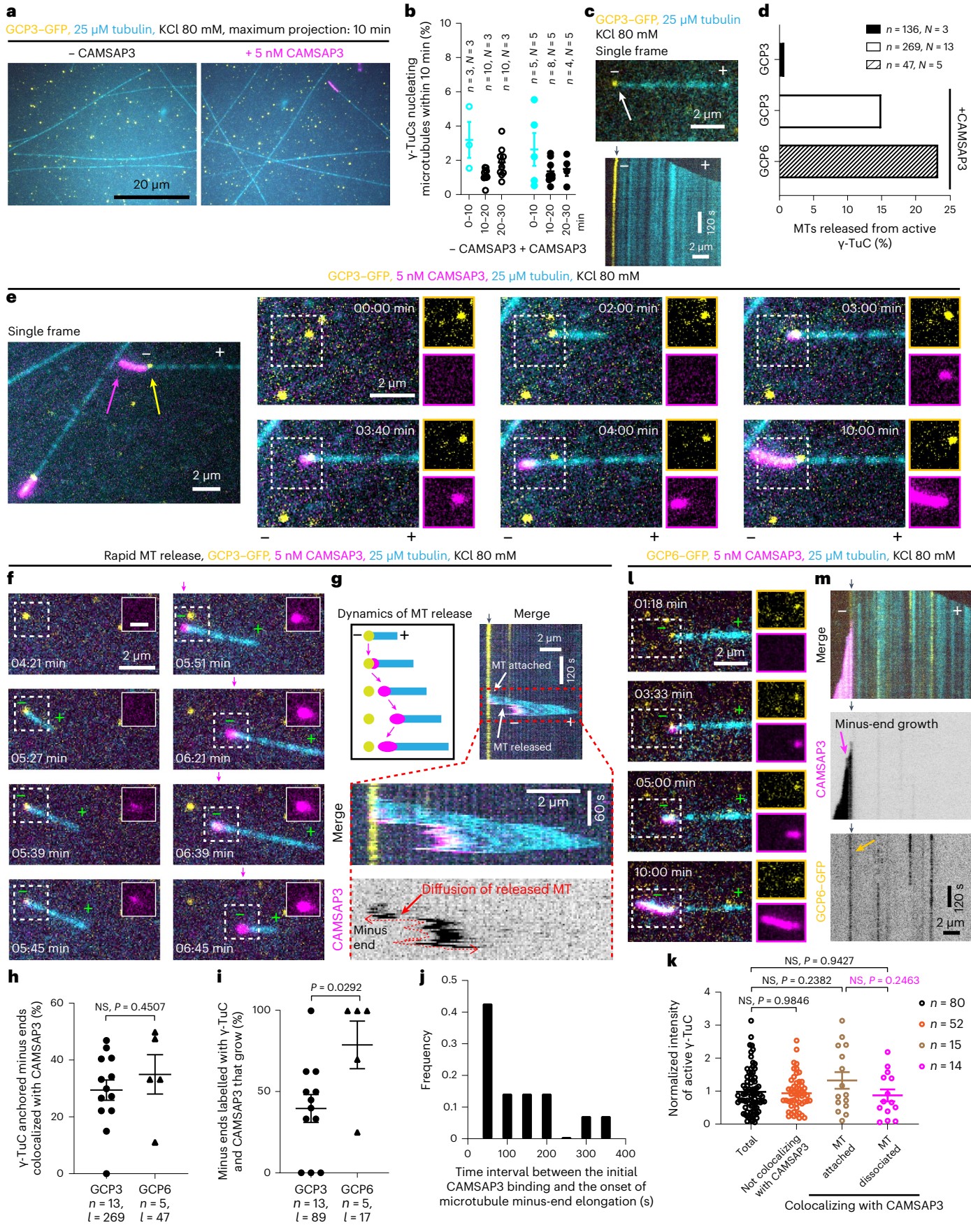

Recent work has shown that also partial/incomplete γ-TuRCs can nucleate microtubules in vitro[22]. To test whether CAMSAP3 preferentially binds to and detaches the minus ends anchored by incomplete γ-TuRCs, we have measured the fluorescence intensity of γ-TuCs in the GFP channel. We found that the intensities of the total active γ-TuC population, γ-TuCs that nucleated microtubules but did not recruit CAMSAP3, and γ-TuCs that did recruit CAMSAP3 after microtubule nucleation and either stayed attached or got released were similar (Fig. 3k). Further, to exclude that microtubules were preferentially released from γ-TuCs lacking the GCP6-containing part of the ring, we generated γ-TuRC that were purified using a homozygous knockin HEK293T cell line where the GCP6 subunit of γ-TuRC was C-terminally tagged with GFP and SII (Extended Data Fig. 3g,h). MS showed that γ-TuC purified from these cells was similar in terms of components and associated proteins to that purified using GCP3–GFP–SII (Extended Data Fig. 3i and Supplementary Tables 2 and 6). We observed that GCP6–GFP–SII-containing γ-TuCs could nucleate microtubules, which then could recruit CAMSAP3, initiate minus-end growth and detach from γ-TuC, and the frequency of CAMSAP3 binding and minus-end growth was slightly higher than for γ-TuCs purified using GCP3–GFP–SII (Fig. 3h,i,l,m and Supplementary Video 7). We conclude that CAMSAP3 can bind to the minus ends of a subset of microtubules nucleated and anchored by γ-TuCs, promote minus-end polymerization and trigger their release.

## Microtubule decoration by CAMSAPs drives γ-TuRC release

Since microtubules rarely dissociated from γ-TuCs spontaneously, their CAMSAP3-induced detachment must be an active process. To get more insight into this process, we compared the effects of CAMSAP2 and CAMSAP3, which decorate and stabilize microtubules grown from the minus end and may also alter microtubule lattice conformation[14,43,45], and CAMSAP1, which tracks the tips of free minus ends but does not decorate them[43,46]. All three CAMSAPs could bind to γ-TuC-anchored minus ends (Fig. 4a–c (CAMSAP colocalization: ~30% for CAMSAP3, ~15% for CAMSAP2 and ~25% for CAMSAP1) and Extended Data Fig. 3d). However, although CAMSAP1 and CAMSAP3 bound to microtubule minus ends equally well, CAMSAP1 had little effect on microtubule release (Fig. 4d,e; γ-TuC displacement frequency: ~40% for CAMSAP3, ~90% for CAMSAP2 and ~5% for CAMSAP1, Supplementary Videos 8 and 9). These data support the view that γ-TuRC is displaced from the minus ends due to their elongation and/or conformational change, which can be driven by CAMSAP2 or CAMSAP3.

## Effects of CDK5RAP2 and CLASP2 on microtubule release

To test whether NPFs affect CAMSAP3 binding and microtubule release from γ-TuRC, we first examined whether they were active in the same conditions. Whereas CDK5RAP2 and CLASP2 could still potentiate γ-TuC-dependent microtubule nucleation up to ~20% and ~9%, respectively, in the presence of 80 mM KCl and 25 μM tubulin, this was not the case for chTOG (Fig. 5a,b). CAMSAP3 binding to γ-TuC-anchored minus ends and subsequent microtubule release could be observed in the presence of either of the three proteins (Fig. 5c–e and Supplementary Video 10). However, CDK5RAP2, but not CLASP2 or chTOG, strongly suppressed CAMSAP3 binding to the minus ends of γ-TuC-nucleated microtubules (Fig. 5f; CAMSAP3 colocalization: ~30% for control, ~3% for CDK5RAP2, ~35% for CLASP2 and ~25% for chTOG). Microtubule minus ends that did recruit CAMSAP3 started to grow and detached from γ-TuC with a comparable frequency in all conditions (Fig. 5g: 40% for control, ~22% for CDK5RAP2, ~46% for CLASP2 and ~20% for chTOG, and Supplementary Videos 10 and 11), though the data for chTOG were less reliable because the combination of high ionic strength and chTOG made microtubule growth events very short lived and thus limited the time when microtubule release could be observed (example 3 in Supplementary Video 11). Altogether, CDK5RAP2 strongly suppressed

CAMSAP3-driven microtubule detachment from γ-TuC, while this was not the case for CLASP2, and no conclusions could be made for chTOG (Fig. 5h).

A global analysis of γ-TuC fluorescence intensities did not provide indications of preferential CAMSAP3 binding and microtubule release from incomplete rather than complete γ-TuRCs (Fig. 3k). We next performed GFP counting of individual γ-TuCs that were engaged in microtubule nucleation, CAMSAP3 binding and microtubule release (Fig. 6). The difference with the experiment shown in Fig. 1a was that here, GFP proteins were not adsorbed on glass directly but immobilized with anti-GFP nanobodies. Unfortunately, this strongly affected the quality of GFP counting, since even GFP–EB3 dimers could not be reliably distinguished from GFP monomers, and the intensities were underestimated (GFP–EB3 was only ~1.3× brighter than GFP) (Fig. 6a,b). This means that complete γ-TuRCs might appear only three to four times brighter than a single GFP. Still, this approach provided some indication of the number of GCP3–GFP molecules present in individual complexes. It revealed that CDK5RAP2 and CLASP2 by themselves had no noticeable effect on the size distribution of the γ-TuCs, indicating that they did not induce oligomerization of γ-TuSCs into γ-TuRC (distributions are similar in Fig. 6c–e). Rather, they preferentially activated γ-TuCs with more GCP3–GFP subunits (Fig. 6f). Active γ-TuCs in the presence of CDK5RAP2 and CLASP2 probably included substantial amounts of complete γ-TuRCs (Fig. 6f; γ-TuCs corresponding to three to four GFPs constituted ~30% of the active population in control, ~56% with CDK5RAP2 and ~49% with CLASP2). Without NPFs, the fluorescence intensity of the complexes that bound CAMSAP3 and released microtubules was very similar to that of the complexes that did not bind CAMSAP3, suggesting that CAMSAP3 recruitment and microtubule release did not preferentially occur on partial complexes (Fig. 6g–i). In the presence of CDK5RAP2, very few events of CAMSAP3 binding and microtubule release were observed, and these occurred with complexes that contained fewer GCP3–GFP subunits than CDK5RAP2-activated γ-TuRCs that did not recruit CAMSAP3 (Fig. 6j). No such effect was observed with CLASP2: fluorescence intensity distribution was the same for the γ-TuCs that displayed CAMSAP3 binding and microtubule release, and those that did not. CLASP2-activated γ-TuCs that colocalized with CAMSAP3 or released microtubules exhibited intensities falling within the range of complete γ-TuRCs (Fig. 6k). These data suggest that both CDK5RAP2 and CLASP2 preferentially activate complete γ-TuRCs, but only CDK5RAP2 protects such γ-TuRCs from CAMSAP3 binding and microtubule detachment.

## CDK5RAP2 regulates CAMSAP2-bound microtubules in cells

Next, we examined whether CDK5RAP2 affects the number of CAMSAP-stabilized microtubule minus ends in cells. Our previous work showed that in RPE1 cells, CAMSAP2 is the major CAMSAP isoform, and CAMSAP2-bound microtubule minus ends are often attached to the Golgi membranes by binding to the scaffolding protein AKAP450 (refs. 43,47). Since crowding at the Golgi might complicate the quantification of CAMSAP2-positive microtubule minus ends, we used cells where AKAP450 was knocked out[47] and found that it did not have a significant effect on the number of CAMSAP2 stretches per cell area or microtubule density (Fig. 7a,b and Extended Data Fig. 4a–f). In contrast, simultaneous knockout of CDK5RAP2 and its paralogue myomegalin (MMG), which might be redundant, significantly increased the number of CAMSAP2-stabilized microtubules in AKAP450 knockout cells, without a major effect on microtubule density (Fig. 7a,b and Extended Data Fig. 4a–f). Since no microtubules are attached to the Golgi in such cells, it means that the microtubule network became less radial (Extended Data Fig. 4d). Conversely, stable overexpression of GFP–CDK5RAP2 (Extended Data Fig. 4a), which localized to the centrosomes in all cells and to the Golgi membranes in control but not in AKAP450 knockout cells (Fig. 7a and Extended Data Fig. 4d), significantly reduced the

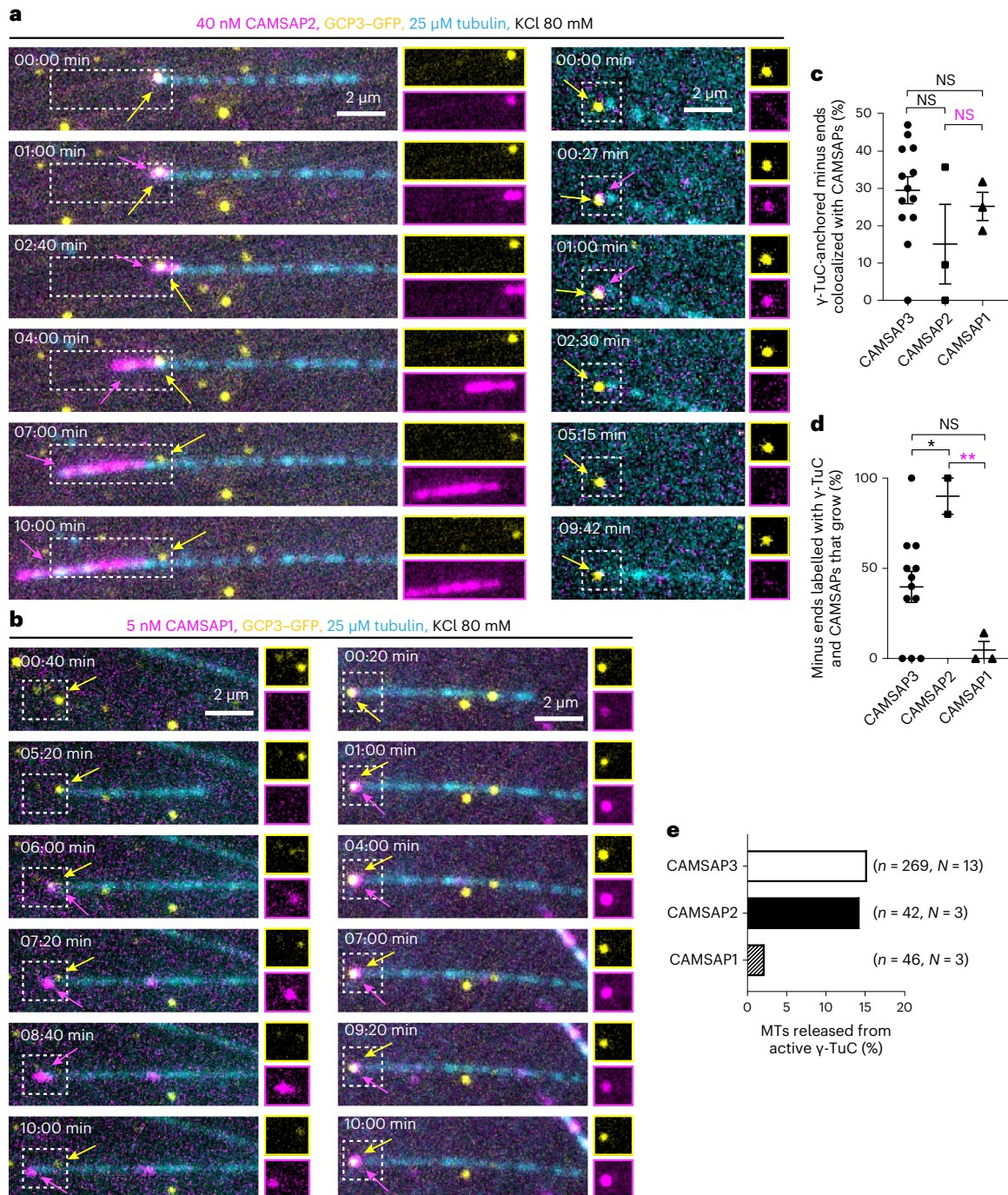

**Fig. 4 | CAMSAPs cause γ-TuRC detachment by decorating growing microtubule minus ends. a,b,** Still frames (with 0:00 min being the starting point of the video) from a 10 min video showing two different examples of γ-TuC interplay with CAMSAP2 (**a**) or CAMSAP1 (**b**) in the indicated conditions. Next to the merged images, individual channels (magnification is the same as merged images) for γ-TuC (top, yellow) and CAMSAPs (bottom, magenta) are shown for the ROIs marked with white rectangles. Left: microtubule (cyan) release and right: occasions when CAMSAP2 (**a**) or CAMSAP1 (**b**) do not displace γ-TuC from microtubule minus ends. The yellow arrows indicate γ-TuC, while the magenta arrows indicate CAMSAPs. **c,** Colocalization frequency (mean ± s.e.m.) of γ-TuC-anchored microtubule minus ends with CAMSAPs, from experiments shown in **a** and **b**. The data points represent single experiments from which the percentage of growing microtubule minus ends that were labelled with γ-TuC and CAMSAPs were quantified. $n = 3$ and $l = 42$, NS $P = 0.1068$ for CAMSAP2 and $n = 3$ and $l = 46$, NS $P = 0.6149$ and NS $P = 0.3622$ (magenta) for CAMSAP1. **d,** The

percentage (mean ± s.e.m.) of microtubules released from γ-TuCs colocalizing with CAMSAPs, as shown in **a** and **b**. $n = 3$ and $l = 7$, *$P = 0.0277$ for CAMSAP2 and $n = 3$ and $l = 12$, NS $P = 0.0629$ and **$P = 0.0037$ (magenta) for CAMSAP1. Control data (CAMSAP3 values) are from Fig. 3h ($n = 13$, $l = 269$) for **c** and from Fig. 3i ($n = 13$, $l = 89$) for **d** (GCP3 values), replotted here for comparison. $n$, the number of independent experiments and $l$, the total number of active γ-TuCs in **c** and the number of active γ-TuCs colocalizing with CAMSAPs in **d**, analysed over 10 min duration. One-way ANOVA with uncorrected Fisher's least significant difference (LSD) tests were used to compare the means with each other. **e,** The frequency of microtubule (MT) release from active γ-TuC in the presence of CAMSAPs under the experimental conditions shown in **a** and **b**. Data for CAMSAP3 are from Fig. 3d (GCP3 values with CAMSAP3), replotted here for comparison. The numbers of active γ-TuCs analysed, $n$, pooled from $N$ independent experiments, are indicated.

abundance of CAMSAP2-labelled minus ends in all analysed cell lines (Fig. 7a,b and Extended Data Fig. 4c).

To address whether CAMSAP2 stretches arise soon after microtubule nucleation, as predicted by our in vitro assays, we performed nocodazole washout assays[43]. We found that, as expected, simultaneous knockout of CDK5RAP2 and MMG, but not AKAP450, decreased microtubule nucleation from the centrosome, and this effect was fully rescued by overexpressing GFP–CDK5RAP2 (Fig. 7c,d). CAMSAP2-positive microtubule minus ends were visible around the centrosome already 1 min after nocodazole washout (Fig. 7e, see also ref. 48), but their emergence was strongly suppressed in cells stably overexpressing GFP–CDK5RAP2 (Fig. 7e and Extended Data Fig. 4g). In cells lacking both CDK5RAP2 and MMG, the abundance of CAMSAP2 stretches around the centrosome was significantly increased compared with control RPE1 or AKAP450 knockout cells when the lower microtubule nucleation efficiency in AKAP450/CDK5RAP2/MMG knockout cells was taken into account (Fig. 7d,f and Extended Data Fig. 4g). CDK5RAP2 thus suppresses microtubule minus end release from the centrosome and generation of CAMSAP2-stabilized non-centrosomal microtubules.

### CDK5RAP2 promotes microtubule capping by γ-TuRC

We hypothesized that CDK5RAP2 exerts its effect by altering γ-TuRC geometry so that it would be more similar to a 13-protofilament microtubule. We first tested this possibility by performing negative-stain transmission EM of γ-TuRC, either alone or incubated in the presence of 120 nM CDK5RAP2 (Extended Data Fig. 5). Three-dimensional reconstructions of density maps of γ-TuRC alone fitted well into the density map of a published model (from the Protein Data Bank (PDB) ID: 6V6S (ref. 20)) (Extended Data Fig. 5a–c,i), but the addition of 120 nM CDK5RAP2 did not cause any noticeable differences in the γ-TuRC structure (Extended Data Fig. 5d–j). Reconstructions with and without CDK5RAP2 substantially deviated from the closed conformation of γ-TuSC oligomers (from PDB ID: 5FLZ (ref. 49)), and did not match the microtubule geometry (from PDB ID: 2HXF ref. 50 and EMD-5193 (ref. 51)) (Extended Data Fig. 5h). We note that the densities of terminal γ-TuSC at the γ-TuRC seam and in the luminal bridge were not clearly resolved in our reconstructions and cannot make conclusions about potential conformational changes at these sites.

As an alternative approach, we used a microtubule-capping assay (Fig. 8a), whereby γ-TuCs bound to the ends of preformed microtubules in the presence of 5 μM tubulin. Microtubules were monitored for 5 min to distinguish plus and minus ends by their growth behaviour, and if the microtubule end that did not display fast growth dynamics was stably bound to γ-TuC for at least 2 min, the microtubule was scored as γ-TuC capped. We compared Taxol-stabilized microtubules, which in our hands have 12 or 13 protofilaments with guanylyl-(a,b)-methylene-diphosphonate (GMPCPP)- and docetaxel-stabilized

microtubules, which predominantly have 14 protofilaments[52]. The overall capping efficiency was in the range of ~30–60% (Fig. 8b,c), which was higher than the ~20–25% efficiency reported in the absence of free tubulin in microtubule gliding assays[53]. Whereas the above-mentioned study found no difference in capping of Taxol- or GMPCPP-stabilized microtubules[53], in our experimental conditions, Taxol-stabilized microtubules were capped by γ-TuCs more efficiently than docetaxel-stabilized ones, and GMPCPP-stabilized were capped even less well, possibly because they have more curved terminal protofilaments than the taxane-stabilized ones[54,55] (Fig. 8b,c). It is possible that free tubulin, present in our assays, incorporates into microtubule ends and affects the microtubule–γ-TuRC interface, thus increasing the capping efficiency and the sensitivity of capping to lattice geometry. Interestingly, the addition of CDK5RAP2 significantly inhibited capping of both types of 14-protofilament microtubules, without affecting the capping efficiency of Taxol-stabilized microtubules (Fig. 8b,c). These data are in line with the idea that CDK5RAP2 promotes a conformational change or conformational flexibility in γ-TuRC that would allow a better match with the 13-protofilament microtubule structure.

In addition to altering γ-TuRC conformation, CDK5RAP2 could also act at the interface between γ-TuRC and the newly nucleated microtubule. It was previously shown that CDK5RAP2 can interact with EB1, track growing microtubule plus ends and regulate microtubule dynamics[56,57]. In vitro reconstitution of microtubule dynamics using GMPCPP-stabilized seeds showed that although CDK5RAP2 did not bind to microtubules on its own, it was readily recruited to microtubules by EB3 (Fig. 8d,e). In these assays, CDK5RAP2 frequently tracked growing minus ends and occasionally tracked growing plus ends together with EB3. It also bound along microtubule shafts, suggesting that it has some microtubule affinity (Fig. 8e). Therefore, although CDK5RAP2 does not bind to microtubules autonomously, it can interact with microtubule ends and shafts in the presence of a microtubule-binding partner, possibly though the disordered basic and serine-rich region that contains the EB-binding site[56]. It is thus possible that CDK5RAP2 can bind to microtubule minus ends at the interface with γ-TuRC, thereby interfering with CAMSAP binding.

## Discussion

In this study, we have uncovered a mechanism of generation of free and stable microtubule minus ends through CAMSAP-driven displacement of γ-TuRC from newly nucleated microtubules. Our in vitro reconstitution assays showed that CAMSAPs are sufficient to mediate microtubule release from γ-TuRC. CAMSAPs are diffusely distributed in mammalian cells when microtubules are depolymerized[43] and associate with microtubule minus ends released from the nucleation centres such as the centrosome or the Golgi[43,47,48]. Microtubule release can, in principle,

---

**Fig. 5 | CDK5RAP2, but not CLASP2 inhibits CAMSAP3 binding to the minus ends of γ-TuRC-anchored microtubules and their release. a**, Maximum intensity projections of 10 min time-lapse videos, acquired after 20 min of incubation, showing microtubules nucleated from γ-TuC in the indicated conditions. **b**, The average microtubule nucleation efficiency (mean ± s.e.m.) from experiments shown in **a**. The numbers of fields of view analysed, *n*, from *N* independent experiments are indicated. Control data are from Fig. 3b, replotted here for comparison, colours of the data points are the same as in Fig. 3b. **c–e**, Still frames from 10 min videos (0:00 min is the starting point of the video) showing CAMSAP3 binding to the γ-TuC-anchored microtubule minus ends under the indicated experimental conditions. Left: microtubule release from γ-TuC (yellow arrows) colocalizing with CAMSAP3 (magenta arrows). Right: CAMSAP3 binding without microtubule release. The insets show cropped individual channels for γ-TuC (left) and CAMSAP3 (right), magnification is the same as merged images. Left: microtubule release from incomplete γ-TuRC (dim GFP signal) (**c**). Left: γ-TuC dissociation from minus end and also from glass surface within 6 min,

followed by rescue at the microtubule plus end from CAMSAP3-stabilized stretch (**e**). **f**, The percentage (mean ± s.e.m) of γ-TuC-anchored microtubule minus ends colocalizing with CAMSAP3, from experiments represented in **a** and **c–e**. **g**, The percentage (mean ± s.e.m) of microtubules released from the γ-TuC colocalizing with CAMSAP3, shown in **f**. *n*, the number of independent experiments (plotted) and *l*, the total number of active γ-TuCs in **f** and the number of active γ-TuCs colocalized with CAMSAP3 in **g**, analysed over 10 min duration. Control data are from Fig. 3h for **f** and from Fig. 3i for **g** (GCP3 values), replotted here for comparison. One-way ANOVA with uncorrected Fisher's LSD tests were used to compare the means with control. In **b** and **f–h**, control is black, CDK5RAP2 is red, CLASP2 is blue and chTOG is purple. **h**, The frequency of microtubule (MT) release from active γ-TuC in the presence of SNAP–AF647–CAMSAP3 under the experimental conditions shown in **a** and **c–e**. *n*, the numbers of active γ-TuCs analysed from *N* independent experiments. Data for GCP3 with or without CAMSAP3 are from Fig. 3d, replotted here for comparison. CDK5RAP2 is abbreviated as C5R2 in the plots.

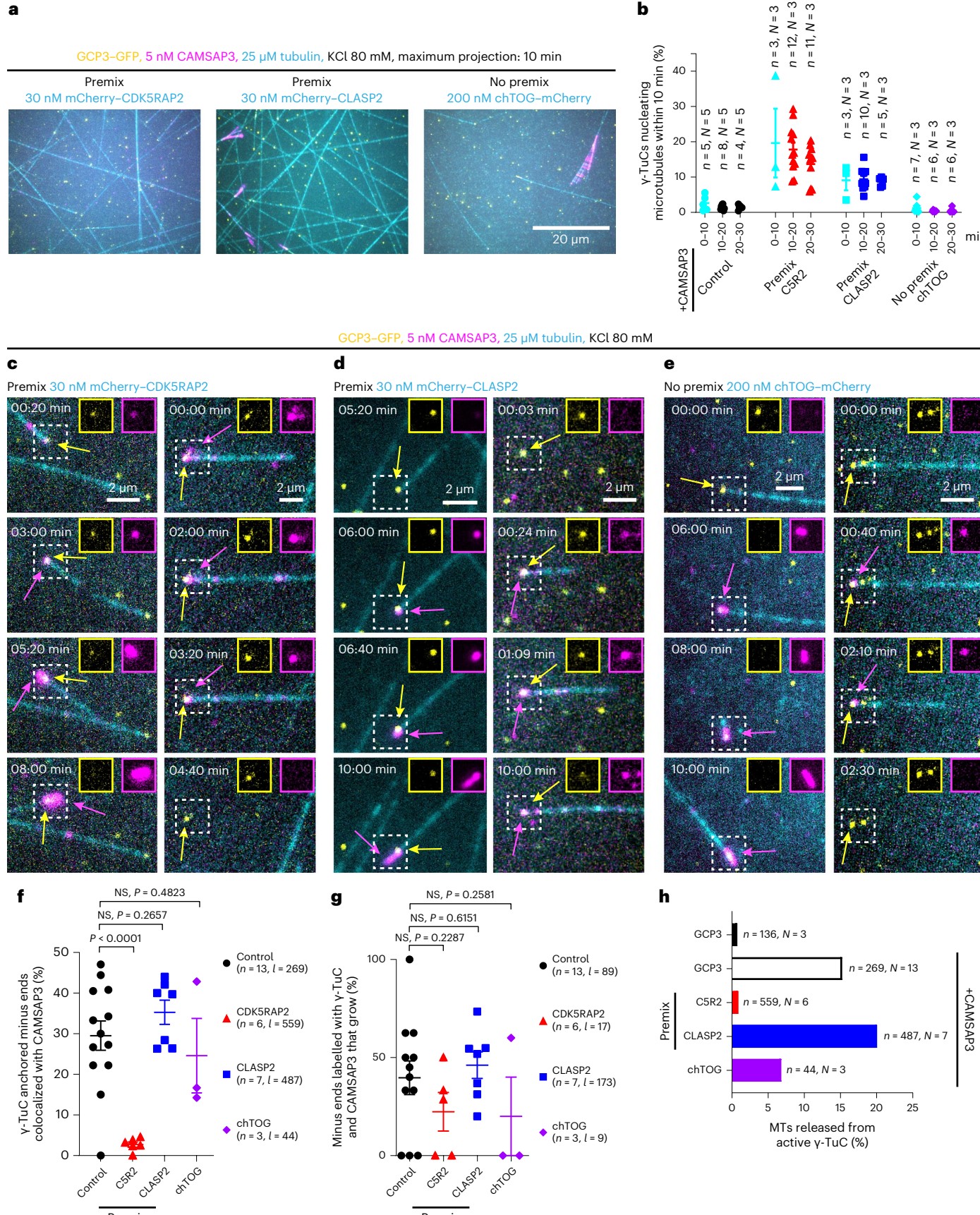

be induced by severing enzymes; however, previous work showed that the severing enzyme katanin acts after its binding partner CAMSAP3 is recruited to microtubule minus ends close to the centrosome[48]. Since mammalian katanin requires microtubule-bound cofactors for its activity[58], it probably acts downstream of CAMSAPs to control the length and the number of CAMSAP-decorated microtubules[43].

CAMSAPs recognize the minus ends through the CKK domain, which binds to a site between two flared protofilaments[14], an arrangement that would not occur at microtubule minus ends fully attached to γ-TuRC[16,17]. However, given the asymmetric γ-TuRC structure[18–21], it is possible that some protofilaments at the γ-TuRC-bound minus end might not be capped by γ-tubulin but are just laterally associated with the capped ones and can acquire a curved conformation (Fig. 8f). Such conformation would permit CAMSAP binding, followed by CAMSAP-driven protofilament stabilization and elongation. Protofilament extension at the minus end would generate a pushing force, similar to growing microtubule plus ends[59], potentially causing γ-TuRC detachment. Furthermore, since γ-TuRC displacement depends on the ability of CAMSAP isoforms to stably decorate minus-end-grown microtubule lattice, it is possible that CAMSAP-driven lattice changes, such as axial expansion[45] or altered protofilament skew[14], would propagate to the neighbouring protofilaments, perturb the microtubule–γ-tubulin interface and trigger γ-TuRC detachment.

To control the abundance and positioning of free minus ends, microtubule detachment from γ-TuRC is expected to be tightly regulated. We found that two factors promoting microtubule nucleation, CDK5RAP2 and CLASP2, play a role in microtubule release. CDK5RAP2 is well known for its ability to bind and activate γ-TuRC through the domain called γ-TuRC nucleation activator (γ-TuNA)[19,27,28,60]. Whereas micromolar concentrations of purified γ-TuNA were needed to activate γ-TuRC[60], full-length CDK5RAP2 could exert this effect already at 30 nM, consistent with previous work[27]. CDK5RAP2 suppressed CAMSAP binding to γ-TuC-anchored minus ends and γ-TuC capping of 14- but not 13-protofilament microtubules. This could be due to the ability of CDK5RAP2 to promote closure of the γ-TuRC ring to match the 13-protofilament microtubule geometry, and indeed, previous structural work demonstrated that the α-helical centrosomin motif 1 of γ-TuNA binds to an interface of GCP2 and MZT2 at the outer surface of γ-TuRC and could promote a shift to a more 'closed' γ-TuRC structure[24,61]. Unfortunately, we were not able to find further structural support for this idea, possibly due to technical reasons. We note that the GFP counting data show that not all γ-TuCs activated by CDK5RAP2 in our assays are complete rings, and CDK5RAP2 must thus be able to activate partial γ-TuRCs and also suppress microtubule release from incomplete γ-TuRC. Since previous work[56,57] and our in vitro data indicate that CDK5RAP2 has some microtubule affinity, CDK5RAP2 may directly stabilize the interface between γ-tubulin and the newly nucleated microtubule and interfere with CAMSAP binding.

Using overexpression and knockout experiments, we found direct support for the role of CDK5RAP2 in suppressing CAMSAP-mediated

microtubule release in cells. CDK5RAP2 is a potent, but not essential, γ-TuRC activator: cells lacking both CDK5RAP2 and its paralogue MMG are viable[47] and have normal microtubule density in interphase, although their ability to nucleate microtubules after nocodazole-induced disassembly is reduced. This is consistent with studies that showed only a minor reduction in γ-tubulin signal at centrosomes in CDK5RAP2 knockout cells[62,63] and that homozygous mutations in the CDK5RAP2-encoding gene in humans are not lethal, though they cause developmental disorders such as microcephaly and Seckel syndrome, probably due to perturbations of cell division[64,65]. In interphase epithelial cells, the role of CDK5RAP2 in maintaining the microtubule pool is thus relatively minor, but it controls the balance between centrosomal and non-centrosomal, CAMSAP-stabilized microtubules. Formation of non-centrosomal microtubules by γ-tubulin-dependent nucleation followed by CAMSAP-mediated release is supported by a previous study in neurons showing a 70% reduction in CAMSAP2 intensity upon γ-tubulin depletion[66]. Still, we cannot exclude that CAMSAP-stabilized minus ends are generated in CDK5RAP2 knockout cells in a γ-tubulin-independent manner, since formation of microtubules dependent on CLASP1, chTOG and CAMSAPs has been described in γ-tubulin-depleted cells[67], and future studies would be needed to dissect different microtubule nucleation pathways.

Since CDK5RAP2 and MMG are not essential, there must be other mechanisms controlling γ-TuRC activation. On the basis of our results, one such mechanism involves CLASP2. Previous work has shown that CLASP2 potently promotes formation of complete tubes from incomplete tubulin assemblies and enables microtubule outgrowth from seeds at low tubulin concentration[29,68]. CLASP2 thus probably acts by inducing or stabilizing microtubule nucleation intermediates rather than by affecting γ-TuRC geometry, and this would explain why CLASP2 did not inhibit CAMSAP binding to γ-TuRC-anchored minus ends and microtubule release.

Unlike CDK5RAP2, the loss of CLASPs in cells causes a strong decrease in microtubule density; this must be due to multiple effects, such as suppression of catastrophes, induction of rescues and stimulation of microtubule repair[69,70]. Still, control of nucleation and formation of non-centrosomal microtubules, particularly at non-centrosomal sites such as the Golgi membranes[71], where other γ-TuRC activators are less abundant, are likely to play an important role in CLASP-mediated regulation of microtubule numbers. CLASP-mediated activation of γ-TuRC-dependent microtubule nucleation with subsequent microtubule release by CAMSAPs would be a mechanism to generate numerous CAMSAP-stabilized non-centrosomal microtubules. Indeed, simultaneous depletion of CLASP1 and CLASP2 in cells leads to a very strong loss of CAMSAP2-bound microtubule minus ends[47].

Altogether, our work suggests that repeated activity of γ-TuRC in the presence of CAMSAP2 or CAMSAP3 can lead to generation of a pool of stable microtubule minus ends that are not directly attached to their nucleation sites. This 'handover' mechanism, which can affect

**Fig. 6 | Quantification of the number of GCP3–GFP molecules in the nucleation assays with γ-TuCs in the absence or presence of NPFs.**
**a**, Representative images of single molecules of the indicated purified proteins (GFP in dark green, GFP–EB3 in light green and GCP3–GFP in yellow) immobilized on coverslips with anti-GFP nanobody. **b**, Histograms of single-molecule fluorescence intensities shown for one experiment represented in **a**. $n$ = 12,841 for GFP, $n$ = 21,670 for GFP–EB3 and $n$ = 16,420 for GCP3–GFP, where $n$ is the number of molecules analysed. **a** and **b** are representative of six independent experiments that yielded similar results. **c**–**e**, Averaged histogram of weights of $N$-mers of GFP determined from the fitting to the intensities of GCP3–GFP puncta (fitting similar to shown in Fig. 1a, right), showing the numbers of GFP molecules per GCP3–GFP puncta in control (black) (**c**) or in the presence of either mCherry–CDK5RAP2 (red) pre-incubated with γ-TuC (**d**) or mCherry–CLASP2 (blue) pre-incubated with γ-TuC (**e**). Tubulin and SNAP–AF647–CAMSAP3 in the

experimental conditions identical to Fig. 5a. For control, $N$ = 6, $n$ = 12,841 for GFP and $n$ = 16,420 for GCP3–GFP; for premix CDK5RAP2, $N$ = 3, $n$ = 6,286 for GFP and $n$ = 14,768 for GCP3–GFP; for premix CLASP2, $N$ = 4, $n$ = 8,530 for GFP and $n$ = 17,023 for GCP3–GFP, where $N$ is the number of independent experiments and $n$ is the number of molecules analysed. The plots present mean ± s.d. **f**–**h**, Histograms of the number of GFP molecules per GCP3–GFP puncta that were active (nucleating microtubules (MTs)) (**f**), active and colocalizing with CAMSAP3 (**g**) or active and releasing microtubules upon CAMSAP3 binding to the minus end (**h**), determined from the plots and experiments shown in **c**–**e**. **i**–**k**, Histograms of the number of GFP molecules per GCP3–GFP puncta that were active (nucleating microtubules), colocalizing with CAMSAP3 and releasing microtubules in control (**i**), premix CDK5RAP2 (**j**) and premix CLASP2 (**k**). Values are replotted here from **f**–**h**. The colour code in plots **c**–**k**: control, black; CDK5RAP2, red and CLASP2, blue.

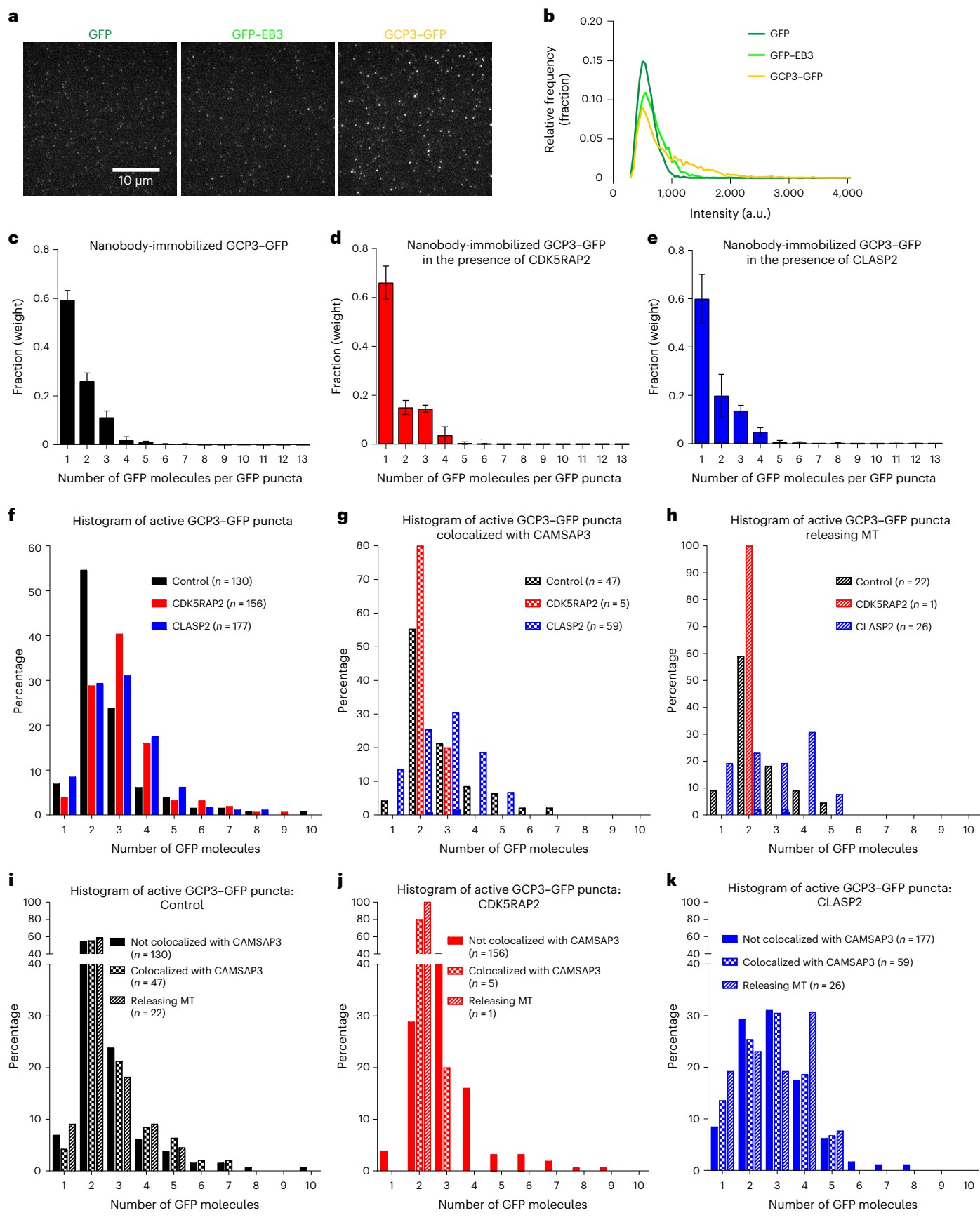

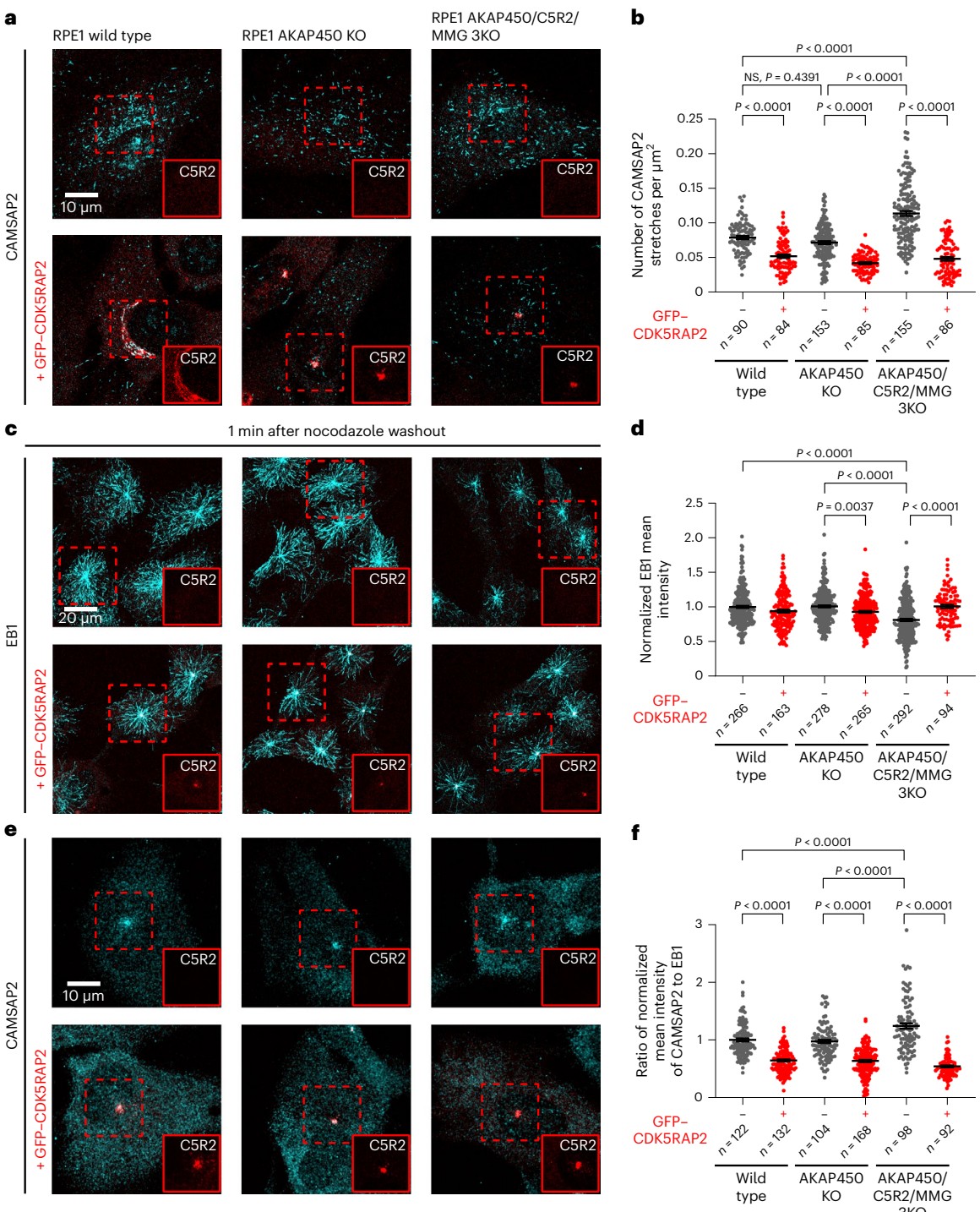

**Fig. 7 | CDK5RAP2 regulates the abundance of CAMSAP2-bound microtubules in cells. a,c,e,** Representative immunofluorescence images of the indicated RPE1 cell lines with or without stable overexpression of GFP–CDK5RAP2 and stained for CAMSAP2 or EB1, as indicated in untreated cells (**a**) and 1 min after nocodazole washout (**c** and **e**). The insets show cropped GFP–CDK5RAP2 channel in red, magnification is the same as merged images. Cells with clearly visible GFP signal at the centrosome (or the centrosome and the Golgi in the wild-type cells) were selected for the analysis in all three cell lines. **b,** The number of CAMSAP2 stretches per square micron area (mean ± s.e.m.) quantified from experiments represented in **a** and using values from Extended Data Fig. 4b,c. **d,** EB1 mean intensity (mean ± s.e.m.) normalized to wild-type average quantified from experiments represented in **c. f,** The intensity ratio (mean ± s.e.m.) of CAMSAP2 over EB1 normalized to wild-type average quantified from experiments represented in **e** and using values from panel **d** and Extended Data Fig. 4g. In all plots, the numbers of cells analysed per genotype, *n*, from three independent experiments are indicated, and one-way ANOVA test with Tukey's multiple comparisons corrected for multiple testing was used. C5R2, CDK5RAP2; KO, knockout.

microtubule organization and increase microtubule density through efficient reuse of a limited number of microtubule nucleation sites, can be directly controlled by the composition of the nucleation sites themselves, because different γ-TuRC activators can differentially affect the destiny of the newly generated microtubule minus ends.

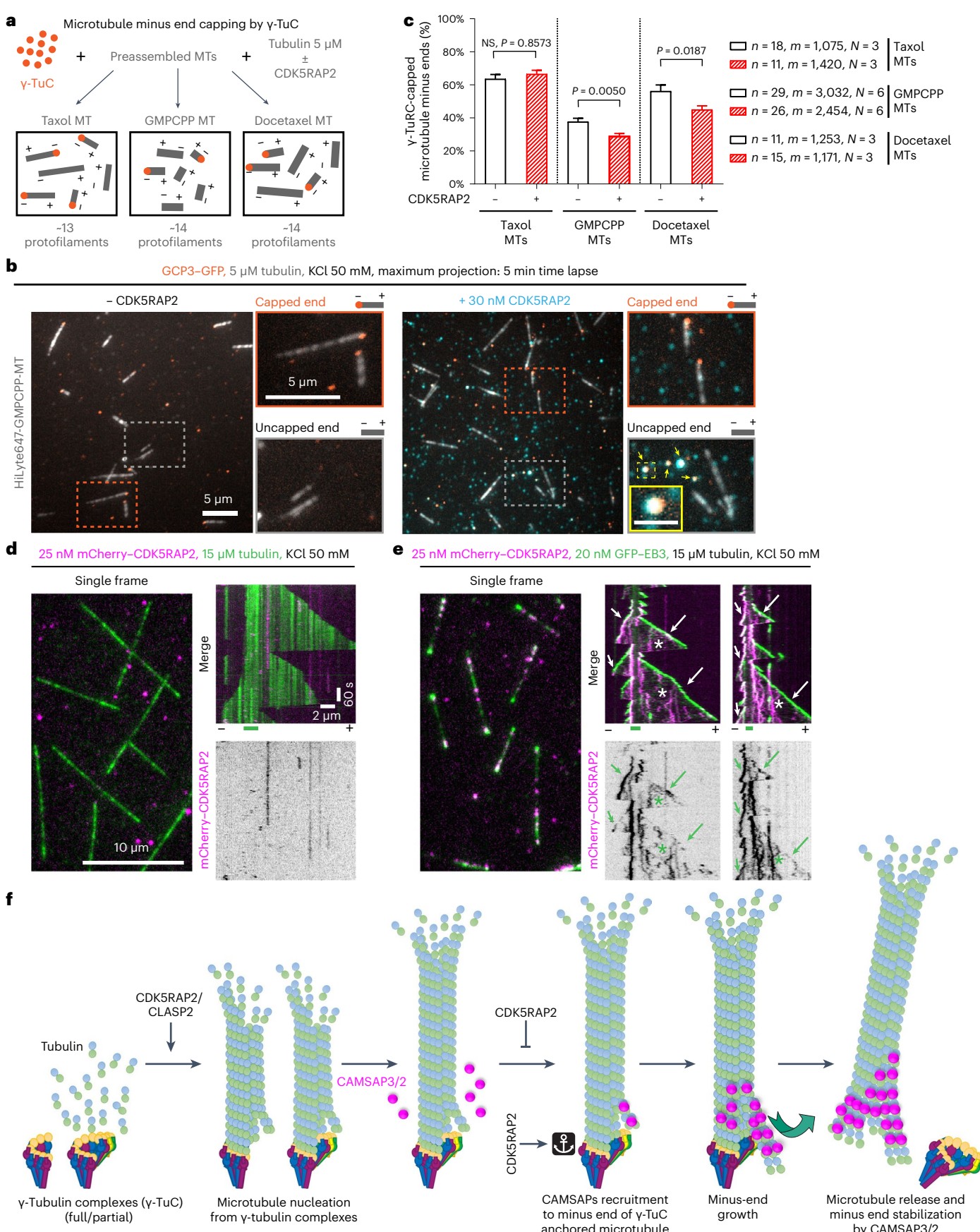

**Fig. 8 | Effects of CDK5RAP2 on microtubule capping and dynamics and a model of CAMSAP-driven γ-TuRC detachment. a**, A scheme of the γ-TuRC capping assays. **b**, Maximum intensity projections of 5 min videos showing γ-TuC capping of GMPCPP-stabilized microtubules (MTs), in the presence of tubulin with (right) or without (left) mCherry–CDK5RAP2. The enlarged views show capped (top) and uncapped minus ends (bottom), distinguished from the plus ends by the absence of growth or slow growth dynamics. The yellow arrowheads and inset (scale bar, 1 μm) at the right bottom shows colocalization of γ-TuC with CDK5RAP2. **c**, Minus-end capping efficiency (mean ± s.e.m.) of γ-TuC for stabilized microtubules with different protofilament numbers in absence or presence of mCherry–CDK5RAP2 from experiments represented in **a** and **b**. *n*, number of fields of view analysed (plotted) and *m*, number of microtubule minus ends analysed from *N* independent experiments, as indicated. NS, one-way ANOVA test with Šídák's multiple comparisons corrected for multiple testing. **d**,**e**, Single frames and representative kymographs from 10 min videos, showing microtubules growing from GMPCPP-stabilized microtubule seeds (short green lines below kymographs) in the presence of either mCherry–CDK5RAP2 and tubulin only (**d**) or together with

GFP–EB3 (**e**) in the indicated conditions. Fluorescent tubulin was substituted with unlabelled tubulin in the assays with GFP–EB3. The arrowheads show CDK5RAP2 binding to microtubule minus ends, arrows show binding to plus ends and asterisks show binding to microtubule lattice. Minus and plus represent the two microtubule ends. The magnification is the same in **d** and **e**. **f**, A model of microtubule NPFs, CDK5RAP2 and CLASP2, which can activate full and partial γ-TuRCs to nucleate a microtubule. The minus end of such microtubules may or may not be fully anchored to γ-TuC allowing some protofilaments that are not attached to the γ-tubulin subunits to attain a flared conformation permissive for CAMSAP binding. CAMSAP2 or 3 bind to the minus end of the γ-TuRC-capped microtubule at an intradimer site between two protofilaments[14], where they can promote minus-end polymerization, stabilize the growing minus end or alter lattice conformation. Elongating protofilaments at the minus end can generate a pushing force or can alter the conformation of the microtubule lattice[14,45], causing detachment of neighbouring protofilaments from γ-TuRC and microtubule release. CDK5RAP2 inhibits microtubule release by suppressing CAMSAP binding, probably by modifying the γ-TuRC conformation and/or γ-TuRC–microtubule interface.

## Online content

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

## Methods

### DNA constructs

We used a previously described SII–mCherry–CLASP2 construct[29]. The chTOG construct was a gift from S. Royle (University of Warwick). chTOG–mCherry–SII was made by cloning the full-length construct in a modified pTT5 expression vector (Addgene no. 44006) with a C-terminus mCherry–SII. GFP–CDK5RAP2 was a gift from Robert Z. Qi (The Hong Kong University of Science and Technology). SII–mCherry–CDK5RAP2, SII–SNAP–CAMSAP3, Biotinylation tag (Bio)-tobacco etch virus protease cleavage site (TEV)–mCherry–CAMSAP2 and SII–SNAP–CAMSAP1 were made by cloning the full-length constructs[43] in modified C1 vectors with either a SII–mCherry, a SII–SNAP or a Bio-TEV–mCherry tag at the N-terminus.

### Cell lines and cell culture

We used the following previously published cell lines: HEK293T cells (American Type Culture Collection, cat. no. CRL-3216), hTERT immortalized RPE-1 (RPE1) wild-type cells (American Type Culture Collection, cat. no. CRL-4000), RPE1 AKAP450 knockout and RPE1 AKAP450/CDK5RAP2/MMG triple knockout cell lines[47], and RPE1 wild-type and RPE1 AKAP450 knockout transgenic cell lines stably expressing GFP–CDK5RAP2 (ref. 72). All these cell lines were cultured in Dulbecco's modified Eagle medium/Ham's F10 media (1:1) supplemented with 10% fetal calf serum and 1% antibiotics (penicillin and streptomycin). The cell lines used were not found in the commonly misidentified cell lines database maintained by the Immunization Coalition of Los Angeles County. No further cell line authentication was performed. The cell lines were routinely checked for mycoplasma contamination using the LT07–518 Mycoalert assay. Polyethylenimine 'Max' (PEI Max, Polysciences) was used to transfect HEK293T cells with plasmids for Strep-Tactin- and streptavidin-based protein purification at a 3:1 ratio of PEI Max:plasmid.

### Generation of homozygous HEK93T knockin cell lines endogenously tagged with a GFP–SII for GCP3 and GCP6

GCP3–GFP–SII and GCP6–GFP–SII knockin cell lines were generated using CRISPR–Cas9 technology[73]. To generate knockin cell lines, CRISPR guide RNAs were designed using the web tool from the Zhang lab (https://www.zlab.bio/resources). The guide RNAs (gRNAs) were designed to overlap with the stop codon to disrupt the recognition site after insertion of the tag avoiding any further cleavage. Annealed oligos were inserted into pSpCas9(BB)–2A-Puro (px459, Addgene 62988) using BbsI. Donor plasmids were designed by selecting 800–1,000 bp of homology flanking both sides of the stop codon of the targeted gene. The two homology arms were obtained by genomic DNA PCR from HeLa cells. The GFP–SII tag was amplified by PCR using primers with complementary domains for the homology arms. Using Gibson assembly, the two homology arms and GFP–SII were cloned into the donor plasmid. FuGENE6 (Roche) was used to cotransfect cells with px459 containing humanized Cas9, guide RNA followed by tracrRNA and a puromycin resistance marker together with a donor construct. At 24 h post-transfection, cells were selected for 2 days using 2 µg ml$^{-1}$ puromycin and subsequently subcloned to a single-cell dilution. Positive clones were confirmed using immunofluorescence, genomic DNA PCR genotyping and western blotting. The following guide RNA sequences and primers were used: for GCP3, gRNA, 5′-GGACCGCGAGCTTCACGTGT-3′; 5′-homology arm, 5′-TCAACACAGCAGAGCCTGTGC-3′ and 5′-CGTGTGGGAGCTGCGCCGCC-3′; 3′-homology arm, 5′-AGCTCGCGGTCCTCCCAGGG-3′ and 5′-CGAATGCATCTGAAAGATAATTGC-3′; and genotyping, 5′-GGAAGGAAAAACAGACCCAACC-3′ and 5′-CGAATGCATCTGAAAGATAATTGC-3′. For GCP6, gRNA, 5′-CAGAGCAGCCTCAGGCGTCC-3′; 5′-homology arm, 5′-TTTCTGCCTAGCTTGGAGCTG-3′ and 5′-GGCGTCCTGGTAGTAGTTGTTGAAGTTG-3′; and 3′-homology arm, 5′-GGCTGCTCTGCGGGGGAC-3′ and 5′-CTACAGGCGTACAGGTGAGC-3′.

### Lentivirus packaging and generation of RPE1 transgenic stable cell lines

Lentiviruses were generated by cotransfection of HEK293T cells with a transfer vector bearing GFP–CDK5RAP2 with the packaging vector psPAX2 and the envelope vector pMD2.G (psPAX2 and pMD2.G were gift from Didier Trono, Addgene plasmids 12,259 and 12260; RRID: Addgene_12259 and RRID: Addgene_12260) based on the PEI Max at a 3:1 ratio of PEI Max:DNA. In brief, PEI Max/DNA was mixed in fresh serum-free Gibco Opti-MEM medium, incubated for 15 min and added to the cell culture. The medium was changed with fresh complete medium after incubation overnight. Supernatants from packaging cells were collected 48–72 h after transfection, filtered through a 0.45 µm filter, incubated overnight at 4 °C with a polyethylene glycol (PEG)-6000-based precipitation solution, and then centrifuged at 1,500g for 30 min to concentrate the virus. The lentiviral pellet was resuspended in phosphate-buffered saline (PBS).

To generate the transgenic cell line of RPE1 AKAP450/CDK5RAP2/MMG triple knockout stably expressing GFP–CDK5RAP2, the RPE1 AKAP450/CDK5RAP2/MMG triple knockout cells were infected with the above-mentioned lentivirus and cultured in complete medium supplemented with 8 µg ml$^{-1}$ polybrene (Sigma-Aldrich). After 24 h, the medium was replaced with fresh complete medium. Starting 72 h after viral transduction, the RPE1 AKAP450/CDK5RAP2/MMG triple knockout cells were selected with puromycin at 10 µg ml$^{-1}$, for up to 3 days (until the majority of untransduced control cells treated with the same concentration of antibiotic died). After selection, cells were grown in complete medium for 3 days and were confirmed by immunofluorescence staining for GFP–CDK5RAP2 expression level and its colocalization with other centrosomal proteins.

### Purification of γ-TuRC from HEK293T GCP3– and GCP6–GFP–SII knockin cells

Human γ-TuCs used in the in vitro reconstitution assays were purified using SII and the Strep-Tactin affinity purification method[74]. Specifically, homozygous HEK293T GCP3– and GCP6–GFP–SII knockin cells, cultured in the dark, were collected from eight 15 cm dishes each and resuspended and lysed in lysis buffer (50 mM HEPES, 150 mM NaCl, 0.5% Triton-X-100, 1 mM MgCl$_2$, 1 mM ethylene glycol-bis(2-aminoethylether)-N,N,N′,N′-tetraacetic acid (EGTA), 0.1 mM GTP and 1 mM dithiothreitol (DTT), pH 7.4) supplemented with ethylenediaminetetraacetic acid (EDTA)-free protease inhibitor cocktail (Roche). Cell lysates were subjected to centrifugation at 21,000g for 20 min at 4 °C. The supernatants obtained from the previous step were incubated with equilibriated Strep-Tactin Sepharose beads (28-9355-99, GE Healthcare) for 45 min at 4 °C. Following incubation, the beads were washed three times with the wash buffer (50 mM HEPES, 150 mM NaCl and 0.1% Triton-X-100, 1 mM MgCl$_2$, 1 mM EGTA, 0.1 mM GTP and 1 mM DTT, pH 7.4) and γ-TuRC was eluted for 15 min at 4 °C in elution buffer (50 mM HEPES, 150 mM NaCl (300 mM NaCl for GCP6–GFP–SII), 0.05% Triton-X-100, 1 mM MgCl$_2$, 1 mM EGTA, 0.1 mM GTP, 1 mM DTT and 2.5 mM D-Desthiobiotin, pH 7.4). GCP6-tagged purified γ-TuRC was then subjected to buffer exchange using a Vivaspin 500 centrifugal concentrator (10 kDa molecular weight cut-off, Sartorius VS0102) for a final NaCl concentration of 150 mM in eluate. Triton-X-100 was omitted from all the buffers in γ-TuRC preparation used for MP experiments. Purified γ-TuCs were immediately aliquoted, snap frozen in liquid N$_2$ and stored at −80 °C. Throughout the purification process, tubes were covered with aluminium foil wherever possible. The purity and composition of purified γ-TuCs were analysed by western blot and MS.

### Sucrose density gradient centrifugation

First, 5.5 µl of γ-TuRC sample purified using GCP3–GFP–SII was diluted into 200 µl γ-TuRC buffer (50 mM HEPES, 150 mM NaCl, 0.05% Triton-X-100, 1 mM MgCl$_2$, 1 mM EGTA, 0.1 mM GTP and 1 mM DTT, pH 7.4). For standards, 1 mg each bovine serum albumin (BSA) (4.4 S)

and thyroglobulin (19.4 S) was dissolved in 200 µl γ-TuRC buffer. The samples were loaded onto separate 2 ml, 5–40% sucrose gradients and centrifuged in a TLS-55 rotor at 214,000g for 3 h at 4 °C with no brake. Then, 150 µl fractions were collected from the top to the bottom of sucrose gradients by cut-off pipette tips. Each fraction was mixed with 5× sodium dodecyl-sulfate loading buffer and boiled for 5 min at 95 °C and then 15–20 µl sample was loaded onto 4–202% Tris–glycine SDS–polyacrylamide gel electrophoresis (PAGE) gels for Coomassie blue staining or western blotting using an anti-γ-tubulin antibody (1:10,000, T6557, GTU-88, Sigma).

## Immunofluorescence and western blotting
HEK293T GCP3– and GCP6–GFP–SII knockin cells, seeded on coverslips, were fixed with prechilled methanol at −20 °C for 10 min followed by three washes with PBS and mounting on glass slides in Vectashield mounting medium containing DAPI (Vector laboratories).

The RPE1 cells were seeded onto coverslips in 24-well plates and incubated for 24 h. Cells, without or with stable overexpression of GFP–CDK5RAP2, were treated with 5 mM thymidine (cat. no. T9250, Sigma-Aldrich) overnight to block cell cycle. For immunofluorescence, cells were fixed with −20 °C methanol for 5 min, then rinsed in PBS for 5 min followed by permeabilization with 0.15% Triton-X-100 in PBS for 2 min, three washes with 0.05% Tween-20 in PBS for 5 min each, incubation for 20 min in the blocking buffer (2% BSA and 0.05% Tween-20 in PBS), 1 h incubation with primary antibodies (rabbit CAMSAP2 (1:300), cat. no. 17880-1-AP, Proteintech, RRID:AB_2068826; rat anti-α-tubulin, clone YL1/2 (1:600), cat. no. MA1-80017, Pierce, RRID:AB_2210201; homemade rat EB1 (ref. 75) KT51 (1:100), cat. no. ab53358, Abcam) diluted in the blocking buffer. Next, they were washed three times with 0.05% Tween-20 in PBS for 5 min each, followed by incubation for 1 h in secondary antibodies (goat anti-Rabbit, anti-Rat IgG Alexa Fluor −488, −594 and −647, (1:500) Molecular Probes (cat. nos. A-11034, A-11012, A-11006 and A-11007)) diluted in the blocking buffer, washed three times with 0.05% Tween-20 in PBS for 5 min each and air dried after a quick wash in 96% ethanol. The cells were mounted in Vectashield mounting medium without DAPI (Vector laboratories).

For western blotting, HEK293T cell lines grown in 6-well plates and RPE1 cell lines grown in 10 cm dishes were collected and lysed in the lysis buffer supplemented with complete protease inhibitor cocktail (Roche) (20 mM Tris-Cl pH 7.5, 100 mM NaCl, 1% Triton-X-100, 10% glycerol for GCP3– and GCP6–GFP–SII HEK293T cells or radioimmunoprecipitation assay buffer containing 50 mM Tris-Cl pH 7.5, 150 mM NaCl, 1% Triton-X-100, 0.5% sodium deoxycholate and phosphatase inhibitor (Roche) for RPE1 cells). Lysates were cleared by centrifugation at 21,000g for 20 min at 4 °C. Then, 20 µg of supernatant from the above step or 35 µg of purified γ-TuRC samples were loaded on 8% SDS–PAGE gels then transferred onto a nitrocellulose membrane (Sigma-Aldrich). Membranes were blocked in 2% BSA in 0.02% Tween-20 in PBS for 30 min at room temperature followed by overnight incubation with primary antibodies (rabbit polyclonal anti-GFP (1:4,000, ab290, Abcam); mouse anti-GCP3 (1:1,000, sc-373758, Santa Cruz); mouse monoclonal anti-GCP6 (1:500, sc-374063, Santa Cruz), mouse monoclonal anti-GCP5 (1:500, sc-365837, Santa Cruz), mouse monoclonal anti-GCP2 (1:500, sc-377117, Santa Cruz), rabbit polyclonal anti-GCP4 (1:1,000, PA5-30557, Thermo Fisher), mouse monoclonal anti-γ-tubulin (1:10,000, T6557, GTU-88, Sigma), rabbit polyclonal anti-CDK5RAP2 (1:500, A300-554A, Bethyl Laboratories) and mouse monoclonal anti-Ku80 (1:2,000, 611360, BD Biosciences)) at 4 °C, followed by three washes with 0.02% Tween-20 in PBS, 1 h incubation with secondary antibodies (1:15,000, goat anti-rabbit IRDye-800CW (cat. no. 926-32211) and goat anti-mouse IRDye-680LT (cat. no. 926-68020) from Li-Cor Biosciences) at room temperature and a final three washes. Membranes were imaged using Odyssey CLx infra-red imaging system 1.0.20 (Li-Cor Biosciences) controlled by Li-COR Image Studio software 5.2.5.

## Nocodazole washout assay
For the microtubule disassembly and regrowth assay, the RPE1 cells were treated with 10 µM nocodazole (cat. no. M1404, Sigma-Aldrich) for 1 h in an incubator (37 °C, 5% CO$_2$) and followed by another 1 h treatment at 4 °C to achieve complete disassembly of stable microtubule fragments. Nocodazole washout was then carried out by at least six washes with ice-cold complete medium on ice. For microtubule regrowth, 24-well plates were moved to a 37 °C water bath and pre-warmed medium was added to each well to allow microtubule regrowth for 1 min before the cells were fixed.

## Purification of recombinant proteins from HEK293T cells for in vitro reconstitution assays
Human mCherry–CDK5RAP2, mCherry–CLASP2, chTOG–mCherry and SNAP–AF647–CAMSAP1 and mouse SNAP–AF647–CAMSAP3 used in the in vitro reconstitution assays were purified using same SII and Strep-Tactin affinity purification method as described above for γ-TuRC, but with modified buffers and steps. In brief, HEK293T cells transfected with 50 µg of respective constructs per 15 cm dish were collected 36 h post-transfection from four 15 cm dishes each, and resuspended and lysed in lysis buffer (50 mM HEPES, 300 mM NaCl, 0.5% Triton-X-100, 1 mM MgCl$_2$ and 1 mM EGTA, pH 7.4) supplemented with EDTA-free protease inhibitor cocktail (Roche). Cell lysates were clarified and the supernatants obtained were incubated with equilibrated Strep-Tactin Sepharose beads. Following incubation of mCherry–CDK5RAP2, mCherry–CLASP2 and chTOG–mCherry preparations, beads were additionally washed five times using high salt (1 M NaCl)-containing wash buffer (50 mM HEPES, 0.1% Triton-X-100, 1 mM MgCl$_2$ and 1 mM EGTA, pH 7.4) before washing three times with 300 mM NaCl containing wash buffer. For SNAP-tag labelling of CAMSAP3 and CAMSAP1 with Alexa Fluor 647 dye, washed beads were incubated with labelling mix (50 µM Alexa Fluor 647 dye in 50 mM HEPES, 150 mM NaCl and 0.1% Triton-X-100, 1 mM MgCl$_2$, 1 mM EGTA and 1 mM DTT, pH 7.4) for 1 h. Following this incubation, beads were washed five times with wash buffer containing 300 mM NaCl to remove excess dye. Proteins were then eluted in elution buffer containing 50 mM HEPES, 150 mM NaCl, 0.05% Triton-X-100, 1 mM MgCl$_2$, 1 mM EGTA, 1 mM DTT and 2.5 mM D-Desthiobiotin, pH 7.4.

For purification of human mCherry–CAMSAP2, HEK293T cells were transfected with 25 µg of Bio-TEV–mCherry–CAMSAP2 and 25 µg of BirA per 15 cm dish. Cells were collected 36 h post-transfection from four 15 cm dishes and resuspended in lysis buffer (50 mM HEPES, 300 mM NaCl, 0.5% Triton-X-100, 1 mM MgCl$_2$, 1 mM EGTA and 1 mM DTT, pH 7.4) supplemented with EDTA-free protease inhibitor cocktail (Roche). The cell lysate was incubated with Dynabeads M-280 streptavidin (11206D, Invitrogen) for 1 h. The beads were washed three times with lysis buffer without protease inhibitors and three times with the TEV cleavage buffer (50 mM HEPES, 150 mM NaCl, 0.05% Triton-X-100, 1 mM MgCl$_2$, 1 mM EGTA and 1 mM DTT). mCherry–CAMSAP2 was eluted in 50 µl TEV cleavage buffer containing 0.5 µg of glutathione S-transferase-6×-histidine TEV protease site (Sigma-Aldrich) for 2 h at 4 °C.

Purified proteins were immediately aliquoted, snap frozen in liquid N$_2$ and stored at −80 °C. Bacterially expressed mCherry–EB3 (ref. 76) was a gift from Dr. M.O. Steinmetz (Paul Scherrer Institut). The purity of the samples was analysed by Coomassie staining of SDS–PAGE gels and MS.

## MS
Purified γ-TuRC preparations (GCP3- and GCP6-tagged) were resuspended in 20 µl of Laemmli sample buffer (Bio-Rad) and were loaded on a 4–12% gradient Criterion XT Bis-Tris precast gel (Bio-Rad). The gel was fixed with 40% methanol/10% acetic acid and then stained with colloidal Coomassie dye G-250 (GelCode Blue Stain Reagent, Thermo Scientific) for 1 h. Samples were resuspended in 10% formic acid (FA)/5%

dimethylsulfoxide post in-gel digestion, and analysed with an Agilent 1290 Infinity (Agilent Technologies) liquid chromatography (LC), operating in reverse-phase (C18) mode, coupled to an Orbitrap Q-Exactive mass spectrometer (Thermo Fisher Scientific). Peptides were loaded onto a trap column (Reprosil C18, 3 μm, 2 cm × 100 μm; Dr. Maisch) with solvent A (0.1% formic acid in water), at a maximum pressure of 800 bar and chromatographically separated over the analytical column (Zorbax SB-C18, 1.8 μm, 40 cm × 50 μm; Agilent) using a 90 min linear gradient from 7% to 30% solvent B (0.1% formic acid in acetonitrile) at a flow rate of 150 nl min⁻¹. The mass spectrometer automatically switched between MS and MS/MS in a data-dependent acquisition mode. The ten most abundant peptides were subjected to higher energy collisional dissociation fragmentation after a survey scan from 350 to 1,500 $m/z$. MS spectra in high-resolution mode ($R > 30,000$) were acquired, whereas MS2 was in high-sensitivity mode ($R > 15,000$). Raw data were processed using Proteome Discoverer 1.4 (version 1.4.0.288, Thermo Fisher Scientific) and a database search was performed using Mascot (version 2.4.1, Matrix Science) against a Swiss-Prot database (taxonomy human). Whereas oxidation of methionine was set as a variable modification, carbamidomethylation of cysteines was set as a fixed modification. Up to two missed cleavages were allowed by trypsin. Data filtering performed using percolator resulted in 1% false discovery rate (FDR). Additional filters set were search engine rank 1 and mascot ion score >20. For iBAQ analysis, raw files were analysed in MaxQuant (version 2.1.3.0) against the Swiss-Prot *Homo sapiens* protein database using the iBAQ algorithm. Modifications, such as methionine oxidation and protein N-terminal acetylation were set as variables, while carbamidomethylation of cysteines was set as a fixed modification. The proteinGroups.txt file was imported to Microsoft Excel for further analysis. Potential contaminants, reverse sequences and proteins identified by site were not filtered out.

To assess the quality of purified recombinant proteins (mCherry–CDK5RAP2, mCherry–CLASP2 and chTOG–mCherry), samples were digested using S-TRAP micro filters (ProtiFi) according to the manufacturer's protocol. In brief, 4 μg of protein samples were denatured using 5% SDS buffer and reduced and alkylated using DTT (20 mM, 10 min, 95 °C) and iodoacetamide (40 mM, 30 min), followed by acidification and precipitation using a methanol triethylammonium bicarbonate buffer before finally loading on a S-TRAP column. Trapped proteins were washed four times with methanol triethylammonium bicarbonate buffer followed by overnight trypsin (1 μg, Promega) digestion at 37 °C. Before LC–MS analysis, digested peptides were eluted and dried in a vacuum centrifuge.

Samples were analysed by reversed-phase nano-LC–MS/MS using an Ultimate 3000 ultrahigh-performance liquid chromatography coupled to an Orbitrap Q-Exactive HF-X mass spectrometer (Thermo Scientific). Digested peptides were separated over a 50 cm reversed-phase column, packed in house, (Agilent Poroshell EC-C18, 2.7 μm, 50 cm × 75 μm) using a linear gradient with buffer A (0.1% FA) and buffer B (80% acetonitrile, 0.1% FA) ranging from 13% to 44% B over 38 min at a flow rate of 300 nl min⁻¹, followed by a column wash and re-equilibration step. MS data were acquired using a data-dependent acquisition method with the set MS1 scan parameters in profile mode: 60,000 resolution, automatic gain control target equal to 3E6, a scan range of 375–1,600 $m/z$ and a maximum injection time of 20 ms. The MS2 scan parameters were set at 15,000 resolution, with an automatic gain control target set to standard, an automatic maximum injection time and an isolation window of 1.4 $m/z$. Scans were acquired using a fixed first mass of 120 $m/z$, a mass range of 200–2,000 and a normalized collision energy of 28. Precursor ions were selected for fragmentation using a 1 s scan cycle, a 10 s dynamic exclusion time and a precursor charge selection filter for ion possessing +2 to +6 charges. The total data acquisition time was 55 min.

Raw files were processed using Proteome Discoverer (version 2.4, Thermo Scientific). A database search was performed for MS/MS fragment spectra using Sequest-HT against a human database (UniProt, year 2020) that was modified to include protein sequences from our cloned constructs and a common contaminants database. A precursor mass tolerance of 20 ppm and a fragment mass tolerance of 0.06 Da was set for the search parameters. Up to two missed cleavages were allowed by trypsin digestion. Carbamidomethylation was set as fixed modification and methionine oxidation and protein N-terminal acetylation were set as variable modifications. Data filtering performed using percolator resulted in 1% FDR for peptide spectrum match and a 1% FDR was applied to peptide and protein assemblies. An additional filter with a minimum Sequest score of 2.0 was set for peptide spectrum match inclusion. MS1-based quantification was performed using the Precursor Ion Quantifier node with default settings and precursor ion feature matching was enabled using the Feature Mapper node. Common protein contaminants were filtered out from the results table.

## MP

MP experiments were performed using a Refeyn SamuxMP mass photometer (Refeyn) under standard field-of-view settings using in-house prepared microscope coverslips (24 mm × 50 mm; Paul Marienfeld GmbH). Contrast-to-mass calibration was performed using a mixture of thyroglobulin oligomers (T9145, Sigma-Aldrich). Before each sample, the instrument focus was first set using 12 μl of sample buffer (50 mM HEPES pH 7.4, 150 mM NaCl, 1 mM MgCl₂, 1 mM EGTA, 1 mM DTT, 100 μM GTP and 2.5 mM D-Desthiobiotin). Afterward, 3 μl of each sample was immediately transferred to the sample well and recorded for 120 s. The resulting MP data were processed using a combination of DiscoverMP (Refeyn) and in-house developed Python scripts. MP histograms were plotted using 15 kDa bin widths.

## Negative-stain EM and data processing

The purified γ-TuRC sample was thawed and incubated in the presence or absence of 120 nM CDK5RAP2 on ice for 20 min. Then, 2–3 μl of protein was applied to glow-discharged carbon-coated copper grids (Electron Microscopy Sciences; CF-400-Cu) and incubated for 45 s at room temperature. The protein solution was removed by manual blotting with a Whatman No. 1 filter paper. Next, 2–3 μl of protein solution was applied again to improve particle density. The protein solution was manually blotted from one side of the grid while freshly filtered 1% uranyl acetate (wt/vol) was simultaneously pipetted from the opposite side to exchange the solution. Grids were incubated in uranyl acetate for a further 45 s. The stain was removed by manual blotting and grids were air dried for >24–48 h in a sealed container containing desiccant before imaging. The grids were initially screened on a Thermo Fisher Scientific Tecnai F20 located at ETH Zurich's ScopeM facility, and final datasets on suitable grids were collected on an FEI Talos 120 at the University of Zurich's Center for Microscopy and Image Analysis.

For each condition, several thousand micrographs were recorded via the MAPS automated acquisition software at a magnification of 57,000× (2.4 Å per pixel) on a BM-Ceta complementary metal-oxide semiconductor camera. Contrast transfer function parameters were estimated using CTFFIND4 (ref. 77). All subsequent processing was done in RELION version 3.1 and University of California, San Francisco, Chimera and ChimeraX[78–80].

The main 3D reconstruction workflow steps are outlined in Extended Data Fig. 5i,j. Generally, micrographs were imported, and a small set of <1,000 particles was manually picked and subjected to reference-free two-dimensional (2D) classification. The resulting set of averages was used as an initial template for RELION's built-in auto-picking implementation. One or two rounds of auto-picking were performed to yield the best templates for optimal picking. Auto-picked particles were binned by four and subjected to reference-free 2D classification to remove particles probably corresponding to dirt and other contaminants. A random subset of the cleaned, binned particles was used to generate an ab initio model. Then, all cleaned, binned particles

were subjected to an intermediate 3D auto-refinement step using the ab initio model as a reference, re-extracted using the refined coordinates and subjected to 3D classification. Particles that generated 3D classes containing the highest level of detail were re-extracted from micrographs at either bin 2 pixel size (4.8 Å) and subjected to a final round of 3D auto-refinement using one of the classes as a new reference model.

A published model for γ-tubulin bound to GCP2 (PDB ID: 6V6S (ref. 20)) was individually docked into each radial 'spoke' density of the resulting EM density maps using UCSF Chimera's 'Fit in map' function. Only resulting fits for which (1) the number of atoms outside the contour was <20% of the total number of atoms fitted and (2) the resulting fit did not form domain clashes with neighbouring γ-tubulin/GCP2 subunits were kept for further analysis. This procedure led to 12 out of possible 14 γ-tubulin/GCP2 subcomplexes to be reliably fitted into both density maps (Extended Data Fig. 5b,c,e,f).

A model for 12 laterally associated β-tubulin subunits was constructed using PDB ID: 2HXF (ref. 50) and EMD-5193 (ref. 51). The ring was aligned to each γ-tubulin ring using the align command in Pymol; the sixth γ-tubulin subunit was used as an alignment anchor. The centre of mass of each β- or γ-tubulin subunit was then calculated based on its $C\alpha$ coordinates and the radial or axial displacement relative to the helical axis of the β-tubulin ring (that is, the microtubule) was determined using a custom MATLAB script. A similar analysis was performed using the γ-tubulin ring from the γ-TuSC oligomer in the 'closed' conformation (blue; PDB ID: 5FLZ (ref. 49)). The results are plotted in Extended Data Fig. 5h; values close to zero indicate a closer fit to the 13-protofilament microtubule.

### In vitro reconstitution assays

**In vitro reconstitution of microtubule nucleation.** The flow chamber was assembled using plasma-cleaned glass coverslip and microscopic slide attached together using double-sided tape. These chambers were then functionalized by 5 min incubation with 0.2 mg ml$^{-1}$ biotinylated poly(L-lysine)-[g]-poly(ethylene glycol) (PLL–PEG–biotin) (Susos AG) followed by 5 min incubation with 1 mg ml$^{-1}$ NeutrAvidin (Invitrogen) in MRB80 buffer. Next, biotinylated anti-GFP nanobody was attached to the coverslip through biotin–NeutrAvidin links by incubating the chamber for 5 min. During this incubation, γ-TuC–GFP was diluted ten times and either pre-incubated with NPFs or MRB80 buffer for 3 min and then immobilized on the GFP–nanobody-coated coverslips by incubating it with flow chamber by 3 min. Non-immobilized γ-TuC was washed away with MRB80 buffer and flow chambers were further incubated with 0.8 mg ml$^{-1}$ k-casein to prevent non-specific protein binding. The nucleation mix with or without proteins (MRB80 buffer supplemented with 17.5 μM or 25 μM porcine brain tubulin, 50 mM or 80 mM (for CAMSAP assays) KCl, 1 mM GTP, 0.5 mg ml$^{-1}$ k-casein, 0.1% methylcellulose and oxygen scavenger mix (50 mM glucose, 400 mg ml$^{-1}$ glucose-oxidase, 200 mg ml$^{-1}$ catalase and 4 mM DTT)) were added to the flow chambers after centrifugation in an ultracentrifuge (Beckman Airfuge) at 119,000*g* for 5 min. The concentrations of the proteins and composition of the nucleation mix are indicated in either the figures or figure legends. The flow chambers were then sealed with high-vacuum silicone grease (Dow Corning), and three consecutive 10 min time-lapse videos were acquired after 2 min incubation (time, *t* = 0) of the flow chambers with the nucleation reactions on a TIRF microscope stage at 30 °C. All tubulin products were from cytoskeleton.

**Single-molecule GFP counting assays for surface-adsorbed γ-TuC.** To estimate the number of GCP3–GFP molecules in γ-TuC, three parallel flow chambers were assembled on the same plasma-cleaned glass coverslip. The three chambers were incubated with the dilutions of GFP protein (monomeric), GFP–EB3 (dimeric) and GCP3–GFP (test) strongly diluted to single-molecules level. Flow chambers were then washed with MRB80 buffer, sealed with vacuum grease and immediately imaged using a TIRF microscope. Samples were focused first

in one area and 15–20 images of unexposed coverslip regions were acquired with 100 ms exposure time. Acquisition settings were kept constant for the three parallel chambers.

**Single-molecule GFP counting assays for nanobody-immobilized γ-TuC in CAMSAP3 assays.** Three parallel flow chambers were assembled on the same plasma-cleaned glass coverslip and microtubule nucleation assays were performed the same way as described above for CAMSAP assays. However, in two of the three chambers, monomeric control GFP protein and dimeric control GFP–EB3 were immobilized using the biotinylated anti-GFP nanobody, and in the third chamber GCP3–GFP was immobilized. The flow chambers were sealed with vacuum grease and immediately imaged using a TIRF microscope. Samples were focused first in one area and 21–24 images of unexposed coverslip regions (ROIs) were acquired for each chamber but first only for 488 nm illumination with 100 ms exposure time, and then for 642 nm, 561 nm and 488 nm illumination of the same ROIs using a sensitive Photometrics Evolve 512 electron multiplying charge-coupled device (EMCCD) camera (Roper Scientific). Acquisition settings were kept constant for the three parallel chambers. Further, time-lapse videos were acquired only from GCP3–GFP chamber for the same ROIs using the same camera at 20 s time interval with 100 ms exposure time each for 642 nm, 561 nm and 488 nm illumination for 10 min in three consecutive 10 min time lapses.

**Stabilized microtubule preparations.** Taxol- and docetaxel-stabilized and double-cycled GMPCPP-stabilized microtubules used for in vitro γ-TuC capping assays were prepared according to published procedures[52,81]. Specifically, GMPCPP-stabilized microtubules seeds were prepared in the presence of GMPCPP (Jena Biosciences) by two rounds of polymerization and a depolymerization cycle. First, a 20 μM porcine brain tubulin (cytoskeleton) mix composed of 70% porcine unlabelled tubulin, 18% biotin tubulin and 12% HiLyte647-tubulin or HiLyte488-tubulin was incubated with 1 mM GMPCPP in MRB80 buffer (pH 6.8, 80 mM K-PIPES, 1 mM EGTA and 4 mM MgCl$_2$) at 37 °C for 30 min. The polymerization mix was then pelleted by centrifugation in an Airfuge for 5 min at 119,000*g* followed by resuspension and depolymerization in MRB80 buffer on ice for 20 min and subsequent polymerization for 30 min in the presence of fresh 1 mM GMPCPP at 37 °C. GMPCPP-stabilized microtubule seeds were pelleted, resuspended in MRB80 buffer containing 10% glycerol, aliquoted, snap frozen in liquid N$_2$ and stored at −80 °C.

Taxol- and docetaxel-stabilized microtubules were prepared 24 h in advance by polymerizing a 29 μM porcine brain tubulin mix (86% porcine unlabelled tubulin, 10% biotin tubulin and 4% HiLyte647-tubulin) in the presence of 2.5 mM GTP (Sigma-Aldrich) and 20 μM Taxol (Sigma-Aldrich) or docetaxel (Sanofi-Aventis) in MRB80 buffer (pH 6.8, 80 mM K-PIPES, 1 mM EGTA and 4 mM MgCl$_2$) at 37 °C for 30 min. After polymerization, the GTP–tubulin–Taxol mix was diluted five times with pre-warmed 20 μM Taxol or docetaxel made in MRB80 buffer and centrifuged at 16,200*g* for 15 min at room temperature. The microtubule pellet was resuspended in pre-warmed 20 μM Taxol solution in MRB80 buffer and stored at room temperature in the dark covered with aluminium foil.

**In vitro reconstitution of microtubule dynamics.** Flow chambers were functionalized first with PLL–PEG–biotin followed by NeutrAvidin as described above for the in vitro microtubule nucleation assays and then incubated with GMPCPP-stabilized microtubules for 2 min for their attachment to the coverslip through biotin–neutravidin links. Flow chambers were then incubated with 0.8 mg ml$^{-1}$ k-casein to prevent non-specific protein binding. The tubulin polymerization reaction mix with or without 20 nM GFP–EB3 (MRB80 buffer supplemented with 25 nM mCherry–CDK5RAP2, 15 μM porcine brain tubulin, 50 mM KCl, 1 mM GTP, 0.5 mg ml$^{-1}$ k-casein, 0.1% methylcellulose and

oxygen scavenger mix (50 mM glucose, 400 mg ml⁻¹ glucose-oxidase, 200 mg ml⁻¹ catalase and 4 mM DTT)) was added to the flow chambers after centrifugation in an ultracentrifuge (Beckman Airfuge) at 119,000$g$ for 5 min. In the assays without GFP–EB3, 0.5 µM unlabelled porcine brain tubulin was substituted with 0.5 µM HiLyte488-labelled porcine brain tubulin to mark microtubules. The flow chambers were then sealed with high-vacuum silicone grease (Dow Corning) and three consecutive 10 min time-lapse videos were acquired immediately at 2 s time intervals with 100 ms exposure time on a TIRF microscope at 30 °C.

**Capping assays.** For capping assays, two parallel flow chambers on the same coverslip were functionalized as mentioned above and then incubated with GMPCPP- and Taxol-stabilized or GMPCPP- and docetaxel-stabilized microtubules for 2 min for their attachment to the coverslips through biotin–neutravidin links. Flow chambers were then incubated with 0.8 mg ml⁻¹ k-casein to prevent non-specific protein binding and a master mix of reaction mix (5 µM tubulin, 50 mM KCl, 1 mM GTP, 0.5 mg ml⁻¹ k-casein, 0.1% methylcellulose and oxygen scavenger mix (50 mM glucose, 400 mg ml⁻¹ glucose-oxidase, 200 mg ml⁻¹ catalase and 4 mM DTT) in MRB80 buffer centrifuged in an Airfuge for 5 min at 119,000$g$) supplemented with γ-TuC (GCP3–GFP) and with or without 30 nM mCherry–CDK5RAP2 was prepared and divided into two for adding it to the parallel chambers. The flow chambers were then sealed with high-vacuum silicone grease and three 5 min time-lapse videos were acquired from both the parallel chambers alternatively after 2 min incubation (time, $t = 0$) on a TIRF microscope stage at 30 °C. The time interval was kept 20 s.

## Image acquisition, processing and data analysis

**Imaging of fixed cells.** Fixed HEK293T cells and RPE1 cells were imaged on a Nikon Eclipse Ni upright wide-field fluorescence microscope equipped with a Nikon DS-Qi2 camera (Nikon), an Intensilight CHGFI pre-centred fibre illuminator (Nikon), ET-DAPI and ET-EGFP filters (Chroma), controlled by Nikon NIS Br software. Slides were imaged using a Plan Apo Lambda 60× numerical aperture (NA) 1.4 oil or Plan Apo Lambda 100× NA 1.45 oil objectives (Nikon).

For CAMSAP2 immunofluorescence, gated stimulated emission depletion (STED) imaging was performed with a Leica TCS SP8 STED 3X microscope using HC PL Apo 100×/1.4 oil STED white objective, white laser (633 nm) for excitation and 775 nm pulsed laser for depletion, driven by LAS X software. An internal Leica HyD hybrid detector with a time gate of $1 \leq tg \leq 8$ ns was used and depletion laser power was equal to 90% of maximum power. Images were acquired in 2D STED mode with vortex phase mask.

**Analysis of CAMSAP2 stretches, EB1 and tubulin intensity in fixed cells.** Quantification of the number of CAMSAP2 stretches and immunofluorescence signal intensity of tubulin, EB1 and CAMSAP2 was done in ImageJ. GFP–CDK5RAP2 cell lines in the wild-type and AKAP450 background were clonal[72], whereas GFP–CDK5RAP2 triple CDK5RAP2/myomegalin/AKAP450 knockout cells were a mixed cell population. For analysis, cells with similar GFP intensity, with clearly visible GFP signal at the centrosome (or the centrosome and the Golgi in the wild-type cells; Golgi signal of GFP–CDK5RAP2 is absent in AKAP450 knockouts) but without strong cytoplasmic signal were selected for the analysis in the rescue cell lines. To analyse the number of CAMSAP2 stretches in RPE1 cells, images were thresholded by setting the threshold at $T = 13$. The Particle Analysis plugin of ImageJ was used to quantify the number of CAMSAP2 stretches per cell (minimum particle size parameter was set to eight pixel units). The cell area was calculated by manually drawing the ROIs around each cell. Tubulin raw integrated density per cell was divided by cell area to obtain tubulin density per square micron. To analyse the intensity of CAMSAP2 concentrated at centrosomes after nocodazole washout in RPE1 cells, ROIs of 5 µm² were drawn to

select the areas occupied by CAMSAP2 signal and mean the intensity was obtained. The extent of microtubule nucleation at centrosomes in the microtubule regrowth assays were measured by calculating EB1 mean intensity. All protein intensities quantified per cell across different cell lines, and conditions from one experiment were normalized to the average of wild-type controls from that experiment to control for the variability across independent experiments.

**TIRF microscopy.** In vitro assays were imaged on an iLas2 TIRF microscope setup[74]. Specifically, the iLas2 system (Roper Scientific) is a dual-laser illuminator for azimuthal spinning TIRF illumination and powered with a custom modification for targeted photomanipulation. This system was installed on the Nikon Eclipse Ti-E inverted microscope with the perfect focus system. This microscope was equipped with Nikon Apo TIRF 100× 1.49 NA oil objective (Nikon), charge-coupled device (CCD) camera CoolSNAP MYO M-USB-14-AC (Roper Scientific), EMCCD Evolve mono FW DELTA 512 × 512 camera (Roper Scientific) with the intermediate lens 2.5× (Nikon C mount adaptor 2.5×), 150 mW 488 nm laser, 100 mW 561 nm laser and 49002 and 49008 Chroma filter sets and controlled with MetaMorph 7.10.2.240 software (Molecular Devices). The final magnification using the Evolve EMCCD camera was 0.064 µm per pixel and for the CoolSNAP Myo CCD camera it was 0.045 µm per pixel. Temperature was maintained at 30 °C to image the in vitro assays using a stage top incubator model INUBG2E-ZILCS (Tokai Hit). Time-lapse movies were acquired using a CoolSNAP Myo CCD camera (Roper Scientific) at either 3 s (for reconstitution of microtubule dynamics with GMPCPP-stabilized microtubule seeds), 5 s (for nucleation assays without CAMSAPs) or 5 s and 20 s (for assays with CAMSAPs) time intervals with 200 ms, 300 ms and 500 ms exposure time for 642 nm, 561 nm and 488 nm illumination, respectively, for 10 min. Time-lapse images acquired on the more sensitive Photometrics Evolve 512 EMCCD camera (Roper Scientific) were taken at 3 s time intervals with 100 ms exposure time for 10 min, unless specified otherwise.

**γ-TuC nucleation efficiency.** For each independent nucleation assay, γ-TuCs that nucleated microtubules or were anchored to an already nucleated microtubule within 10 min of a time-lapse movie were manually counted. Also, all the γ-TuC (GCP3–GFP or GCP6–GFP) particles were detected and counted from the first or second frame of that time-lapse movie using an open-source ImageJ plugin ComDet v.0.5.4 (ref. [82]) and the percentage of active ones out of the total was calculated and plotted for nucleation efficiency. From each independent assay, nucleation efficiency was calculated for three consecutive 10 min time-lapse movies from three different fields of view and plotted as 0–10, 10–20 and 20–30 min.

**Microtubule growth dynamics analysis.** Images and movies were processed using Fiji[83]. Kymographs from the in vitro reconstitution assays were generated using the ImageJ plugin KymoResliceWide v.0.4 (ref. [84]). Microtubule dynamics parameters viz. plus-end growth rate and catastrophe frequency were determined from kymographs using an optimized version of the custom made JAVA plugin for ImageJ[76,81]. The relative standard error for catastrophe frequency was calculated as described in[74].

**Single-molecule GFP counting of γ-TuC.** Single-molecule GFP counting analysis was done according to a published procedure[85]. Specifically, single-molecule fluorescence puncta were detected, measured and fitted with 2D Gaussian functions using custom written ImageJ plugin DoM_Utrecht v.1.1.6 (ref. [86]). Using single GFP distribution as a starting point, we were able to build expected intensity distributions of 2, 3, 4 and so on, GFP molecules by using the convolution of its probability density function. The intensity distributions of GFP $N$-mers, where $N$ corresponds to the oligomers with increasing number of GFP molecules, formed the 'basis' distributions. Then, we fitted GCP3–GFP

intensity distribution as a sum of these 'basis' distributions with different weights, serving as the fit parameters.

**Single-molecule intensity analysis of γ-TuC from CAMSAP3 assays.** All the γ-TuC (GCP3–GFP) particles immobilized on coverslip using biotinylated-GFP–nanobody within a field of view were detected from the first frame or second frame of a 10 min time-lapse movie using above-mentioned ImageJ plugin ComDet. Integrated intensity values for individual active γ-TuCs that nucleated microtubules and either colocalized or did not colocalize with CAMSAP3 or dissociated from microtubules after CAMSAP3 recruitment were manually extracted and normalized to the average integrated intensity of total active γ-TuCs that nucleated microtubules within that field of view.

**Colocalization frequency of γ-TuC and NPFs.** All the γ-TuC (GCP3–GFP) particles or control GFP particles immobilized on a coverslip within a field of view were detected from the first frame or second frame of 10 min time-lapse movies or single images (for assays without tubulin) using the ImageJ plugin ComDet and their colocalization percentage with the respective NPF was calculated using the colocalization function of this ComDet plugin.

**Analysis of γ-TuC–CAMSAPs colocalization and microtubule release frequency.** All the active γ-TuCs that nucleated or anchored a microtubule were counted manually. Each of these γ-TuC-anchored microtubule minus ends were monitored for any binding of CAMSAPs within 10 min of a time-lapse movie and were scored manually if they recruited CAMSAPs and then plotted. Out of these CAMSAP-colocalized γ-TuC-anchored minus ends, the minus ends that started growing during this 10 min duration were quantified as released microtubules. For quantification of the percentage of micrtotubules released from active γ-TuC, all the γ-TuC-anchored minus ends that started growing within 10 min were counted as released microtubules.

**γ-TuC capping efficiency.** For quantification of γ-TuC capping efficiency, stabilized microtubules were monitored for 5 min during a time-lapse movie to distinguish plus and minus ends from their growth behaviour, and if the end of a microtubule that did not display fast growth dynamics was stably bound to γ-TuC for at least 2 min, the microtubule was scored as being γ-TuC-capped at the minus end.

### Statistics and reproducibility

All statistical details of experiments including the definitions, exact values of number of measurements, precision measures and statistical tests performed are mentioned in the figures or figure legends, unless stated here. All the experiments were repeated at least three times except the gels, western blots and micrographs shown in Extended Data Figs. 1b–d, f, 2b, 3d,h, 4a and 5a,d, which were repeated at least two times and Extended Data Fig. 1h, which was performed once. Data processing and statistical analysis were done in Excel and GraphPad Prism 9 (GraphPad Software). Data distribution was assumed to be normal but this was not formally tested. No randomization was performed in our study as samples were not required to be allocated into experimental groups. Data collection and analysis were not performed blind to the conditions of the experiments. No data were excluded from the analyses. Significance was defined as *$P < 0.05$, **$P < 0.01$, ***$P < 0.001$ and ****$P < 0.0001$.

### Reporting summary

Further information on research design is available in the Nature Portfolio Reporting Summary linked to this article.

### Data availability

MS data that support the findings of this study have been deposited to the ProteomeXchange Consortium via the PRIDE partner repository[87] with the dataset identifier PXD048637. Previously published data that were re-analysed here are available, for γ-tubulin bound to GCP2 under accession code PDB ID: 6V6S (ref. 20), for β-tubulin subunits within a microtubule, PDB ID: 2HXF (ref. 50) and EMD-5193 (ref. 51) and for γ-TuSC, PDB ID: 5FLZ (ref. 49). All data that support the conclusions are either available in the manuscript itself or available from the authors on request. Source data are provided with this paper.

### Code availability

MATLAB script used for analysis of structural data is available from the authors on request. MATLAB script used for the GFP stoichiometry analysis is either available online at ref. 88 or from the corresponding authors on request. ImageJ macros used in this study are either available online at refs. 82,84,86 or from the corresponding authors on request.

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

### Acknowledgements

Negative-stain EM grid screening was performed with support of the ScopeM imaging center, ETH Zurich. Negative-stain EM data collection

was performed with support of the Center for Microscopy and Image Analysis, University of Zurich. The authors gratefully acknowledge L. Carlini for her help in developing the MATLAB analysis scripts used to compare γ-TuRC structural data. The authors acknowledge support from the Dutch Research Council-funded Netherlands Proteomics Centre through the National Roadmap for Large-scale Research Infrastructures programme X-Omics (project 184.034.019) for the MS and MP analysis. This work was supported by the European Research Council Synergy grant 609822, the ZonMW TOP 91216006 programme and the Spinoza Prize to A.A., Swiss National Science Foundation Project Grant (no. 310030_208120) to M.W., and grants from National Natural Science Foundation of China (31871356 and 32070705) and the Fundamental Research Funds for the Central Universities (2042022dx0003 and 2042023kf0212) to K.J.

## Author contributions

D.R. designed and performed protein purifications and in vitro reconstitution experiments, analysed data and wrote the paper; Y.S. and F.C. performed and analysed experiment with cells. S.H., J.L.M., Y.Z. and Y.X. generated and characterized essential reagents. K.S., R.S. and M.A. performed, analysed and supervised MS experiments. V.Y. performed and analysed and A.J.R.H. supervised MP experiments. E.A.K. facilitated and performed data analysis. M.W. generated and analysed transmission EM data. K.J. generated reagents, performed experiments, supervised the project and wrote the paper. A.A. coordinated the project and wrote the paper.

## Competing interests

The authors declare no competing financial and non-financial interests.

## Additional information

**data** is available for this paper at https://doi.org/10.1038/s41556-024-01366-2.

**Correspondence and requests for materials** should be addressed to Kai Jiang or Anna Akhmanova.

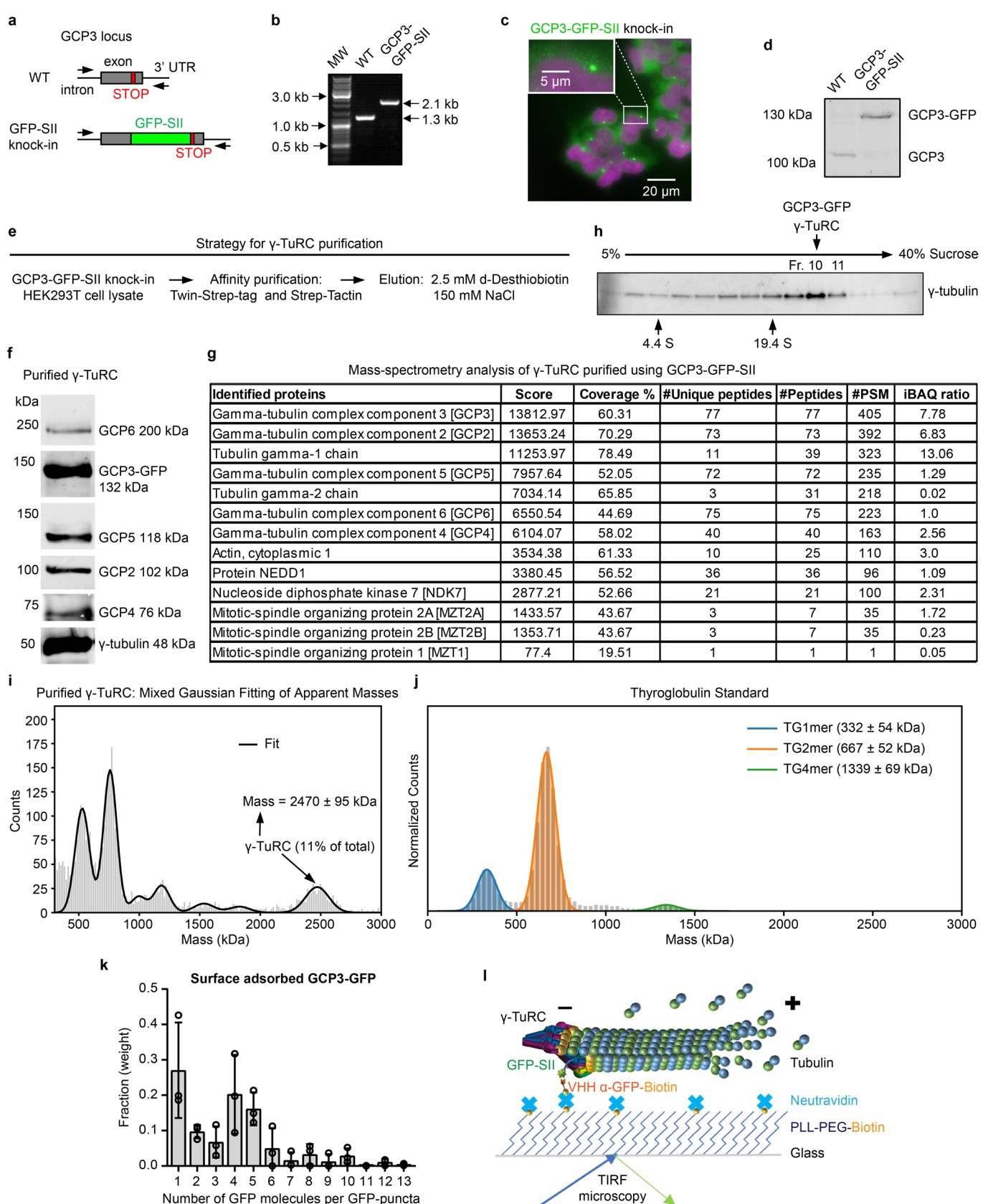

**Extended Data Fig. 1 | See next page for caption.**

**Extended Data Fig. 1 | Characterization of HEK293T GCP3-GFP-SII homozygous knock-in cell line and γ-TuRC purified using GCP3-GFP-SII.**
**a**, Scheme showing GCP3 gene locus and knock-in strategy. **b**, 1% Agarose gel showing genomic DNA PCR products for wild-type HEK293T cells and GCP3-GFP-SII knock-in cells. MW, molecular weight DNA ladder; WT, wild-type. **c**, Wide-field fluorescent image of fixed HEK293T GCP3-GFP-SII homozygous knock-in cells showing GFP fluorescence (green). Nuclei (magenta) were stained with DAPI. **d**, Western blot for wild-type and GCP3-GFP-SII knock-in HEK293T cell lysate, blotted using mouse anti-GCP3 antibody. **e**, Strategy for γ-TuRC purification. **f**, Western blot results showing the presence of all core components in the γ-TuRC sample purified using GCP3-GFP-SII using antibodies against GCP6, GFP, GCP5, GCP2, GCP4 and γ-tubulin. **g**, Mass spectrometry results showing the presence of all core components and their relative stoichiometry (iBAQ ratio) in γ-TuRC sample purified using GCP3-GFP-SII. iBAQ intensity ratio is relative to the iBAQ intensity of GCP6. **h**, Immunoblotting analysis of the γ-TuRC purified using GCP3-GFP-SII after sucrose density gradient centrifugation. Fractions were resolved by SDS–PAGE and blotted using γ-tubulin antibody. The preparation shows the presence of both complete and incomplete γ-TuRCs. **i,j**, Histogram and mixed Gaussian fitting of masses of all the molecular species detected in mass photometry of γ-TuC purified using GCP3-GFP-SII (**i**) and Thyroglobulin standard (**j**) showing the abundance of full γ-TuRC (11% of detected species; mass = 2470±95 kDa) and incomplete γ-TuRCs and other contaminants of lower molecular mass. The plots are representative of five measurements for γ-TuC (**i**) and a single measurement for Thyroglobulin standard (**j**). **k**, Quantification of GCP3-GFP stoichiometry of purified γ-TuCs adsorbed on coverslip. Averaged histogram of weights of N-mers of GFP determined from the fitting to the GCP3-GFP puncta intensities (as shown in Fig. 1a, **right**) showing the number of GFP molecules per immobilized GCP3-GFP puncta. The plot presents mean±s.d. for three independent experiments represented in Fig. 1a. **l**, Scheme showing experimental TIRF microscopy setup for in vitro reconstitution of microtubule nucleation from γ-TuRC.

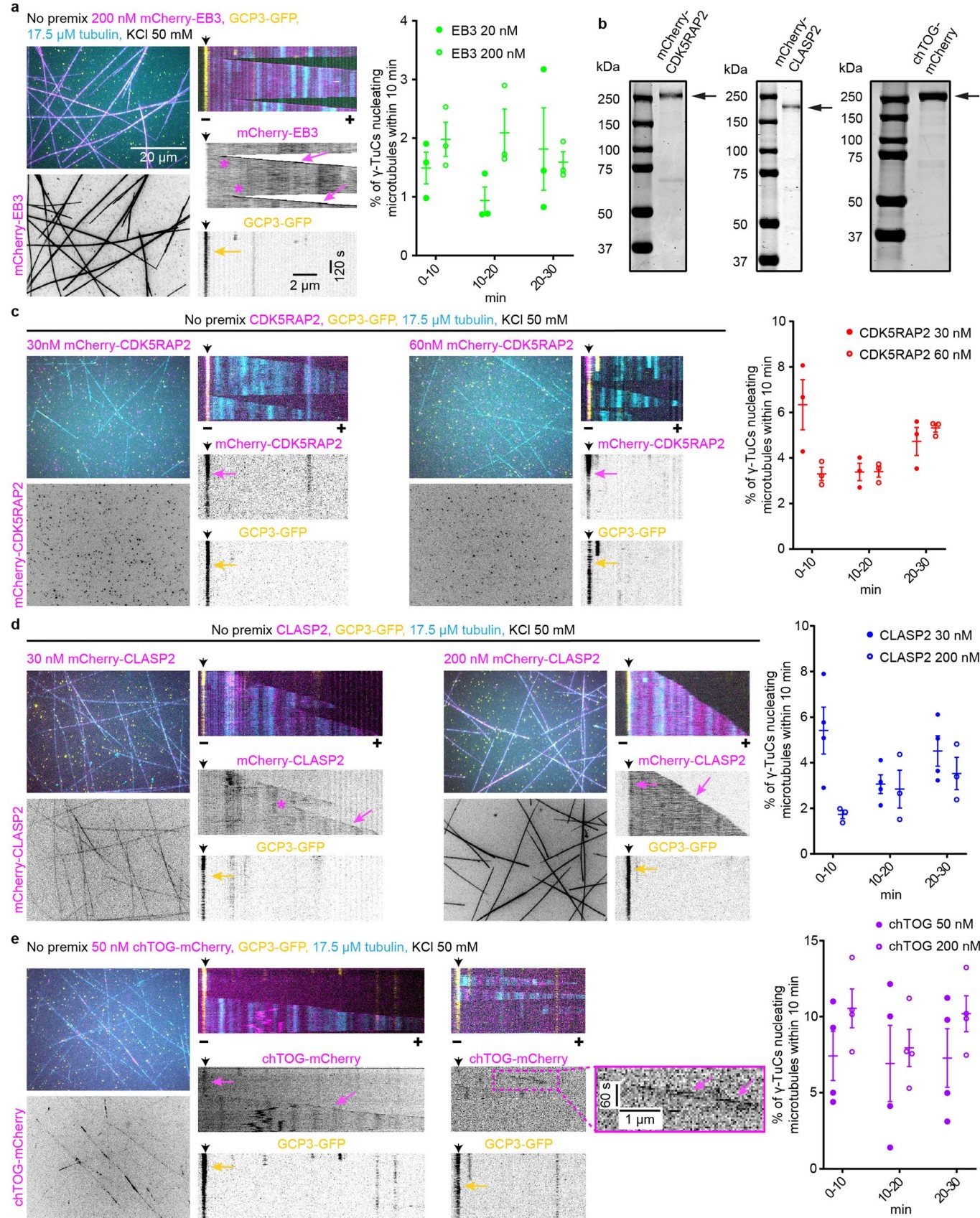

**Extended Data Fig. 2 | See next page for caption.**

**Extended Data Fig. 2 | Characterization of effective concentrations of nucleation-promoting factors. a,c,d,e**, Left: maximum intensity projections and representative kymographs illustrating microtubule dynamics in 10 min time-lapse videos, acquired after 20 min of incubation, showing microtubules (cyan) nucleated from γ-TuC (GCP3-GFP, yellow) in the presence of 17.5 μM tubulin (17 μM unlabeled porcine tubulin and 0.5 μM HiLyte647-tubulin), 50 mM KCl and together with indicated concentrations of indicated proteins (magenta) and without any preincubation: 200 nM mCherry-EB3 (**a**); or 30 nM and 60 nM mCherry-CDK5RAP2 (**c**); or 30 nM and 200 nM mCherry-CLASP2 (**d**); or 50 nM chTOG-mCherry (**e**). Minus and plus represent the two microtubule ends. Black arrowheads on top of kymographs indicate γ-TuC position. Magenta arrows point to the signal of proteins, while yellow arrows point towards γ-TuC. Asterisks show

rescues. Magnification for **c-e** is same as in **a**. Right: Quantification of average microtubule nucleation efficiency of γ-TuC as indicated: 200 nM mCherry-EB3 (n = 3); 30 nM (n = 3) or 60 nM (n = 3) mCherry-CDK5RAP2; 30 nM (n = 4) or 200 nM mCherry-CLASP2 (n = 3); 50 nM (n = 4) or 200 nM chTOG-mCherry (n = 4); where n is the number of independent experiments analyzed, also see Fig. 1c–h. The plots present mean±s.e.m., and each data point represents a single field of view for the given time points per experiment. Data points at 0–10 min were acquired from a smaller field of view, whereas the data points at 10–20 min and 20–30 min acquired from a larger field of view (shown in the panels on the left). Data points for concentrations already shown in Fig. 1e, h are from Fig. 1d, replotted here for comparison. **b**, Purified mCherry-CDK5RAP2, mCherry-CLASP2 or chTOG-mCherry analyzed by Coomassie-stained SDS-PAGE.

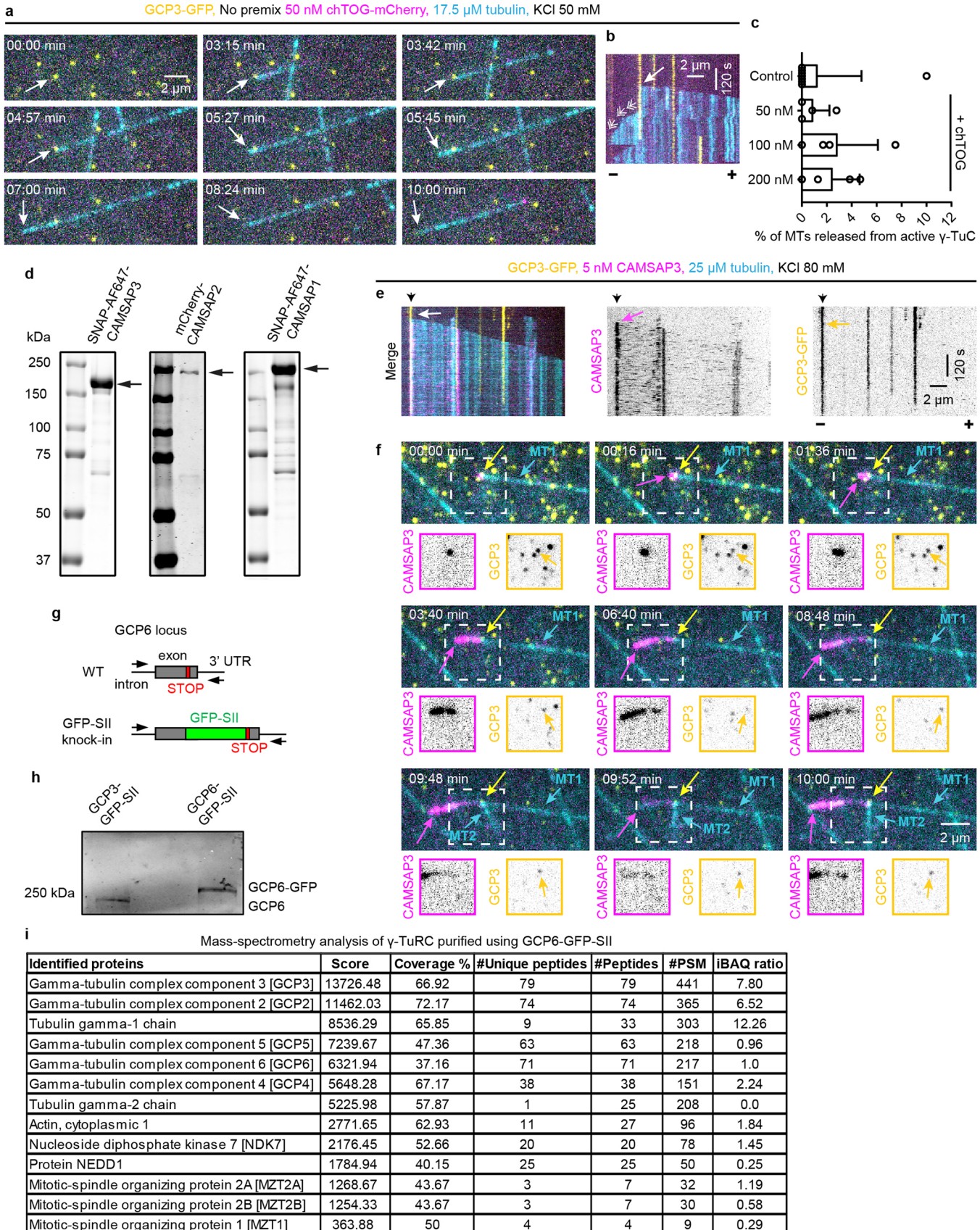

**i** Mass-spectrometry analysis of γ-TuRC purified using GCP6-GFP-SII

| Identified proteins | Score | Coverage % | #Unique peptides | #Peptides | #PSM | iBAQ ratio |
|---|---|---|---|---|---|---|
| Gamma-tubulin complex component 3 [GCP3] | 13726.48 | 66.92 | 79 | 79 | 441 | 7.80 |
| Gamma-tubulin complex component 2 [GCP2] | 11462.03 | 72.17 | 74 | 74 | 365 | 6.52 |
| Tubulin gamma-1 chain | 8536.29 | 65.85 | 9 | 33 | 303 | 12.26 |
| Gamma-tubulin complex component 5 [GCP5] | 7239.67 | 47.36 | 63 | 63 | 218 | 0.96 |
| Gamma-tubulin complex component 6 [GCP6] | 6321.94 | 37.16 | 71 | 71 | 217 | 1.0 |
| Gamma-tubulin complex component 4 [GCP4] | 5648.28 | 67.17 | 38 | 38 | 151 | 2.24 |
| Tubulin gamma-2 chain | 5225.98 | 57.87 | 1 | 25 | 208 | 0.0 |
| Actin, cytoplasmic 1 | 2771.65 | 62.93 | 11 | 27 | 96 | 1.84 |
| Nucleoside diphosphate kinase 7 [NDK7] | 2176.45 | 52.66 | 20 | 20 | 78 | 1.45 |
| Protein NEDD1 | 1784.94 | 40.15 | 25 | 25 | 50 | 0.25 |
| Mitotic-spindle organizing protein 2A [MZT2A] | 1268.67 | 43.67 | 3 | 7 | 32 | 1.19 |
| Mitotic-spindle organizing protein 2B [MZT2B] | 1254.33 | 43.67 | 3 | 7 | 30 | 0.58 |
| Mitotic-spindle organizing protein 1 [MZT1] | 363.88 | 50 | 4 | 4 | 9 | 0.29 |

**Extended Data Fig. 3 | See next page for caption.**

**Extended Data Fig. 3 | Characterization of purified CAMSAPs and GCP6-tagged γ-TuC and microtubule release from γ-TuC. a,b**, Still frames (at indicated time points in min, with 0:00 min being the starting point of the video) (**a**) and representative kymograph (**b**) from a 10 min time-lapse video showing microtubule (cyan) nucleation and subsequent microtubule release from γ-TuC (GCP3-GFP, yellow) in the indicated conditions. Thin arrows indicate microtubule minus end. Barbed arrowheads in the kymograph indicate the growth of minus end. **c**, Frequency of microtubule release from active γ-TuC over 10 min duration in the presence of either 17.5 μM tubulin alone (control, n = 52, N = 8); or together with 50 nM (n = 280, N = 4); 100 nM (n = 183, N = 4); or 200 nM mCherry-chTOG (n = 301, N = 4); where n is the number of active γ-TuCs analyzed from N independent experiments. Representative images are shown in **a. d**, Purified SNAP-AF647-CAMSAP3, mCherry-CAMSAP2 or SNAP-AF647-CAMSAP1 analyzed by Coomassie-stained SDS-PAGE. **e,f**, Two different examples of γ-TuC-CAMSAP3 interplay at the γ-TuC-anchored microtubule minus-ends under indicated experimental conditions, also shown in Fig. 3e, from a 10 min time-lapse video.

Example 1 (kymographs, **e**) illustrates occasions when CAMSAP3 fails to displace γ-TuC from microtubule minus-end. Black arrowheads on top of the kymographs indicate γ-TuC position. Example 2 (still frames at indicated time points, with 0:00 min being the starting point of the video, **f**) illustrates microtubule re-nucleation (cyan arrows, MT2) from the same γ-TuC that released previously nucleated microtubule (cyan arrows, MT1) upon CAMSAP3 binding and minus-end growth. At the bottom, individual channels for γ-TuC (right) and CAMSAP3 (left) are shown. Yellow arrows indicate γ-TuC, while magenta arrows indicate CAMSAP3. In **f**, magnification in individual channels is the same as merged images. **g**, Scheme showing GCP6 gene locus and knock-in strategy. **h**, Western blot for GCP3-GFP-SII and GCP6-GFP-SII knock-in HEK293T cell lysate, blotted using mouse anti-GCP6 antibody. **i**, Mass spectrometry results showing the presence of all the core components and their relative stoichiometry (iBAQ ratio) in γ-TuC purified using GCP6-GFP-SII. iBAQ intensity ratio is relative to the iBAQ intensity of GCP6. Minus and plus represent the two microtubule ends.

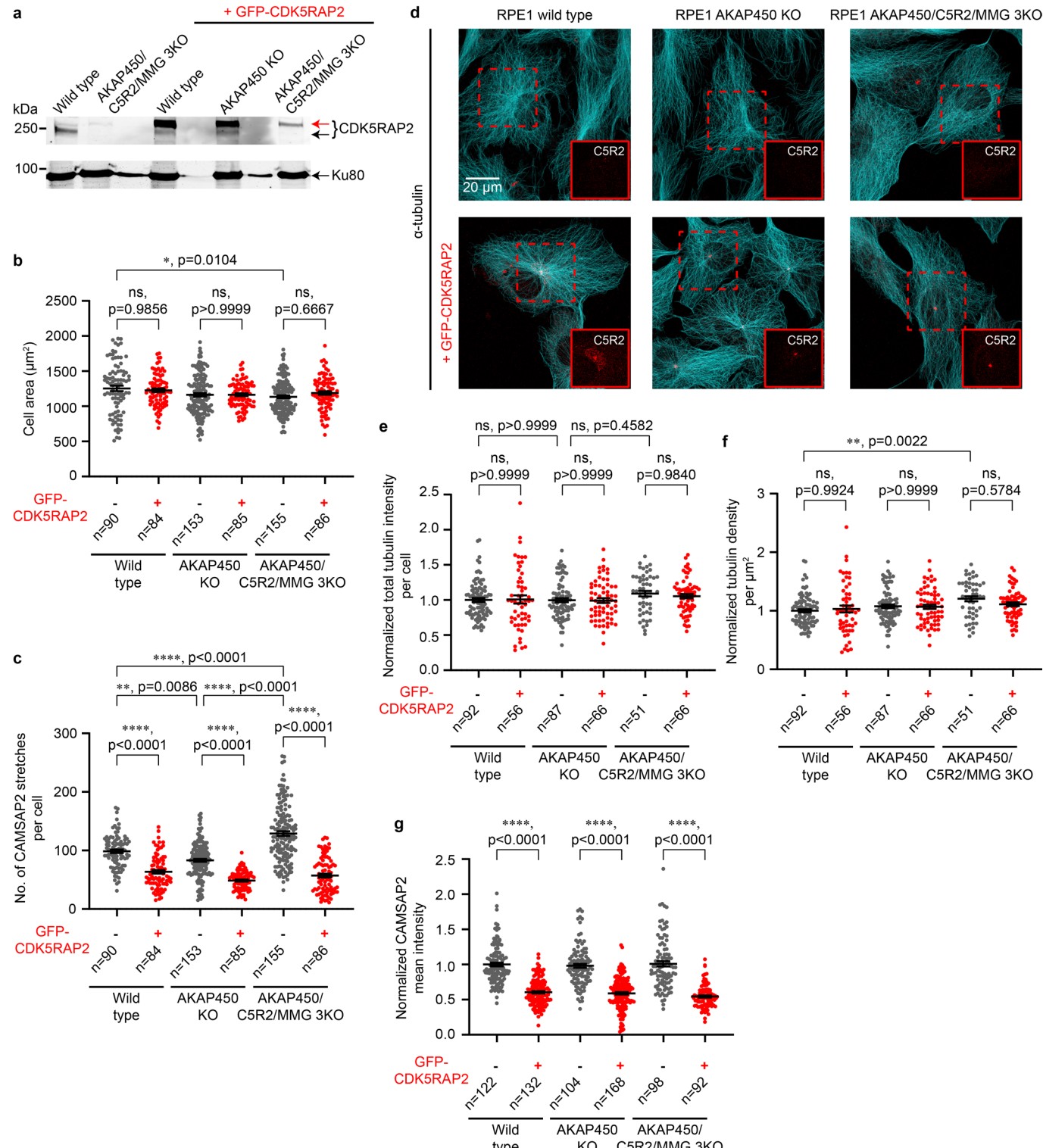

**Extended Data Fig. 4 | See next page for caption.**

**Extended Data Fig. 4 | Effects of CDK5RAP2 overexpression on the abundance of microtubules and CAMSAP2-bound microtubule minus ends in cells.**
**a**, Western blot of cell lysates from wild type, AKAP450/CDK5RAP2/MMG knockouts, and wild type, AKAP450 knockouts and AKAP450/CDK5RAP2/ MMG knockouts stably expressing GFP-CDK5RAP2, blotted using rabbit anti-CDK5RAP2 antibody (top) and mouse anti-Ku80 antibody (bottom). Top: red arrow indicates overexpressed GFP-CDK5RAP2 and black arrow indicates endogenous CDK5RAP2. GFP-CDK5RAP2 cell lines in the wild-type and AKAP450 background were clonal[71], whereas GFP-CDK5RAP2 triple CDK5RAP2/ Myomegalin/AKAP450 knockout cells were a mixed cell population with respect to the GFP-CDK5RAP2 transgene. In the clonal lines, GFP-CDK5RAP2 overexpression was estimated to be 6–8-fold to the respective endogenous levels. See also the Methods section. **b, c**, Area of cells in square microns (mean±s.e.m.) (**b**) and number of CAMSAP2 stretches per cell (mean±s.e.m.) (**c**) quantified from experiments represented in Fig. 7a. Number of cells analyzed, n, from three independent experiments in all conditions, are indicated. **d**, Representative immunofluorescence images of indicated cell lines with or without stable expression of GFP-CDK5RAP2 (red), stained for α-tubulin (cyan). Insets show cropped CDK5RAP2 channel in red, magnification is same as merged images. **e,f**, Total tubulin intensity per cell (mean±s.e.m.) normalized to wild-type average (**e**) and tubulin density per square micron area (mean±s.e.m.) normalized to wild-type average (**f**) quantified from experiments represented in **d** and Fig. 7a and using values in panels **b** and **e**. Number of cells analyzed, n, from three independent experiments in all conditions, are indicated. **g**, CAMSAP2 mean intensity per cell (mean±s.e.m.) normalized to wild-type average quantified from experiments represented in Fig. 7e. Number of cells analyzed, n, from three independent experiments in all conditions, are indicated. One-way ANOVA test with Tukey's multiple comparisons corrected for multiple testing in **b,c,e-g**. ns, not significant. CDK5RAP2 is abbreviated as C5R2 in figure panels.

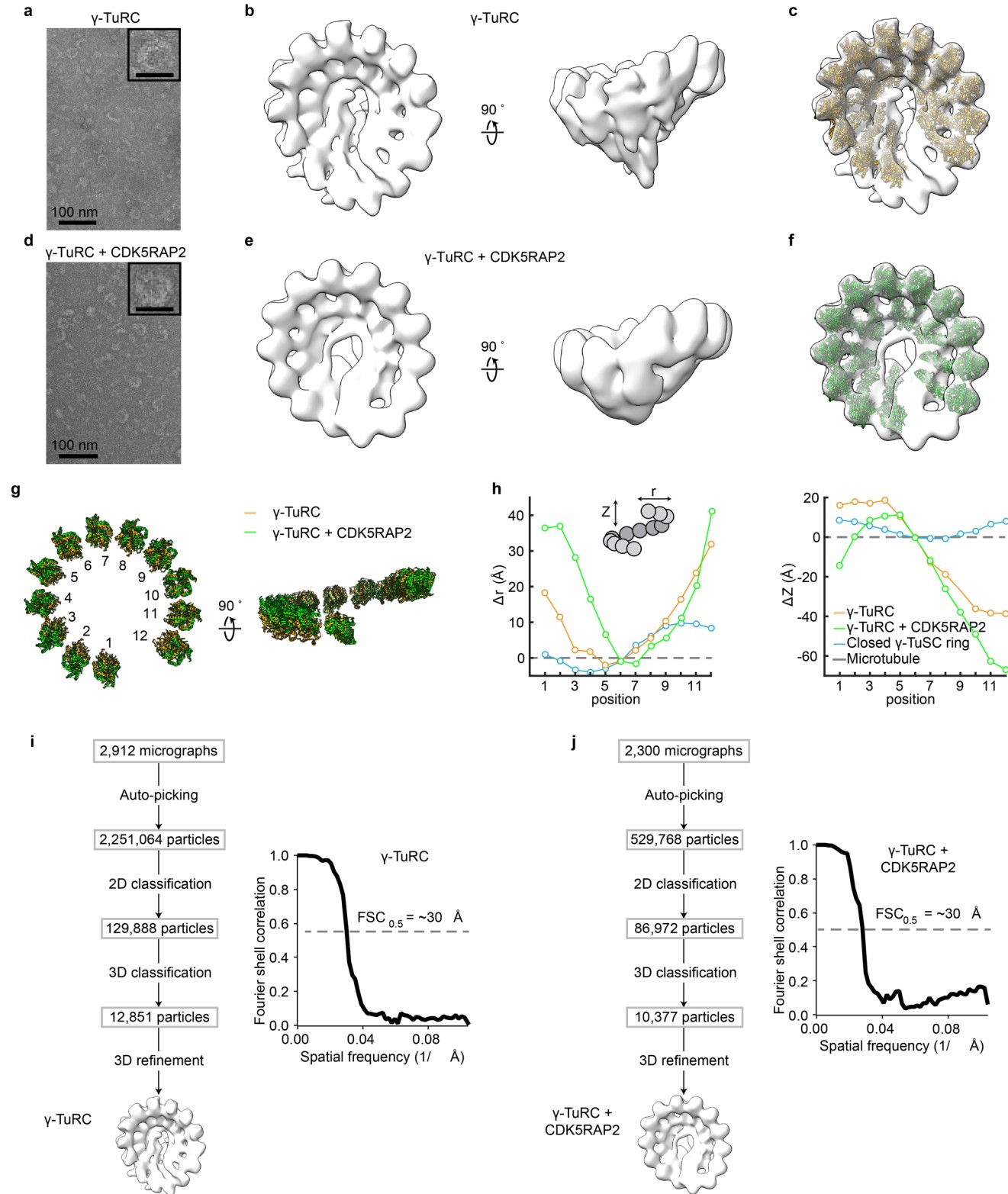

**Extended Data Fig. 5 | See next page for caption.**

**Extended Data Fig. 5 | Characterization of γ-TuRC by EM in the absence or presence of CDK5RAP2. a**, Transmission EM (TEM) micrograph of negatively stained γ-TuRC. Inset shows 4X magnified view of a single γ-TuRC, scale bar-25nm. **b**, Two views of a 3D reconstruction of the γ-TuRC from negative-stain EM data. **c**, Rigid body fit of repeating γ-tubulin/GCP2 subcomplexes (from PDB ID: 6V6S ref. 20) individually docked into the γ-TuRC density map. Fits for two subcomplexes at the γ-TuRC 'seam' were not reliable; and are therefore, omitted for clarity. **d**, Transmission EM (TEM) micrograph of negatively stained γ-TuRC prepared in complex with 120 nM CDK5RAP2. Inset shows 4X magnified view of a single γ-TuRC, scale bar-25nm. **e**, Two views of a 3D reconstruction of the γ-TuRC + CDK5RAP2 preparation from negative-stain EM data. **f**, Rigid body fit of repeating γ-tubulin/GCP2 subcomplexes (from PDB ID: 6V6S ref. 20) individually docked into the γ-TuRC + CDK5RAP2 density map. As in **c**, fits for two subcomplexes at the γ-TuRC 'seam' were not reliable; and are therefore, omitted for clarity.

**g**, Two views of the γ-tubulin rings from rigid body fitted models in **c** (γ-TuRC) and **f** (γ-TuRC + CDK5RAP2). **h**, Plots of the change in helical radius (*r*) and helical pitch (*Z*) relative to *β*-tubulin in the 13-protofilament microtubule lattice (grey dashed line; PDB ID: 2HXF ref. 49 and EMD-5193 ref. 50) calculated for γ-tubulin rings from γ-TuRC alone (orange), γ-TuRC + 120 nM CDK5RAP2 (green), and the γ-TuSC oligomer in the 'closed' state (blue; PDB ID: 5FLZ ref. 48). See Methods for analysis details. **i**, Left: Processing workflow for generating a negative-stain EM 3D reconstruction of γ-TuRC. Right: Unmasked FSC curve for the γ-TuRC reconstruction. FSC = 0.5 is indicated by a dashed gray line, and an estimate of the corresponding resolution is indicated. **j**, Left: Processing workflow for generating a negative-stain EM 3D reconstruction of γ-TuRC in the presence of 120 nM CDK5RAP2. Right: Unmasked FSC curve for the γ-TuRC + CDK5RAP2 reconstruction. FSC = 0.5 is indicated by a dashed gray line, and an estimate of the corresponding resolution is indicated.

# Reporting Summary

## Statistics

For all statistical analyses, confirm that the following items are present in the figure legend, table legend, main text, or Methods section.

| n/a | Confirmed | |
|---|---|---|
| ☐ | ☒ | The exact sample size (*n*) for each experimental group/condition, given as a discrete number and unit of measurement |
| ☐ | ☒ | A statement on whether measurements were taken from distinct samples or whether the same sample was measured repeatedly |
| ☐ | ☒ | The statistical test(s) used AND whether they are one- or two-sided<br>*Only common tests should be described solely by name; describe more complex techniques in the Methods section.* |
| ☒ | ☐ | A description of all covariates tested |
| ☐ | ☒ | A description of any assumptions or corrections, such as tests of normality and adjustment for multiple comparisons |
| ☐ | ☒ | A full description of the statistical parameters including central tendency (e.g. means) or other basic estimates (e.g. regression coefficient) AND variation (e.g. standard deviation) or associated estimates of uncertainty (e.g. confidence intervals) |
| ☐ | ☒ | For null hypothesis testing, the test statistic (e.g. *F*, *t*, *r*) with confidence intervals, effect sizes, degrees of freedom and *P* value noted<br>*Give P values as exact values whenever suitable.* |
| ☒ | ☐ | For Bayesian analysis, information on the choice of priors and Markov chain Monte Carlo settings |
| ☒ | ☐ | For hierarchical and complex designs, identification of the appropriate level for tests and full reporting of outcomes |
| ☒ | ☐ | Estimates of effect sizes (e.g. Cohen's *d*, Pearson's *r*), indicating how they were calculated |

*Our web collection on statistics for biologists contains articles on many of the points above.*

## Software and code

Policy information about availability of computer code

| Data collection | MetaMorph 7.10.2.240 software, Nikon NIS Br software, MAPS automated acquisition software, LI-COR Image Studio software 5.2.5, Refeyn SamuxMP software, Leica LAS X software. Software details are also provided in the methods section. |
|---|---|
| Data analysis | ImageJ 1.45s, ImageJ 1.53c (Fiji), ImageJ plugin KymoResliceWide v.0.4 (https://github.com/ekatrukha/KymoResliceWide), custom made JAVA plugin, ImageJ plugin ComDet v.0.5.4 (https://github.com/ekatrukha/ComDet), ImageJ plugin DoM_Utrecht v.1.1.6 (https://github.com/ekatrukha/DoM_Utrecht), custom MATLAB script for GFP stoichiometry analysis (https://doi.org/10.6084/m9.figshare.23943150.v4), Proteome Discoverer version 1.4.0.288 and 2.4, Mascot version 2.4.1, Sequest HT, MaxQuant version 2.1.3.0, Refeyn DiscoverMP, Python scripts, CTFFIND4, RELION version 3.1, UCSF Chimera and ChimeraX, PyMOL, custom MATLAB script, Microsoft Excel for Microsoft 365, GraphPad Prism 9. |

For manuscripts utilizing custom algorithms or software that are central to the research but not yet described in published literature, software must be made available to editors and reviewers. We strongly encourage code deposition in a community repository (e.g. GitHub). See the Nature Portfolio guidelines for submitting code & software for further information.

## Data

Policy information about availability of data

All manuscripts must include a data availability statement. This statement should provide the following information, where applicable:
- Accession codes, unique identifiers, or web links for publicly available datasets
- A description of any restrictions on data availability
- For clinical datasets or third party data, please ensure that the statement adheres to our policy

> Previously published data that were re-analysed here are available, for γ-tubulin bound to GCP2 under accession code PDB ID: 6V6S, for β-tubulin subunits within a microtubule, PDB ID: 2HXF and EMD-5193, for γ-TuSC, PDB ID: 5FLZ. Custom MATLAB script developed for GFP stoichiometry analysis is available on https://doi.org/10.6084/m9.figshare.23943150.v4. Mass spectrometry data have been deposited to the ProteomeXchange Consortium via the PRIDE partner repository with the dataset identifier PXD048637. All data that support the conclusions are either available in the manuscript itself or available from the authors on request.

## Human research participants

Policy information about studies involving human research participants and Sex and Gender in Research.

| | |
|---|---|
| Reporting on sex and gender | N/A |
| Population characteristics | N/A |
| Recruitment | N/A |
| Ethics oversight | N/A |

Note that full information on the approval of the study protocol must also be provided in the manuscript.

# Field-specific reporting

Please select the one below that is the best fit for your research. If you are not sure, read the appropriate sections before making your selection.

☒ Life sciences ☐ Behavioural & social sciences ☐ Ecological, evolutionary & environmental sciences

For a reference copy of the document with all sections, see nature.com/documents/nr-reporting-summary-flat.pdf

# Life sciences study design

All studies must disclose on these points even when the disclosure is negative.

| | |
|---|---|
| Sample size | No statistical methods were used to predict the sample size. All datasets were pooled from at least three or more independent experiments. Sample size was chosen based on the reproducibility, our previous experience or the standards in the field. Data distribution was assumed to be normal but this was not formally tested. In each case, sample size, number of independent experiments and statistical tests, when used, along with p values were indicated in the figure panels, legends or in the statistical and reproducibility subsection within the methods section. |
| Data exclusions | No data were excluded from the analyses. |
| Replication | Each experimental condition was repeated at least three times or more unless stated otherwise. All attempts at replication were successful. |
| Randomization | No randomization was performed in our study as samples were not required to be allocated into experimental groups. |
| Blinding | Investigators were not blinded to group allocation during data collection and analyses as group allocation was not required. Data was collected immediately after each experiment was performed for each experimental condition with different proteins or different protein concentrations, and therefore blinding was not possible. The analyses were done using automated methods or using strictly defined parameters as mentioned in the methods section, therefore preventing any subjective error. |

# Reporting for specific materials, systems and methods

We require information from authors about some types of materials, experimental systems and methods used in many studies. Here, indicate whether each material, system or method listed is relevant to your study. If you are not sure if a list item applies to your research, read the appropriate section before selecting a response.

## Materials & experimental systems

| n/a | Involved in the study |
|---|---|
| ☐ | ☒ Antibodies |
| ☐ | ☒ Eukaryotic cell lines |
| ☒ | ☐ Palaeontology and archaeology |
| ☒ | ☐ Animals and other organisms |
| ☒ | ☐ Clinical data |
| ☒ | ☐ Dual use research of concern |

## Methods

| n/a | Involved in the study |
|---|---|
| ☒ | ☐ ChIP-seq |
| ☒ | ☐ Flow cytometry |
| ☒ | ☐ MRI-based neuroimaging |

## Antibodies

| | |
|---|---|
| Antibodies used | Commercial antibodies used in this study are listed here:<br>rabbit anti-CAMSAP2, Proteintech, Cat#17880-1-AP, RRID:AB_2068826; rat anti-α-tubulin, clone YL1/2, Pierce, Cat#MA1-80017, RRID:AB_2210201; goat anti-Rabbit and anti-Rat IgG Alexa Fluor -488, -594, -647, Molecular Probes, Cat#A-11034, Cat#A-11012, Cat#A-11006, Cat#A-11007; rabbit polyclonal anti-GFP, Abcam, ab290; mouse anti-GCP3, Santa Cruz, sc-373758; mouse monoclonal anti-GCP6, Santa Cruz, sc-374063; mouse monoclonal anti-GCP5, Santa Cruz, sc-365837; mouse monoclonal anti-GCP2, Santa Cruz, sc-377117; rabbit polyclonal anti-GCP4, ThermoFisher, PA5-30557; mouse monoclonal anti-γ-tubulin, Sigma, Cat#T6557, GTU-88; rabbit polyclonal anti-CDK5RAP2, Bethyl Laboratories, A300-554A; mouse monoclonal anti-Ku80, BD Biosciences, 611360; goat anti-rabbit IRDye-800CW, Cat#926-32211 and goat anti-mouse IRDye-680LT, Cat#926-68020 from Li-Cor Biosciences, Lincoln, LE. Detailed information on commercial antibodies can also be found in the methods section. The biotinylated anti-GFP nanobody and rat monoclonal anti-EB1 antibody were home-made. The same rat monoclonal anti-EB1 antibody (KT51) is also available through Abcam (#ab53358). |
| Validation | Commercial antibodies were validated in species mentioned above for immunofluorescence and western blots as noted on manufacturer's website.<br>Home-made biotinylated anti-GFP nanobody:<br>Katrukha, E. A., Mikhaylova, M., van Brakel, H. X., van Bergen En Henegouwen, P. M., Akhmanova, A., Hoogenraad, C. C., & Kapitein, L. C. (2017). Probing cytoskeletal modulation of passive and active intracellular dynamics using nanobody-functionalized quantum dots. Nature communications, 8, 14772. https://doi.org/10.1038/ncomms14772<br>Home-made anti-EB1 antibody:<br>Komarova, Y., Lansbergen, G., Galjart, N., Grosveld, F., Borisy, G.G. & Akhmanova, A. EB1 and EB3 control CLIP dissociation from the ends of growing microtubules. Mol Biol Cell 16, 5334-5345 (2005). https://doi.org/10.1091/mbc.e05-07-0614 |

## Eukaryotic cell lines

Policy information about cell lines and Sex and Gender in Research

| | |
|---|---|
| Cell line source(s) | Human embryonic kidney 239T (HEK293T) cells (Cat#CRL-3216) were obtained from ATCC and used for generating knock-in cell lines.<br>hTERT RPE-1 (RPE1) wild type cells (Cat#CRL-4000) are available on ATCC.<br>RPE1 AKAP450 knockout and RPE1 AKAP450/CDK5RAP2/MMG triple knockout cell lines were previously published:<br>Wu, J., de Heus, C., Liu, Q., Bouchet, B. P., Noordstra, I., Jiang, K., Hua, S., Martin, M., Yang, C., Grigoriev, I., Katrukha, E. A., Altelaar, A. F. M., Hoogenraad, C. C., Qi, R. Z., Klumperman, J., & Akhmanova, A. (2016). Molecular Pathway of Microtubule Organization at the Golgi Apparatus. Developmental cell, 39(1), 44–60. https://doi.org/10.1016/j.devcel.2016.08.009<br>RPE1 AKAP450/CDK5RAP2/MMG triple knockout cell line mentioned above was used for generating transgenic cell line stably expressing GFP-CDK5RAP2.<br>RPE1 wild type and RPE1 AKAP450 knockout transgenic cell lines stably expressing GFP-CDK5RAP2 were previously published:<br>Chen, F., Wu, J., Iwanski, M. K., Jurriens, D., Sandron, A., Pasolli, M., Puma, G., Kromhout, J. Z., Yang, C., Nijenhuis, W., Kapitein, L. C., Berger, F., & Akhmanova, A. (2022). Self-assembly of pericentriolar material in interphase cells lacking centrioles. eLife, 11, e77892. https://doi.org/10.7554/eLife.77892 |
| Authentication | ATCC performs short-tandem repeat profiling for cell line authentication, and no additional cell line authentication was performed. |
| Mycoplasma contamination | The cell lines were routinely checked for mycoplasma contamination using LT07-518 Mycoalert assay and has been verified as mycoplasma free. |
| Commonly misidentified lines<br>(See ICLAC register) | The cell lines used are not present in the list of commonly misidentified lines. |

