## [Peer Review File · Nature Cell Biology]

Peer Review Information

Journal: Nature Cell Biology

Manuscript Title: CAMSAPs and nucleation-promoting factors control microtubule release from γ -TuRC

Corresponding author name(s): Professor Anna Akhmanova

Editorial Notes:

comments, excluding minor textual revisions, have been copied into this Peer Review File.

Reviewer Comments & Decisions:

Decision Letter, initial version:

Dear Professor Akhmanova,

I once again apologize for the delay. Your manuscript "CAMSAp-driven microtubule release from γ -TuRC and its regulation by nucleation-promoting factors", has now been seen by 3 referees, who are experts in γ -TuRC and MT dynamics (referee 1); structural biology and microtubule dynamics (referee 2); and microtubule nucleation (referee 3), and whose comments are pasted below. In light of their advice, we regret that we cannot offer to publish the study in Nature Cell Biology.

As you will see, although the reviewers find this work interesting, they raise a number of concerns that question the strength of the data and of the conclusions that can be drawn, including unclear action of the asymmetric γ -TuRC complex on microtubule nucleation in cells (see for example points from Reviewers #1 and #2), and in light of the points they raise, we find the present data-set too preliminary to pursue at this stage.

We are very sorry that we could not be more positive on this occasion, but we thank you for the opportunity to consider this work.

With kind regards,
Daryl Jason David

Daryl J.V. David, PhD

Senior Editor, Nature Cell Biology
Consulting Editor, Nature Communications
Nature Portfolio

Heidelberger Platz 3, 14197 Berlin, Germany
Email: daryl.david@nature.com
ORCID: <https://orcid.org/0000-0002-9253-4805>

Reviewers' comments:

Reviewer #1 (Remarks to the Author):

In this study the authors investigate how microtubule nucleation and minus end capping by the

nucleator gamma-TuRC are regulated. The authors use a TIRF-based assay that measures in vitro microtubule nucleation from purified and immobilized gamma-TuRC to test the effect of various, in part previously reported gamma-TuRC interactors including chTOG, CLASP2, and CDK5RAP2. Using this system, the authors show that all of these can stimulate gamma-TuRC nucleation activity in vitro. The authors then test the effect of CAMSAP proteins and find that they bind to the minus-end of a newly nucleated microtubule bound to gamma-TuRC and that CAMSAP2 and CAMSAP3, but not CAMSAP1, trigger minus end elongation and microtubule release from gamma-TuRC. Further, the authors find that pre-incubation of gamma-TuRC with the activator CDK5RAP2, abolishes CAMSAP3-dependent microtubules release from gamma-TuRC. Finally, the authors provide evidence that CDK5RAP2-bound gamma-TuRC preferentially binds to 12/13-protofilament over 14-protofilament microtubule ends, which they interpret as support for a conformational bias in gamma-TuRC that is exerted by CDK5RAP2. The authors propose a model in which CAMSAPs exploit an imperfect interface between γ -TuRC and the nucleated microtubule to bind and release the microtubule from gamma-TuRC. CDK5RAP2 is proposed to enhance nucleation and prevent microtubule release by CAMSAPs by improving the fit between γ -TuRC and the microtubule en.

While a similar in vitro setup including activation of gamma-TuRC was already described in previous studies (Consolati et al., 2020, Thawani et al., 2020), this study provides additional support for the somewhat controversial finding of CDK5RAP2 being sufficient for activating gamma-TuRC in vitro. In addition, the study provides insight into how CAMSAPs may be able to release microtubules from the gamma-TuRC template. The observation that CLASP2 stimulates nucleation from gamma-TuRC in vitro is also new and provides means for follow-up studies, e.g. its mechanistic basis. Perhaps the most interesting part of the manuscript is that CDK5RAP2 inhibits CAMSAP3-driven microtubule release from gamma-TuRC, which suggests that gamma-TuRC may nucleate microtubules in two different ways, at least in vitro. One way would be through a nucleation-complex that does not tightly cap the microtubule end and allows it to be released by CAMSAP2/3. The other way would be through an activator/nucleator-complex that stably caps the microtubule end and prevents CAMSAP binding. While this is an attractive concept that bears novelty, it is not sufficiently supported by the data presented and several conclusions are speculative. Another major issue is conceptual: there is no evidence that imperfect templates (non-activated, asymmetric gamma-TuRCs) as used here would ever nucleate microtubules in cells. In fact, the asymmetric conformation of cytosolic gamma-TuRC were activators reside (e.g. Zupa et al. 2021, Curr Opin Struct Biol; Liu et al. 2020, Nature; Consolati et al., 2020, Dev Cell; Zimmerman et al. 2020, Science Adv). I also have technical concerns regarding the quality of the gamma-TuRC preparation and the high variability in several experiments. For the regulators, recombinant expression is used and the quality of the proteins seems fair, but the authors do not test any mutant interactors to support their findings and add mechanistic detail, which would make the manuscript richer.

Major points:

1. The quality of the gamma-TuRC preparation is unclear. This is of particular concern as the authors interpret the observed effects and draw conclusions based on the assumption that their assays contain fully assembled gamma-TuRC, a prerequisite and fundamental to the whole study. Previous studies have demonstrated that stable gamma-TuRC subcomplexes exist that contain GCP3 (Zimmermann et al. 2020, Wieczorek et al. 2021, Wurtz et al. 2022). In the Methods section, the authors state they have taken care to avoid photobleaching during protein purification. This is reflected in Fig.1b as the authors measure a good correlation between GFP intensity alone vs. GFP-EB3 intensity. By contrast, the fluorescence intensity distribution of GCP3-GFP in Fig.1b is very broad. Assuming the authors have taken equal care during gamma-TuRC purification and TIRF setup, this broad distribution suggests that the gamma-TuRC preparation is very heterogenous. To ensure that the GCP3-containing complexes that the authors analyse in their assays are indeed mostly gamma-TuRCs, they need to provide a more quantitative measure of how much full gamma-TuRC vs. gamma-TuRC subcomplexes are in the preparation. The same applies to their gamma-TuRC preparation based on GCP6-

purification. Mass spec and cryo-EM analyses are not suited to address this. How does this material fractionate on a sucrose gradient? A related concern is that only a western blot analysis of the gamma-TuRC prep is shown. I understand that the yields may be low, but a protein stain would be needed to estimate overall composition and purity.

2. Related to point 1: even if there was a significant number of full gamma-TuRCs in the prep, as long as subcomplexes are also present, how can the authors distinguish between immobilized partial and full complexes on their glass? This is an important issue in single molecule studies since certain events (e.g. MT release) may occur specifically in association with a certain type of complex.

3. There is no explanation or discussion of what makes CAMSAP1 different from CAMSAP2/3 and how this relates to the observed effects regarding microtubule release.

4. Fig. 2g,h; 3f,g; 4c,d: There are too few data points considering the highly variable data (in several cases even the control condition ranges from 0-100%). I am not convinced by conclusions based on these data. More replicates/data points are needed to make it more robust.

5. It is stated that: "It was shown previously that a subcomplex containing GCP4, 5 and 6 is by itself nucleation-incompetent and needs to be supplemented with cell extracts containing GCP2 and 3 to nucleate microtubules". However, another recent study (Wieczorek et al., 2021, JCB) has shown that GCP6-containing partial complexes have similar (low) nucleation activity as (non-activated) full gamma-TuRC. Therefore, the authors cannot argue that their GCP6-based purification contains more full gamma-TuRCs than their GCP3-based purification based on (low) activity. In any case the complex identity remains unclear, as it is not directly shown.

6. The cryo-EM data do not show a conformational change in the gamma-TuRC containing CDK5RAP2, yet the rather indirect microtubule end binding assay is interpreted to indicate that this is the case. However, while CDK5RAP2 slightly reduces binding of gamma-TuRC to 14 pf microtubules, there is no effect on binding to 13 pf microtubules (opposite to what would be expected if CDK5RAP2 binding facilitates a more perfect, 13-fold template). Based on this assay and this result the conclusion that CDK5RAP2 binding induces a conformational change or flexibility in gamma-TuRC towards a better template remains speculative.

7. The authors show that CDK5RAP2 binding to gamma-TuRC makes it resistant to CAMSAP-mediated microtubule minus end growth and release. Contrary to the authors' conclusion, this could as well be interpreted as evidence against a role of CAMSAPs in microtubule release from gamma-TuRCs, at least in a physiological setting in cells. Here, nucleation needs to be tightly controlled and the predominant nucleation mechanism likely depends on activated gamma-TuRCs that properly fit the microtubule minus end. Thus, in cells and without the help of additional factors, based on the author's model CAMSAPs would not be able to access the minus ends of gamma-TuRC-nucleated microtubules.

Minor points:

1. In the introduction it is stated "...likely because CDK5RAP2 was present in excess, or due to the autoinhibitory regulation of CDK5RAP2 that is controlled by its phosphorylation state". The cited studies are not suited to support this statement, work by the Conduit lab should be included.

2. What are the error bars in 2j?

3. The results part often describes data presented in extended figures only partially, and the remaining data is only described later in a different section. This makes it hard to follow the descriptions and corresponding figures.

Reviewer #2 (Remarks to the Author):

Microtubule nucleation is mediated by the gamma tubulin ring complex (g-TuRC) and is among the most critical and poorly explored activities, which are conserved across eukaryotes. Microtubule nucleation regulates the organization of this cytoskeleton network during cell division and cell

differentiation and impacts many biological processes in eukaryotes. Although recent advances have resolved the molecular components and organization of the g-TURC complex, how g-TURC is impacted by regulatory activities of a vast array of factors such as ch-TOG, CLASP, EB3, CDK5RAP2 and minus-end binding proteins CAMSAP. The interactions of these factors to activate and regulate the microtubule nucleation/elongation from g-TURC have remained mostly unknown due to lack of biochemical reconstitution studies.

In this manuscript, Rai et al undertake a massive effort to explore the role of g-TURC in microtubule nucleation and how its activities are regulated by the above factors. The authors prepare CRISPER CAS genetic replacements leading to GFP-tagged GCP3 and GFP-GCP6 in HEK293 cells and purify full active g-TURC complexes. The authors study how a group of regulators impacts the nucleation process, in addition to each regulatory activities on microtubule polymerization. The most crucial discoveries involve the role of the CLASP2, CDK5RAP2 in activating the g-TURC microtubule nucleation if premixed with prior to microtubule assembly. The authors turn their attention to understand how these factors impact the release of microtubules from g-TURC after nucleation under the effect of minus end stabilizing proteins, CAMSAP1-3. The work reveals how microtubule minus ends are partially freed from g-TURC, released and stabilized or and polymerized by CAMSAP1, revealing the role of CDK5RAP2 in inhibiting microtubule release.

The studies presented are worthy of publication at Nature Cell biology and will have an impact on wide variety of audiences. However, there are concerns regarding the presentation and the additional data that should be presented that would enhance the reader experience and would set this paper as a landmark paper in the field.

The major concerns relate to the quality of the figure presentation which are currently below the standard to reveal the details of the conclusions in the images / videos. I am also making suggestions regarding biochemical studies to discern how the regulators may or may not interact directly with g-TURC:

- 1) Quality of the image/video presentation and information provided in kymograph presentations which call into question some important qualitative observations described by authors throughout. I believe the image/video presentation in the manuscript do not do justice to the observations presented in the text. This can be repaired by adding additional videos, or montages for many different experiments to help the reader see some of the complex behaviors presented.
- 2) Thee quality of kymographs revealing the localization of regulators do not match the quality of conclusions. For example, CLASP or ch-TOG localization is very difficult to discern. This is likely due to fluorescence thresholds, color overlays or compressed time frames. Also repairing the quality of the kymograph presentation to present narrower time frames or localized regions to focus on the detailed of localization or behavior of regulators such as CLASP, chTOG or CDK5RAP2.
- 3) Some of the conclusions for how these regulators interact with g-TURC would benefit dramatically from biochemical analyses studying the potential interactions these regulators may have with the g-TURC, with and without tubulin. For example, the lack of difference negative reconstructions does not rule out that CDK5RAP2 may or may not bind to g-TURC under the conditions used for the assembly of the complex and providing evidence that CDK5RAP2 indeed forms complexes with g-TURC in the same conditions would help address concerns of how to interpret these reconstructions. Similar experiments should be performed with ch-TOG and CLASP2 to help the reader understand their interactions with g-TURC in the conditions used in this paper.

Reviewer #3 (Remarks to the Author):

Rai and coworkers report that reconstitution of microtubule nucleation from γ -TuRCs. First, they test the effects of nucleation promoting factors. This section is not novel, but since this has been a controversial area, it is important to publish these data as part of this report because it provides confirmation of recent results and validates their reconstitution system. They then examine the activity of CAMSAPs, microtubule minus-end binding proteins with γ -TuRC nucleated microtubules. They find that surprisingly, all CAMSAPs can bind to the minus ends of microtubules even with γ -TuRC bound there. Since CAMSAPs bind curved microtubule protofilaments, this strongly suggests that γ -TuRCs are not a perfect cap on the end of the microtubule but must leave some curving protofilaments available to bind CAMSAPs, thus providing evidence for the poor match between γ -TuRC and a microtubule even when bound to the microtubule. The imperfect match was apparent from the recent structures of free γ -TuRCs, however the results shown here show that there is an imperfect match even on microtubules nucleated by γ -TuRC. The authors go on to show that CAMSAPs 2 and 3 can both increase the rate of microtubule release from γ -TuRC. Finally, CDK5RAP2, which binds γ -TuRC and anchors it to the centrosome, inhibits release of the microtubule from γ -TuRC by CAMSAP. This suggests that CDK5RAP2 induces a better fit between the end of the microtubule and γ -TuRC as hypothesized by previous work in yeast. This is a well done, well quantified, and interesting study that significantly adds to our knowledge of microtubule nucleation and its regulation.

1. Page 3: "Furthermore, since the microtubule nucleating activity of purified γ -TuRC turned out to be quite low^{18, 22}." The low microtubule nucleating activity of γ -TuRC is also clear in one of the original papers describing γ -TuRC, which I think should also be referenced here. See figure legend to Figure 2 in Oegema et al., JCB 144:721.
2. Page 7: "Finally, we also examined the effect of chTOG, because it can weakly promote microtubule nucleation from free tubulin³⁵ and strongly promote γ -TuRC-dependent microtubule nucleation¹⁸, and its *Xenopus* homolog XMAP215 can synergize with γ -TuRC²⁵ and promote outgrowth from seeds³⁶." This is a thorough brief review of the literature, but should also include a reference to King et al., MBOC, 2020, 31:2187, which quantified the ability of XMAP215 to promote nucleation from lateral gamma-tubulin arrays.
3. Figure 1c and all kymographs. Please label the origin. Without the origin labeled, kymographs are difficult to interpret.
4. Figure 3i: These images do not clearly show that the authors can detect colocalization. Improved images showing what counts as colocalized and what does not would be helpful.

Author Rebuttal to Initial comments

Point-by-point Response to the Reviewers' comments

Reviewer #1 (Remarks to the Author):

In this study the authors investigate how microtubule nucleation and minus end capping by the nucleator gamma-TuRC are regulated. The authors use a TIRF-based assay that measures in vitro microtubule nucleation from purified and immobilized gamma-TuRC to test the effect of various, in part previously reported gamma-TuRC interactors including chTOG, CLASP2, and CDK5RAP2. Using this system, the authors show that all of these can stimulate gamma-TuRC nucleation activity in vitro. The authors then test the effect of CAMSAP proteins and find that they bind to the minus-end of a newly nucleated microtubule bound to gamma-TuRC and that CAMSAP2 and CAMSAP3, but not CAMSAP1, trigger minus end elongation and microtubule release from gamma-TuRC. Further, the authors find that pre-incubation of gamma-TuRC with the activator CDK5RAP2, abolishes CAMSAP3-dependent microtubules release from gamma-TuRC. Finally, the authors provide evidence that CDK5RAP2-bound gamma-TuRC preferentially binds to 12/13- protofilament over 14-protofilament microtubule ends, which they interpret as support for a conformational bias in gamma-TuRC that is exerted by CDK5RAP2. The authors propose a model in which CAMSAPs exploit an imperfect interface between γ -TuRC and the nucleated microtubule to bind and release the microtubule from gamma-TuRC. CDK5RAP2 is proposed to enhance nucleation and prevent microtubule release by CAMSAPs by improving the fit between γ -TuRC and the microtubule end.

While a similar in vitro setup including activation of gamma-TuRC was already described in previous studies (Consolati et al., 2020, Thawani et al., 2020), this study provides additional support for the somewhat controversial finding of CDK5RAP2 being sufficient for activating gamma-TuRC in vitro. In addition, the study provides insight into how CAMSAPs may be able to release microtubules from the gamma-TuRC template. The observation that CLASP2 stimulates nucleation from gamma-TuRC in vitro is also new and provides means for follow-up studies, e.g. its mechanistic basis. Perhaps the most interesting part of the manuscript is that CDK5RAP2 inhibits CAMSAP3-driven microtubule release from gamma-TuRC, which suggests that gamma-TuRC may nucleate microtubules in two different ways, at least in vitro. One way would be through a nucleation-complex that does not tightly cap the microtubule end and allows it to be released by CAMSAP2/3. The other way would be through an activator/nucleator-complex that stably caps the microtubule end and prevents CAMSAP binding.

While this is an attractive concept that bears novelty, it is not sufficiently supported by the data presented and several conclusions are speculative. Another major issue is conceptual: there is no evidence that imperfect templates (non-activated, asymmetric gamma-TuRCs) as used here would ever nucleate microtubules in cells. In fact, the asymmetric conformation of cytosolic gamma-TuRC was interpreted as a regulatory feature, preventing unregulated nucleation and restricting it to MTOCs where activators reside (e.g. Zupa et al. 2021, Curr Opin Struct Biol; Liu et al. 2020, Nature; Consolati et al., 2020, Dev Cell; Zimmerman et al. 2020, Science Adv).

We respectfully disagree: it is currently not known whether gamma-TuRC is symmetric or asymmetric when it nucleates microtubules in cells. As the reviewer indicates, the observed gamma-TuRC asymmetry has indeed been interpreted as a regulatory feature, but no actual proof for this concept has been provided until now. There is no doubt that in vitro, asymmetric and even partial gamma-TuRCs can nucleate microtubules (see, for example, Wieczorek et al., 2021, J. Cell Biol., <https://doi.org/10.1083/>

jcb.202009146). Furthermore, unlike GCP2 and GCP3, GCP4, 5 or 6 are not essential for viability of *Drosophila* (Verollet et al., 2006, J. Cell Biol., <https://doi.org/10.1083/jcb.200511071>; Vogt et al., 2006, Development, <https://doi.org/10.1242/dev.02570>), fission yeast (Anders et al., 2006, Mol. Biol. Cell, <https://doi.org/10.1091/mbc.e05-11-1009>) or *Aspergillus* cells (Xiong and Oakley, 2009, J. Cell Sci., <https://doi.org/10.1242/jcs.059196>). Cells depleted of GCP4, 5 and 6 do nucleate microtubules from gamma-tubulin spots localized to centrosomes, indicating that symmetric gamma-TuRCs may not be essential in cells (Cota et al., 2017, J. Cell Sci., <https://doi.org/10.1242/jcs.195321>). Importantly, our approach of combining gamma-TuRC with CAMSAPs in the same assay provides the first direct readout probing the attachment between the gamma-TuRC and the microtubule it nucleates. This assay shows that the mechanisms of gamma-TuRC activation can be fundamentally different, and that these mechanisms have consequences for the subsequent destiny of nucleated microtubules. In fact, not all activator/nucleator-complexes have to work in the same way, by stably capping microtubule ends. We show that, unlike CDK5RAP2, CLASP2 activator/nucleator complex does not tightly cap the microtubule end and allows CAMSAP-binding and subsequent microtubule release. Therefore, depending on the gamma-TuRC activator, microtubule minus ends will either stay attached to gamma-TuRC that nucleated them or will detach and become stabilized by CAMSAPs, allowing the same gamma-TuRC to be reused for another round of microtubule nucleation.

To support this conclusion, in the revised version of the manuscript we analyzed cells where CDK5RAP2 and its paralog Myomegalin were simultaneously knocked out, to exclude their potential redundancy. We found that these knockout cells had more CAMSAP2-decorated non-centrosomal microtubules, whereas overexpression of CDK5RAP2 suppressed the formation of CAMSAP2-bound minus-ends (new Figure 5 of the revised manuscript). Highly pertinent in this respect is the observation that CDK5RAP2, while potent in activating gamma-TuRC, is not at all essential for mammalian cell survival, and the knockout of CDK5RAP2 together with its paralog Myomegalin has no major impact on microtubule density (new Extended data Figure 6c-e). In contrast, CLASPs are major regulators of microtubule abundance in cells (Mimori-Kiyosue et al., 2005, J. Cell Biol., <https://doi.org/10.1083/jcb.200405094>; Lawrence et al., 2020, J. Cell Sci., <https://doi.org/10.1242/jcs.243097>). The major novel conclusion of our study is thus that nucleation-promoting factors can not only stimulate formation of microtubules at the gamma-TuRC-containing sites but also control the destiny of their minus ends and thus microtubule organization, such as the balance between centrosomal versus non-centrosomal microtubules.

I also have technical concerns regarding the quality of the gamma-TuRC preparation and the high variability in several experiments.

We fully agree that we should have provided a better characterization of the gamma-TuRC preparation we are using, and in the revised paper, we have included multiple additional analyses of our gamma-TuRC, as outlined below.

For the regulators, recombinant expression is used and the quality of the proteins seems fair, but the authors do not test any mutant interactors to support their findings and add mechanistic detail, which would make the manuscript richer.

Our study already includes combinations of six recombinant proteins (CDK5RAP2, CLASP2, chTOG, CAMSAP1, CAMSAP2 and CAMSAP3), and it is not feasible to perform and describe detailed analysis of their mutants together with the other data, including newly added work in cells, in one manuscript. Furthermore, detailed domain analyses have already been described for CAMSAPs (Jiang et al., 2014, Dev.

Cell, <https://doi.org/10.1016/j.devcel.2014.01.001>; Hendershott and Vale, 2014, Proc Natl Acad Sci U S A., <https://doi.org/10.1073/pnas.1404133111>; Liu and Shima, 2023, Life Sci. Alliance, <https://doi.org/10.26508/lsa.202201714>), CDK5RAP2 (Tovey et al., 2021, J. Cell Biol., <https://doi.org/10.1083/jcb.202010020>; Rale et al., 2022, eLife, <https://doi.org/10.7554/eLife.80053>; Yang et al., 2023, J. Cell Biol., <https://doi.org/10.1083/jcb.202007101>) and chTOG (Thawani et al., 2018, Nat. Cell Biol., <https://doi.org/10.1038/s41556-018-0091-6>; Ali et al., 2023, Nat. Commun., <https://doi.org/10.1038/s41467-023-35955-w>). Moreover, for CAMSAPs, we make use of the different behavior of the three paralogs (CAMSAP1 vs CAMSAP2/CAMSAP3) to show that not just CAMSAP binding to the gamma-TuRC associated minus end, but also microtubule decoration is important for microtubule release from gamma-TuRC.

We have tested whether individual domains of CLASP2, the novel nucleation activator we have characterized here, are sufficient to promote nucleation by gamma-TuRC, but found this not to be the case. Therefore, unlike the activity of CLASP2 in microtubule catastrophe suppression and microtubule repair that requires a single TOG domain (Aher et al., 2018, Dev. Cell, <https://doi.org/10.1016/j.devcel.2018.05.032>; Aher et al., 2020, Curr. Biol., <https://doi.org/10.1016/j.cub.2020.03.070>), CLASP2 uses multiple domains to promote gamma-TuRC mediated nucleation, and the exact mechanism would require separate study.

Major points:

1. The quality of the gamma-TuRC preparation is unclear. This is of particular concern as the authors interpret the observed effects and draw conclusions based on the assumption that their assays contain fully assembled gamma-TuRC, a prerequisite and fundamental to the whole study. Previous studies have demonstrated that stable gamma-TuRC subcomplexes exist that contain GCP3 (Zimmermann et al. 2020, Wieczorek et al. 2021, Wurtz et al. 2022). In the Methods section, the authors state they have taken care to avoid photobleaching during protein purification. This is reflected in Fig.1b as the authors measure a good correlation between GFP intensity alone vs. GFP-EB3 intensity. By contrast, the fluorescence intensity distribution of GCP3-GFP in Fig.1b is very broad. Assuming the authors have taken equal care during gamma-TuRC purification and TIRF setup, this broad distribution suggests that the gamma-TuRC preparation is very heterogenous. To ensure that the GCP3-containing complexes that the authors analyse in their assays are indeed mostly gamma-TuRCs, they need to provide a more quantitative measure of how much full gamma-TuRC vs. gamma-TuRC subcomplexes are in the preparation. The same applies to their gamma-TuRC preparation based on GCP6-purification. Mass spec and cryo-EM analyses are not suited to address this. How does this material fractionate on a sucrose gradient? A related concern is that only a western blot analysis of the gamma-TuRC prep is shown. I understand that the yields may be low, but a protein stain would be needed to estimate overall composition and purity.

We fully agree with this comment. In the revised manuscript, we have improved the characterization of our gamma-TuRC in several ways. We have characterized the gamma-TuRC preparations by mass photometry (new Extended data Figure 1i,j), sucrose gradient fractionation (new Extended data Figure 1f) and mass spectrometry quantitation using intensity-based absolute quantification (iBAQ) algorithm (new Extended data Figure 1h and 3i, iBAQ ratio) to estimate overall composition and purity. Mass photometry and sucrose gradient fractionation analyses showed that both complete and incomplete gamma-TuRC complexes are present in our preparations. Furthermore, we have made use of the fact that we isolate gamma-TuRC from cells where all GCP3 subunits are labeled with GFP, and therefore we can use single

molecule counting to estimate the number of GCP3 subunits in the complexes we study. We note that this method does have limitations, because even if all complexes would be complete and contain 5 GCP3 molecules, the overall distribution of fluorescence intensities will still be broad, because of 1) wide distribution of the intensities of single GFPs, which makes the distributions of 2xGFP,3xGFP, etc, even wider; 2) some GFP molecules may not be mature, can show photoblinking or can be photobleached or damaged, so effectively even some complete complexes would have lower intensity. Still, the quantification of GFP molecules present in our gamma-TuRC preparations immobilized on glass in comparison to GFP monomers and GFP-EB3 dimers can be used as an indication and provides evidence for the presence of a significant number of complete gamma-TuRC complexes along with the partial subcomplexes (complexes with 4-5 GFPs, new Figure 1a and Extended data Figure 1k). The presence of complete complexes was also confirmed by 3D reconstruction of negative stain transmission EM micrographs (Figure 1b).

Given that both complete and incomplete complexes are present in our gamma-TuRC samples, we next wanted to know how this relates to their ability to nucleate microtubules, be activated by CDK5RAP2 and CLASP2, recruit CAMSAPs and release the nucleated microtubule. To address these questions, we have measured fluorescence intensity of individual gamma-TuRCs for all these events. This approach had an additional caveat, because, whereas direct immobilization on glass provided excellent distinction between GFP monomers and the dimeric GFP-containing control, GFP-EB3, the intensity of GFP-EB3 dimers appeared underestimated when the same proteins were attached to passivated surface using anti-GFP nanobodies, which were used in all our microtubule nucleation assays (compare Figure 1a with new Extended data Figure 4a,b). Therefore, we could not obtain precise numbers for the GFP molecules present within each gamma-TuRC complex we detected. Still, the quantification has revealed several interesting trends:

1. Both brighter and dimmer (likely complete and partial) gamma-TuRC complexes could nucleate microtubules.
2. Both complete and partial gamma-TuRC complexes could be activated by CDK5RAP2 and CLASP2. However, both CDK5RAP2 and CLASP2 preferentially activated brighter and thus likely complete complexes.
3. CAMSAP3 binding and release occurred at dim and bright gamma-TuRC complexes both in control conditions and when premixed with CLASP2 (Extended data Figure 4i and k). These data indicate that although CLASP2 does not protect the interface between gamma-TuRC and microtubule minus end from CAMSAP3 binding and subsequent release, this is not due to the fact that it preferentially generates microtubules from incomplete gamma-TuRC subcomplexes. In contrast, the few events of CAMSAP3 binding and microtubule release observed in the presence of CDK5RAP2 appeared to occur with dimmer and thus likely partial gamma-TuRC complexes. It should be noted that because CDK5RAP2 activated gamma-TuRC complexes with a broad range of intensities, it can likely inhibit CAMSAP3 binding to the minus ends of microtubules bound to incomplete gamma-TuRC rings. This could be due to the ability of full length CDK5RAP2, used in our experiments, to bind not only to gamma-TuRC, but also to microtubules. To support the idea that CDK5RAP2 may directly interact with the microtubule lattice, we have included new in vitro reconstitution assays with mCherry-CDK5RAP2 and dynamic microtubules grown from GMPCPP seeds (new Extended data Figure 5k,l).

All these data and considerations were included in the Results and Discussion of the revised paper, and the model described at the end of the manuscript has been modified accordingly.

2. Related to point 1: even if there was a significant number of full gamma-TuRCs in the prep, as long as subcomplexes are also present, how can the authors distinguish between immobilized partial and full complexes on their glass? This is an important issue in single molecule studies since certain events (e.g. MT release) may occur specifically in association with a certain type of complex.

As described above, this comment was addressed by additional single molecule GFP-counting experiments. Our data indicate that, in agreement with other studies, both complete and incomplete gamma-TuRCs can nucleate microtubules and be activated by CDK5RAP2 and CLASP2, although both nucleation-promoting factors preferentially act on complete ones. Furthermore, CAMSAP3 binding and microtubule release do not preferentially occur on dimmer, incomplete gamma-TuRC control or CLASP2-activated subcomplexes. CDK5RAP2 can also activate both complete and incomplete complexes, and suppress CAMSAP3 binding to both types of complexes, though the few events of microtubule release from CDK5RAP2-activated gamma-TuRC occurred at complexes that were likely incomplete.

3. There is no explanation or discussion of what makes CAMSAP1 different from CAMSAP2/3 and how this relates to the observed effects regarding microtubule release.

As was briefly mentioned in the paper and explained in more detail in the revised version, all CAMSAPs have a CKK domain that recognizes curved protofilaments at free minus ends of microtubules. However, CAMSAP1 does not decorate and stabilize minus-end grown microtubule lattice, because it lacks additional microtubule shaft-binding domains, some of which can also alter the conformation of microtubule lattice (Jiang et al., 2014, Dev. Cell, <https://doi.org/10.1016/j.devcel.2014.01.001>; Hendershott and Vale, 2014, Proc Natl Acad Sci U S A., <https://doi.org/10.1073/pnas.1404133111>; Liu and Shima, 2023, Life Sci. Alliance, <https://doi.org/10.26508/lsa.202201714>).

4. Fig. 2g,h; 3f,g; 4c,d: There are too few data points considering the highly variable data (in several cases even the control condition ranges from 0-100%). I am not convinced by conclusions based on these data. More replicates/data points are needed to make it more robust.

We have added more data points to figures 2h,i; 3f,g; 4c,d to make the data more robust. We would also like to point out that there is more variability in control conditions because nucleation activity of gamma-TuRC by itself is very low (< 2% complexes can nucleate microtubules), and only ~30% of those complexes colocalize to CAMSAPs of which even fewer could release microtubules. However, we have added more replicates and increased the number of complexes that can be analyzed per experiment/replicate.

5. It is stated that: "It was shown previously that a subcomplex containing GCP4, 5 and 6 is by itself nucleation-incompetent and needs to be supplemented with cell extracts containing GCP2 and 3 to nucleate microtubules". However, another recent study (Wieczorek et al., 2021, JCB) has shown that GCP6-containing partial complexes have similar (low) nucleation activity as (non-activated) full gamma-TuRC. Therefore, the authors cannot argue that their GCP6-based purification contains more full gamma-TuRCs than their GCP3-based purification based on (low) activity. In any case the complex identity remains unclear, as it is not directly shown.

We agree that GCP6-containing gamma-TuRCs might also be partial, and therefore counting of GCP3 subunits has provided more direct evidence, as discussed above. Still, we feel that the data using an

alternative GCP subunit to purify gamma-TuRC is a valuable enrichment of our dataset because it shows that microtubule release can also occur from GCP6-containing gamma-TuRC complexes. We have modified the text accordingly.

6. The cryo-EM data do not show a conformational change in the gamma-TuRC containing CDK5RAP2, yet the rather indirect microtubule end binding assay is interpreted to indicate that this is the case. However, while CDK5RAP2 slightly reduces binding of gamma-TuRC to 14 pf microtubules, there is no effect on binding to 13 pf microtubules (opposite to what would be expected if CDK5RAP2 binding facilitates a more perfect, 13-fold template). Based on this assay and this result the conclusion that CDK5RAP2 binding induces a conformational change or flexibility in gamma-TuRC towards a better template remains speculative.

We agree that the idea about a conformational change is only speculative as we do not provide direct evidence. We would like to clarify that the negative-stain EM data have been obtained under conditions without free tubulin, where gamma-TuRC will not nucleate microtubules. Therefore, like reviewer #2 indicated in their point #3, we cannot rule out the possibility that there is no conformational change because such a change occurs when a gamma-TuRC nucleates a microtubule or is bound to a complete microtubule. Another possibility, based on our new data (Extended data Figure 4c,d,f-h,j and 5k,l), is that CDK5RAP2 is somehow strengthening the interface between gamma-TuRC and the minus end of the newly nucleated microtubule to prevent CAMSAP binding and microtubule release. We have adjusted the discussion to make this point more clear. Nevertheless, we think that reduced binding of gamma-TuRC to 14-pf microtubules is a useful piece of data, which provides interesting albeit not conclusive evidence for an effect of CDK5RAP2 on gamma-TuRC structure. The lack of improved gamma-TuRC binding to 13-pf microtubules in the presence of CDK5RAP2 is likely due to the fact that we already see a high capping efficiency in control condition even without CDK5RAP2. The remaining microtubules may not be capped by gamma-TuRC because their terminal protofilament flares are of different length, as often observed by cryo-electron tomography.

7. The authors show that CDK5RAP2 binding to gamma-TuRC makes it resistant to CAMSAP-mediated microtubule minus end growth and release. Contrary to the authors' conclusion, this could as well be interpreted as evidence against a role of CAMSAPs in microtubule release from gamma-TuRCs, at least in a physiological setting in cells. Here, nucleation needs to be tightly controlled and the predominant nucleation mechanism likely depends on activated gamma-TuRCs that properly fit the microtubule minus end. Thus, in cells and without the help of additional factors, based on the author's model CAMSAPs would not be able to access the minus ends of gamma-TuRC-nucleated microtubules.

This comment would be fully valid if CDK5RAP2 were an essential gamma-TuRC activator participating in all microtubule nucleation events. However, gamma-TuRC persists at the centrosomes and can nucleate microtubules in the absence of CDK5RAP2 (see, for example, Gavilan et al., 2018, EMBO Rep., <https://doi.org/10.15252/embr.201845942>; Ali et al., 2023, Nat. Commun., <https://doi.org/10.1038/s41467-023-35955-w>). Importantly, in the revised manuscript, we have added a completely new dataset where we analyzed the simultaneous knockout of CDK5RAP2 and its paralog Myomegalin, to exclude their potential redundancy, and found that the loss of these two proteins reduced but did not abolish microtubule nucleation at the centrosome, did not at all affect microtubule density, but increased the

number of CAMSAP2-positive non-centrosomal minus ends (new Figure 5). In contrast, overexpression of CDK5RAP2 alone was sufficient to diminish the number of CAMSAP2-labeled minus ends even in control cells (new Figure 5). We further showed that when microtubule nucleation at the centrosome was strongly upregulated by first disassembling microtubules with nocodazole and then washing out the drug, there was a strong concentration of CAMSAP2 signal around the centrosome already 1 minute after the washout, pointing to the very rapid release of newly nucleated microtubules (new Figure 5c-f), also seen previously (Jiang et al., 2014, *Dev. Cell*, <https://doi.org/10.1016/j.devcel.2014.01.001>; Dong et al., 2017, *J. Cell Sci.*, <https://doi.org/10.1242/jcs.198010>). This phenomenon was readily observed in control cells, and the emergence of CAMSAP2 signal around the centrosome in nocodazole washout assays was strongly suppressed by overexpressing CDK5RAP2 but enhanced upon the loss of CDK5RAP2 and Myomegalin (Figure 5e-f). These data indicate that CDK5RAP2 and its paralog Myomegalin are not essential cofactors of gamma-TuRC-mediated microtubule nucleation. CDK5RAP2 increases nucleation efficiency and at the same time prevents CAMSAP-mediated detachment from the nucleation site, thus suppressing formation of CAMSAP-bound microtubules that are not attached to microtubule-organizing centers. The important physiological role of CAMSAP2/3 in microtubule release from the nucleation sites is in line with previous publications demonstrating the absence of non-centrosomal microtubules in epithelial cells where these proteins were knocked out (Tanaka et al., 2012, *Proc. Natl. Acad. Sci. U S A*, <https://doi.org/10.1073/pnas.1218017109>; Jiang et al., 2014, *Dev. Cell*, <https://doi.org/10.1016/j.devcel.2014.01.001>).

Furthermore, we showed that gamma-TuRC-mediated microtubule nucleation can also be strongly activated by CLASP2, which has no effect on CAMSAP binding to the minus ends of gamma-TuRC-attached microtubules and their release, although, similar to CDK5RAP2, it also preferentially activates likely complete gamma-TuRCs. These data help to explain why non-centrosomal, CAMSAP-stabilized microtubule minus ends largely disappear in cells depleted for CLASP1 and CLASP2 (Wu et al., 2016, *Dev Cell*, <https://doi.org/10.1016/j.devcel.2016.08.009>). These data also indicate that the mechanisms of gamma-TuRC activation can be fundamentally different and that these mechanisms have consequences for the subsequent destiny of nucleated microtubules.

Minor points:

1. *In the introduction it is stated "...likely because CDK5RAP2 was present in excess, or due to the autoinhibitory regulation of CDK5RAP2 that is controlled by its phosphorylation state". The cited studies are not suited to support this statement, work by the Conduit lab should be included.*

We apologize for the mistake in the citation. We have added the correct citations for the relevant studies from the Conduit lab and the Qi lab.

2. *What are the error bars in 2j?*

We have included the description of the error bar (SEM) in the legend for the former Figure 2j, which became Figure 2k in the revised manuscript.

3. *The results part often describes data presented in extended figures only partially, and the remaining data is only described later in a different section. This makes it hard to follow the descriptions and corresponding figures.*

We have added new data, reorganized the already existing extended data figures and improved their description in the text.

Reviewer #2 (Remarks to the Author):

Microtubule nucleation is mediated by the gamma tubulin ring complex (g-TURC) and is among the most critical and poorly explored activities, which are conserved across eukaryotes. Microtubule nucleation regulates the organization of this cytoskeleton network during cell division and cell differentiation and impacts many biological processes in eukaryotes. Although recent advances have resolved the molecular components and organization of the g-TURC complex, how g-TURC is impacted by regulatory activities of a vast array of factors such as ch-TOG, CLASP, EB3, CDK5RAP2 and minus-end binding proteins CAMSAP. The interactions of these factors to activate and regulate the microtubule nucleation/elongation from g-TURC have remained mostly unknown due to lack of biochemical reconstitution studies.

In this manuscript, Rai et al undertake a massive effort to explore the role of g-TURC in microtubule nucleation and how its activities are regulated by the above factors. The authors prepare CRISPER CAS genetic replacements leading to GFP-tagged GCP3 and GFP-GCP6 in HEK293 cells and purify full active g-TURC complexes. The authors study how a group of regulators impacts the nucleation process, in addition to each regulatory activities on microtubule polymerization. The most crucial discoveries involve the role of the CLASP2, CDK5RAP2 in activating the g-TURC microtubule nucleation if premixed with prior to microtubule assembly. The authors turn their attention to understand how these factors impact the release of microtubules from g-TURC after nucleation under the effect of minus end stabilizing proteins, CAMSAP1-3. The work reveals how microtubule minus ends are partially freed from g-TURC, released and stabilized or and polymerized by CAMSAP1, revealing the role of CDK5RAP2 in inhibiting microtubule release.

The studies presented are worthy of publication at Nature Cell biology and will have an impact on wide variety of audiences. However, there are concerns regarding the presentation and the additional data that should be presented that would enhance the reader experience and would set this paper as a land mark paper in the field.

The major concerns relate to the quality of the figure presentation which are currently below the standard to reveal the details of the conclusions in the images / videos. I am also making suggestions regarding biochemical studies to discern how the regulators may or may not interact directly with g-TURC:

1) Quality of the image/video presentation and information provided in kymograph presentations which call into question some important qualitative observations described by authors throughout. I believe the image/video presentation in the manuscript do not do justice to the observations presented in the text. This can be repaired by adding additional videos, or montages for many different experiments to help the reader see some of the complex behaviors presented.

We acknowledge that the data are often difficult to present, because in these assays, microtubules often do not stay in a single imaging plane and can pivot around their attachment point at the minus end. To improve the presentation, we have enlarged some images and kymographs, added insets, enlarged views and eight additional videos, inverted colors and labeled images and kymographs to better illustrate the complex behaviors and observations presented in the text.

2) The quality of kymographs revealing the localization of regulators do not match the quality of conclusions. For example, CLASP or ch-TOG localization is very difficult to discern. This is likely due to fluorescence thresholds, color overlays or compressed time frames. Also repairing the quality of the kymograph presentation to present narrower time frames or localized regions to focus on the detailed of localization or behavior of regulators such as CLASP, chTOG or CDK5RAP2.

In order to provide better visual support for our conclusions, we have replaced or added better quality kymographs, focused on localized regions using enlarged views, adjusted the color thresholds, inverted colors, drawn some schemes illustrating images and kymographs that were difficult to understand, and labeled kymographs to highlight the localization of regulators.

3) Some of the conclusions for how these regulators interact with g-TURC would benefit dramatically from biochemical analyses studying the potential interactions these regulators may have with the g-TURC, with and without tubulin. For example, the lack of difference negative reconstructions does not rule out that CDK5RAP2 may or may not bind to g-TURC under the conditions used for the assembly of the complex and providing evidence that CDK5RAP2 indeed forms complexes with g-TURC in the same conditions would help address concerns of how to interpret these reconstructions. Similar experiments should be performed with ch-TOG and CLASP2 to help the reader understand their interactions with g-TURC in the conditions used in this paper.

It is difficult to fully mimic the conditions of our in vitro reconstitution assays, which are essentially carried out in the single molecule regime, in bulk biochemical assays. To overcome this problem, we have carefully quantified colocalization of individual gamma-TuRCs with CDK5RAP2, CLASP2 and chTOG with and without tubulin (Figure 1e-j and Extended data Figure 2d, e.g. colocalization of gamma-TuRC premixed with 30 nM CDK5RAP2 without added tubulin is 30%). The reconstructions of negative stain transmission EM of gamma-TuRC incubated in the presence of CDK5RAP2 (Extended data Figure 5d-h,j) were performed exactly in the same conditions except that CDK5RAP2 concentration was increased to 120 nM. We do not see how biochemical binding assays would provide more conclusive data addressing this issue. Therefore, we have depicted the colocalization between regulators and gamma-TuRC better by adding clearly labeled enlarged views and insets in the revised version of the manuscript and elaborated on these data in the text.

Reviewer #3 (Remarks to the Author):

Rai and coworkers report that reconstitution of microtubule nucleation from γ -TuRCs. First, they test the effects of nucleation promoting factors. This section is not novel, but since this has been a controversial area, it is important to publish these data as part of this report because it provides confirmation of recent results and validates their reconstitution system. They then examine the activity of CAMSAPs, microtubule minus-end binding proteins with γ -TuRC nucleated microtubules. They find that surprisingly, all CAMSAPs can bind to the minus ends of microtubules even with γ -TuRC bound there. Since CAMSAPs bind curved microtubule protofilaments, this strongly suggests that γ -TuRCs are not a perfect cap on the end of the microtubule but must leave some curving protofilaments available to bind CAMSAPs, thus providing evidence for the poor match between γ -TuRC and a microtubule even when bound to the microtubule. The imperfect match was apparent from the recent structures of free γ -TuRCs, however the results shown here show that there is an imperfect match even on microtubules nucleated by γ -TuRC. The authors go on to show that CAMSAPs 2 and 3 can both increase the rate of microtubule release from γ -TuRC. Finally, CDK5RAP2, which binds γ -TuRC and anchors it to the centrosome, inhibits release of the microtubule from γ -TuRC by CAMSAP. This suggests that CDK5RAP2 induces a better fit between the end of the microtubule and γ -TuRC as hypothesized by previous work in yeast. This is a well done, well quantified, and interesting study that significantly adds to our knowledge of microtubule nucleation and its regulation.

1. Page 3: “Furthermore, since the microtubule nucleating activity of purified γ -TuRC turned out to be quite low^{18, 22}.” The low microtubule nucleating activity of γ -TuRC is also clear in one of the original papers describing γ -TuRC, which I think should also be referenced here. See figure legend to Figure 2 in Oegema et al., JCB 144:721.

We apologize for omitting this citation, which has been added.

2. Page 7: “Finally, we also examined the effect of chTOG, because it can weakly promote microtubule nucleation from free tubulin³⁵ and strongly promote γ -TuRC-dependent microtubule nucleation¹⁸, and its *Xenopus* homolog XMAP215 can synergize with γ -TuRC²⁵ and promote outgrowth from seeds³⁶.” This is a thorough brief review of the literature, but should also include a reference to King et al., MBOC, 2020, 31:2187, which quantified the ability of XMAP215 to promote nucleation from lateral gamma-tubulin arrays.

We apologize for omitting this citation, which has been added.

3. Figure 1c and all kymographs. Please label the origin. Without the origin labeled, kymographs are difficult to interpret.

This is an excellent suggestion. In the revised paper, we did our best to improve the clarity of data presentation by enlarging and more clearly labeling images and kymographs.

4. Figure 3i: These images do not clearly show that the authors can detect colocalization. Improved images showing what counts as colocalized and what does not would be helpful.

We have revised the panel (Figure 3i,j) by adding improved images, insets, a scheme and illustrations to depict colocalizing and non-colocalizing populations.

Decision Letter, first revision:

Dear Professor Akhmanova,

I apologize for the delay. Your manuscript, "CAMSAP-driven microtubule release from γ -TuRC and its regulation by nucleation-promoting factors", has now been seen by our original referees, who are experts in γ -TuRC and MT dynamics (referee 1); structural biology and microtubule dynamics (referee 2); and microtubule nucleation (referee 3). Please note that Reviewer #3 is satisfied with the current response, although they left only confidential comments currently. As you will see from their comments (attached below) they find this work of interest, but have raised some important points. Although we are also very interested in this study, we believe that their concerns should be addressed before we can consider publication in Nature Cell Biology.

Nature Cell Biology editors discuss the referee reports in detail within the editorial team, including the chief editor, to identify key referee points that should be addressed with priority, and requests that are overruled as being beyond the scope of the current study. To guide the scope of the revisions, I have listed these points below. We are committed to providing a fair and constructive peer-review process, so please feel free to contact me if you would like to discuss any of the referee comments further.

In particular, it would be essential to:

A) Edit your current text in your results and discussion section to more cautiously describe your data and whether this may be truly relevant in cells (or whether this could be further room for future studies) (Reviewers #1 and #2).

B) Assess localization of microtubule regulators with further controls (as requested by Reviewer #2).

C) Address concerns about potential data presentation (Reviewer #2) and citations of relevant literature (Reviewers #1 and #2).

D) All other referee concerns pertaining to strengthening existing data, providing controls, methodological details, clarifications and textual changes, should also be addressed.

E) Finally please pay close attention to our guidelines on statistical and methodological reporting (listed below) as failure to do so may delay the reconsideration of the revised manuscript. In particular please provide:

- a Supplementary Table including all numerical source data in Excel format, with data for different figures provided as different sheets within a single Excel file. The file should include source data giving rise to graphical representations and statistical descriptions in the paper and for all instances where

the figures present representative experiments of multiple independent repeats, the source data of all repeats should be provided.

We therefore invite you to take these points into account when revising the manuscript. In addition, when preparing the revision please:

- ensure that it conforms to our format instructions and publication policies (see below and www.nature.com/nature/authors/).
- provide a point-by-point rebuttal to the full referee reports verbatim, as provided at the end of this letter.
- provide the completed Editorial Policy Checklist (found here <https://www.nature.com/authors/policies/Policy.pdf>), and Reporting Summary (found here <https://www.nature.com/authors/policies/ReportingSummary.pdf>). This is essential for reconsideration of the manuscript and these documents will be available to editors and referees in the event of peer review. For more information see <http://www.nature.com/authors/policies/availability.html> or contact me.

Nature Cell Biology is committed to improving transparency in authorship. As part of our efforts in this direction, we are now requesting that all authors identified as 'corresponding author' on published papers create and link their Open Researcher and Contributor Identifier (ORCID) with their account on the Manuscript Tracking System (MTS), prior to acceptance. ORCID helps the scientific community achieve unambiguous attribution of all scholarly contributions. You can create and link your ORCID from the home page of the MTS by clicking on 'Modify my Springer Nature account'. For more information please visit www.springernature.com/orcid.

[Redacted]

We would like to receive the revision within four weeks. If submitted within this time period, reconsideration of the revised manuscript will not be affected by related studies published elsewhere, or accepted for publication in Nature Cell Biology in the meantime. We would be happy to consider a revision even after this timeframe, but in that case we will consider the published literature at the time of resubmission when assessing the file.

We hope that you will find our referees' comments, and editorial guidance helpful. Please do not hesitate to contact me if there is anything you would like to discuss.

Best wishes,

Daryl

Daryl Jason Verzosa David, PhD

Senior Editor, Nature Cell Biology
Nature Portfolio

Heidelberger Platz 3, 14197 Berlin, Germany
Email: daryl.david@nature.com
ORCID: <https://orcid.org/0000-0002-9253-4805>

Reviewers' Comments:

Reviewer #1:

Remarks to the Author:

Overall the authors have addressed my concerns with explanations, text changes and new experimental data. The manuscript is improved and remaining issues may be addressed by better presentation of some of the new data and by additional changes in text and discussion.

Below are my comments on the authors' responses and revised manuscript.

Authors: We respectfully disagree: it is currently not known whether gamma-TuRC is symmetric or asymmetric when it nucleates microtubules in cells.

Correct and confirming my point: the authors seem to assume that nucleation from imperfect templates or partial gamma-TuRCs *in vitro*, as shown in this manuscript, is relevant in cells – even though it is not known whether this occurs in cells.

Authors: As the reviewer indicates, the observed gamma-TuRC asymmetry has indeed been interpreted as a regulatory feature, but no actual proof for this concept has been provided until now. There is no doubt that *in vitro*, asymmetric and even partial gamma-TuRCs can nucleate microtubules (see, for example, Wieczorek et al., 2021, *J. Cell Biol.*, <https://doi.org/10.1083/jcb.202009146>).

I agree.

Authors: Furthermore, unlike GCP2 and GCP3, GCP4, 5 or 6 are not essential for viability of *Drosophila* (Verollet et al., 2006, *J. Cell Biol.*, <https://doi.org/10.1083/jcb.200511071>; Vogt et al., 2006, *Development*, <https://doi.org/10.1242/dev.02570>), fission yeast (Anders et al., 2006, *Mol. Biol. Cell*, <https://doi.org/10.1091/mbc.e05-11-1009>) or *Aspergillus* cells (Xiong and Oakley, 2009, *J. Cell Sci.*, <https://doi.org/10.1242/jcs.059196>). Cells depleted of GCP4, 5 and 6 do nucleate microtubules from gamma-tubulin spots localized to centrosomes, indicating that symmetric gamma-TuRCs may not be essential in cells (Cota et al., 2017, *J. Cell Sci.*, <https://doi.org/10.1242/jcs.195321>).

If anything, these data actually argue in favor of symmetric templates being used in cells. GCP4, 5

and 6 introduce most of the asymmetry in the gamma-TuRC structure. In their absence, as shown by work in yeast by the Schiebel and Agard labs as well as others, ring-shaped gamma-TuSC oligomers can promote nucleation and these are more symmetric when compared to gamma-TuRC.

Authors: Importantly, our approach of combining gamma-TuRC with CAMSAPs in the same assay provides the first direct readout probing the attachment between the gamma-TuRC and the microtubule it nucleates. This assay shows that the mechanisms of gamma-TuRC activation can be fundamentally different, and that these mechanisms have consequences for the subsequent destiny of nucleated microtubules. In fact, not all activator/nucleator-complexes have to work in the same way, by stably capping microtubule ends. We show that, unlike CDK5RAP2, CLASP2 activator/nucleator complex does not tightly cap the microtubule end and allows CAMSAP-binding and subsequent microtubule release. Therefore, depending on the gamma-TuRC activator, microtubule minus ends will either stay attached to gamma-TuRC that nucleated them or will detach and become stabilized by CAMSAPs, allowing the same gamma-TuRC to be reused for another round of microtubule nucleation.

Again, I like the idea, but there is no direct evidence that this occurs in cells. The newly added data quantifying CAMSAP stretches in cells lacking or overexpressing CDK5RAP2 provides at least some indirect evidence. Considering this new data, I suggest to more cautiously phrase some claims in results and discussion. See below.

Regarding the characterizations and quality controls of the gamma-TuRC preps, the authors have done a good job and now provide a thorough analysis. While variable and depending on the assay used, overall the data confirm my concern that purified "gamma-TuRC" is in fact a mix of gamma-TuRC and significant amounts of smaller, incomplete or sub-complexes.

Specific comments:

- 1) Considering the presence of significant amounts of incomplete/smaller complexes in the gamma-TuRC preps used in the study, this needs to be clearly stated. The statement "This fits with the fact that complete..." on page 5 needs to acknowledge the presence not only of gamma-TuRC (36%) but also of smaller complexes (37%), most likely gamma-TuSC, which is the second major species in human cell extract, as shown in numerous previous studies.
- 2) Since the authors acknowledge that only a fraction of the complexes in the "gamma-TuRC" prep are actually gamma-TuRCs (about 36%) and the rest are incomplete or sub-complexes and since this is crucial for interpreting the results, I strongly suggest that the text does not refer to immobilized gamma-TuRCs but rather more generally to "gamma-tubulin complexes" or similar.
- 3) P. 6 "This was likely because preincubation greatly increased the percentage of γ -TuRCs colocalizing with CDK5RAP2...". An alternative explanation that should be considered here would be that pre-incubation promotes formation of more complete rings from subcomplexes, as shown for CM1 containing adapters that can oligomerize gamma-TuSC in yeast (Agard and Schiebel labs).
- 4) P. 9 "Recent work has shown that also partial/incomplete γ -TuRCs can nucleate microtubules" – "in vitro" should be added to this sentence.
- 5) Discussion: "However, given the asymmetric γ -TuRC structure and the fact that not all γ -tubulin subunits within γ -TuRC are essential for microtubule nucleation, it is possible that some protofilaments

at the γ -TuRC-capped minus end will be unattached and acquire a flared conformation permissive for CAMSAP binding”.

I cannot follow the rationale here – how does the fact that not all gamma-TuRC subunits are essential support the idea that gamma-TuRC may not fully cap the minus-end? Please explain better and/or rephrase.

6) Apart from the presence of asymmetric gamma-TuRCs at MT minus-ends following nucleation in cells, which remains speculative, the discussion should also consider the possibility that in cells CAMSAPs could also preferentially bind and stabilize MTs that may be nucleated independently of gTuRCs. This possibility is still in agreement with the KO and CDK5RAP2 overexpression experiments, which may shift the percentage of gamma-TuRC-dependent vs independent nucleation. Indeed, Tsuchiya and Goshima (JCB, 2021) showed that even in the absence of gamma-tubulin, interphase microtubules still formed and CAMSAP, ch-TOG and CLASP1 were among the factors that were crucial for microtubule generation. This study should also be cited.

7) The new data with KO and CDK5RAP2 overexpression cell lines is a good addition, but needs to be presented better. Also, Fig. 5a,b and Ext. Fig 6a seem redundant.

Presentation issues:

- all figures and panels showing KO/GFP-CDK5RAP2 overexpression experiments: the magnified regions shown in black/white in the insets are supposed to show the GFP-CDK5RAP2 channel that is shown in red in the main image, yet they never match. This may be an exposure/thresholding issue, but it is confusing. They should look the same if they show the same signal.
- related to the above: there is clearly weak red signal all over the cells in the main images, even in the samples without GFP-CDK5RAP2 expression, but this seems to be absent in the magnified b/w images
- are the quantifications done in the dashed line boxes (if so, how are they chosen when there is no CDK5RAP2 signal?) or in whole cells? This should be explained.
- what is the expression level of GFP-CDK5RAP2 relative to the endogenous protein? A western blot would be useful to evaluate the level of overexpression.

Reviewer #2:

Remarks to the Author:

This a revised manuscript by Rai, Jiang, Akhmanova and colleagues. The manuscript presentation has improved. The additional analyses have improved the description of the enormous amount of experiments presented in this manuscript. The authors have added in vivo studies to visualize the impact of CDK5RP regulation of minus end microtubule regulation by CAMSAPs. However there remains some concerns remain that should be addressed before acceptance for publication:

1)The authors present additional characterization of the gamma tubulin ring complex (g-TURC). Although these additional data and analyses have improved the characterization of their g-TURC assemblies, these additional analyses highlight the severe heterogeneity of the purified g-TURC

assemblies used in this study in comparison to work published by other groups by other groups using recombinant over-expressed g-TURC. Although incomplete g-TURC maybe remain active and can nucleate microtubules, they will likely be responsible for the incomplete attachment of newly nucleated microtubules, which impacts some important conclusions of the paper-- not all microtubule protofilaments are attached to g-TURC during nucleation and that that CAMSAP binding promotes the release of microtubules by interacting with these flared microtubules minus ends. I believe the authors should put all their data into consideration and quantitate these events and compare them to the proportion of intact G-TURC assemblies (11%). This is also consistent with the fact that very few g-TURC complexes nucleate microtubules even under conditions where MT regulators such as CLASP, chTOG and EB3 enhance microtubule polymerization. Will more purified g-TURC assemblies allow less CAMSAP binding and release of minus ends.

2)The authors present a new approach to study the initial localization of the microtubule regulators, CDK5RAP2, CLASP, chTOG, and EB3 with respect to the g-TURC assemblies during microtubule nucleation (Figure 1). Although this adds an important dimension to the work, the studies presented lack any form of controls. Control studies should be performed where every aspect of the process should be controlled to understand the source of the co-localization. For example, are microtubule nucleation/polymerization events emerging from G-TURC necessary for the initial localization of CLASP, chTOG, EB3. How would this co-localization change if no microtubule polymerization was allowed (-GTP)? if soluble tubulin was omitted from the assays, will the above regulators bind g-TURC assemblies? Also controls showing the lack of aggregation/association of these regulators on treated glass surfaces in the absence of these g-TURC or polymerizing microtubules are crucial. These controls will be crucial to validate the roles of microtubule polymerization, tubulin binding to g-TURC or regulators in the process of regulating microtubule nucleation.

3) There are problems with the kymograph /time-lapse-still presentations and videos in the manuscript. Although improved, there are still many kymographs that are messy and despite guide images, it is still quite hard to see what the authors are presenting. There are many multi-image stills/kymographs that are not being presented in single colors (even in the supplementary information) The impact of the manuscript will dramatically improve if all the kymographs, time-lapse stills are presented more thoroughly and clearly.

4)Considering the enormous amount of data in this manuscript in its earlier version, the authors added even more studies of the impact of CDK5RAP2 on CAMSAP localization at microtubule ends in vivo. I really don't think presenting these studies is at all necessary. There are many observations presented in the paper which are superficially analyzed due to the lack of space. For example, the totality of studies in figure 1 can be an independent manuscript if studied in more depth and analyzed carefully. The authors choice to add even more in vivo studies to the manuscript is understood in relation to the remainder of the manuscript, but I am not quite sure that the additional data adds a large dimension to the work. The space used by these studies takes focus away from the careful analyses of the complex in vitro reconstitution studies.

Reviewer #3:
None

GUIDELINES FOR SUBMISSION OF NATURE CELL BIOLOGY ARTICLES

ARTICLE FORMAT

ABSTRACT – should not exceed 150 words and should be unreferenced. This paragraph is the most visible part of the paper and should briefly outline the background and rationale for the work, and accurately summarize the main results and conclusions. Key genes, proteins and organisms should be specified to ensure discoverability of the paper in online searches.

TEXT – the main text consists of the Introduction, Results, and Discussion sections and must not exceed 3500 words including the abstract. The Introduction should expand on the background relating to the work. The Results should be divided in subsections with subheadings, and should provide a concise and accurate description of the experimental findings. The Discussion should expand on the findings and their implications. All relevant primary literature should be cited, in particular when discussing the background and specific findings.

FINANCIAL AND NON-FINANCIAL COMPETING INTERESTS – the authors must include one of three declarations: (1) that they have no financial and non-financial competing interests; (2) that they have financial and non-financial competing interests; or (3) that they decline to respond, after the Author Contributions section. This statement will be published with the article, and in cases where financial and non-financial competing interests are declared, these will be itemized in a web supplement to the

article. For further details please see <https://www.nature.com/licenceforms/nrg/competing-interests.pdf>.

REFERENCES – are limited to a total of 70 in the main text and Methods combined,. They must be numbered sequentially as they appear in the main text, tables and figure legends and Methods and must follow the precise style of Nature Cell Biology references. References only cited in the Methods should be numbered consecutively following the last reference cited in the main text. References only associated with Supplementary Information (e.g. in supplementary legends) do not count toward the total reference limit and do not need to be cited in numerical continuity with references in the main text. Only published papers can be cited, and each publication cited should be included in the numbered reference list, which should include the manuscript titles. Footnotes are not permitted.

Methods should be written concisely, but should contain all elements necessary to allow interpretation and replication of the results. As a guideline, Methods sections typically do not exceed 3,000 words. The Methods should be divided into subsections listing reagents and techniques. When citing previous methods, accurate references should be provided and any alterations should be noted. Information must be provided about: antibody dilutions, company names, catalogue numbers and clone numbers for monoclonal antibodies; sequences of RNAi and cDNA probes/primers or company names and catalogue numbers if reagents are commercial; cell line names, sources and information on cell line identity and authentication. Animal studies and experiments involving human subjects must be reported in detail, identifying the committees approving the protocols. For studies involving human subjects/samples, a statement must be included confirming that informed consent was obtained. Statistical analyses and information on the reproducibility of experimental results should be provided in a section titled "Statistics and Reproducibility".

All Nature Cell Biology manuscripts submitted on or after March 21 2016, must include a Data availability statement as a separate section after Methods but before references, under the heading "Data Availability". For Springer Nature policies on data availability see <http://www.nature.com/authors/policies/availability.html>; for more information on this particular policy see <http://www.nature.com/authors/policies/data/data-availability-statements-data-citations.pdf>. The Data availability statement should include:

- Accession codes for primary datasets (generated during the study under consideration and designated as "primary accessions") and secondary datasets (published datasets reanalysed during the study under consideration, designated as "referenced accessions"). For primary accessions data should be made public to coincide with publication of the manuscript. A list of data types for which submission to community-endorsed public repositories is mandated (including sequence, structure, microarray, deep sequencing data) can be found here <http://www.nature.com/authors/policies/availability.html#data>.
- Unique identifiers (accession codes, DOIs or other unique persistent identifier) and hyperlinks for datasets deposited in an approved repository, but for which data deposition is not mandated (see here for details <http://www.nature.com/sdata/data-policies/repositories>).

- At a minimum, please include a statement confirming that all relevant data are available from the authors, and/or are included with the manuscript (e.g. as source data or supplementary information), listing which data are included (e.g. by figure panels and data types) and mentioning any restrictions on availability.
- If a dataset has a Digital Object Identifier (DOI) as its unique identifier, we strongly encourage including this in the Reference list and citing the dataset in the Methods.

We recommend that you upload the step-by-step protocols used in this manuscript to the Protocol Exchange. More details can found at www.nature.com/protocolexchange/about.

DISPLAY ITEMS – main display items are limited to 6-8 main figures and/or main tables. For Supplementary Information see below.

FIGURES – Colour figure publication costs \$395 per colour figure. All panels of a multi-panel figure must be logically connected and arranged as they would appear in the final version. Unnecessary figures and figure panels should be avoided (e.g. data presented in small tables could be stated briefly in the text instead).

All imaging data should be accompanied by scale bars, which should be defined in the legend. Cropped images of gels/blots are acceptable, but need to be accompanied by size markers, and to retain visible background signal within the linear range (i.e. should not be saturated). The boundaries of panels with low background have to be demarked with black lines. Splicing of panels should only be considered if unavoidable, and must be clearly marked on the figure, and noted in the legend with a statement on whether the samples were obtained and processed simultaneously. Quantitative comparisons between samples on different gels/blots are discouraged; if this is unavoidable, it has to be performed for samples derived from the same experiment with gels/blots were processed in parallel, which needs to be stated in the legend.

- For line art, graphs, charts and schematics we prefer Adobe Illustrator (.AI), Encapsulated PostScript (.EPS) or Portable Document Format (.PDF). Files should be saved or exported as such directly from the application in which they were made, to allow us to restyle them according to our journal house style.
- We accept PowerPoint (.PPT) files if they are fully editable. However, please refrain from adding

PowerPoint graphical effects to objects, as this results in them outputting poor quality raster art. Text used for PowerPoint figures should be Helvetica (preferred) or Arial.

Regardless of format, all figures must be vector graphic compatible files, not supplied in a flattened raster/bitmap graphics format, but should be fully editable, allowing us to highlight/copy/paste all text and move individual parts of the figures (i.e. arrows, lines, x and y axes, graphs, tick marks, scale bars etc). The only parts of the figure that should be in pixel raster/bitmap format are photographic images or 3D rendered graphics/complex technical illustrations.

Unprocessed scans of all key data generated through electrophoretic separation techniques need to be

presented in a supplementary figure that should be labeled and numbered as the final supplementary figure, and should be mentioned in every relevant figure legend. This figure does not count towards the total number of figures and is the only figure that can be displayed over multiple pages, but should be provided as a single file, in PDF or TIFF format. Data in this figure can be displayed in a relatively informal style, but size markers and the figures panels corresponding to the presented data must be indicated.

The total number of Supplementary Figures (not including the “unprocessed scans” Supplementary Figure) should not exceed the number of main display items (figures and/or tables (see our Guide to Authors and March 2012 editorial <http://www.nature.com/ncb/authors/submit/index.html#suppinfo>; <http://www.nature.com/ncb/journal/v14/n3/index.html#ed>). No restrictions apply to Supplementary Tables or Videos, but we advise authors to be selective in including supplemental data.

GUIDELINES FOR EXPERIMENTAL AND STATISTICAL REPORTING

REPORTING REQUIREMENTS – To improve the quality of methods and statistics reporting in our papers we have recently revised the reporting checklist we introduced in 2013. We are now asking all life sciences authors to complete two items: an Editorial Policy Checklist (found here <https://www.nature.com/authors/policies/Policy.pdf>) that verifies compliance with all required editorial policies and a Reporting Summary (found here <https://www.nature.com/authors/policies/ReportingSummary.pdf>) that collects information on experimental design and reagents. These documents are available to referees to aid the evaluation of the manuscript. Please note that these forms are dynamic ‘smart pdfs’ and must therefore be downloaded and completed in Adobe Reader. We will then flatten them for ease of use by the reviewers. If you would like to reference the guidance text as you complete the template, please access these flattened versions at <http://www.nature.com/authors/policies/availability.html>.

Information on how many times each experiment was repeated independently with similar results needs to be provided in the legends and/or Methods for all experiments, and in particular wherever

representative experiments are shown.

Author Rebuttal, first revision:

Point-by-point Response to the Reviewers' comments

Editorial comments:

A) Edit your current text in your results and discussion section to more cautiously describe your data and whether this may be truly relevant in cells (or whether this could be further room for future studies) (Reviewers #1 and #2).

We have edited the results and discussion to more cautiously describe our data and indicated that there is room for further studies of the described mechanism of generation of stable microtubule minus ends in cells.

B) Assess localization of microtubule regulators with further controls (as requested by Reviewer #2).

To address this comment of reviewer #2, we have added additional controls of colocalization of microtubule regulators with gamma-tubulin-containing complexes and highlighted relevant controls that were already included in the manuscript.

C) Address concerns about potential data presentation (Reviewer #2) and citations of relevant literature (Reviewers #1 and #2).

We have done our best to improve data presentation (particularly, the figures) and included citations suggested by the reviewers.

D) All other referee concerns pertaining to strengthening existing data, providing controls, methodological details, clarifications and textual changes, should also be addressed.

We have addressed these comments, for example, by adding a new Western blot illustrating the level of overexpression of CDK5RAP2 in the studied cell lines (new Extended data Fig. 6a). We have also added the details and made textual changes suggested by the reviewers; for example, we substituted gamma-TuRC for gamma-TuC (gamma-tubulin-containing complexes) in all situations where we refer to our experiments and the statement may apply to incomplete subcomplexes.

E) Finally please pay close attention to our guidelines on statistical and methodological reporting (listed below) as failure to do so may delay the reconsideration of the revised manuscript. In particular please provide:

Done.

- a Supplementary Table including all numerical source data in Excel format, with data for different figures provided as different sheets within a single Excel file. The file should include

source data giving rise to graphical representations and statistical descriptions in the paper and for all instances where the figures present representative experiments of multiple independent repeats, the source data of all repeats should be provided.

Done.

Reviewer #1 (Remarks to the Author):

Overall the authors have addressed my concerns with explanations, text changes and new experimental data. The manuscript is improved and remaining issues may be addressed by better presentation of some of the new data and by additional changes in text and discussion.

Below are my comments on the authors' responses and revised manuscript.

Authors: We respectfully disagree: it is currently not known whether gamma-TuRC is symmetric or asymmetric when it nucleates microtubules in cells.

Correct and confirming my point: the authors seem to assume that nucleation from imperfect templates or partial gamma-TuRCs in vitro, as shown in this manuscript, is relevant in cells – even though it is not known whether this occurs in cells.

We fully agree with the reviewer that the relevance of asymmetric or partial gamma-TuRCs for microtubule nucleation in cells is currently unclear, and therefore, we do not make any conclusions on this issue, which requires further study. Further, we would like to clarify that we do show microtubule nucleation and subsequential release upon CAMSAP-binding from complete gamma-TuRCs (Ext. data Fig. 4i). This was more prominently observed in the presence of CLASP2 in the nucleator/activator complex (Ext. data Fig. 4k).

Authors: As the reviewer indicates, the observed gamma-TuRC asymmetry has indeed been interpreted as a regulatory feature, but no actual proof for this concept has been provided until now. There is no doubt that in vitro, asymmetric and even partial gamma-TuRCs can nucleate microtubules (see, for example, Wieczorek et al., 2021, J. Cell Biol., <https://doi.org/10.1083/jcb.202009146>).

I agree.

Authors: Furthermore, unlike GCP2 and GCP3, GCP4, 5 or 6 are not essential for viability of Drosophila (Verollet et al., 2006, J. Cell Biol., <https://doi.org/10.1083/jcb.200511071>; Vogt et al., 2006, Development, <https://doi.org/10.1242/dev.02570>), fission yeast (Anders et al., 2006, Mol. Biol. Cell, <https://doi.org/10.1091/mbc.e05-11-1009>) or Aspergillus cells (Xiong and Oakley, 2009, J. Cell Sci., <https://doi.org/10.1242/jcs.059196>). Cells depleted of GCP4, 5 and 6 do

nucleate microtubules from gamma-tubulin spots localized to centrosomes, indicating that symmetric gamma-TuRCs may not be essential in cells (Cota et al., 2017, J. Cell Sci., <https://doi.org/10.1242/jcs.195321>).

If anything, these data actually argue in favor of symmetric templates being used in cells. GCP4, 5 and 6 introduce most of the asymmetry in the gamma-TuRC structure. In their absence, as shown by work in yeast by the Schiebel and Agard labs as well as others, ring-shaped gamma-TuSC oligomers can promote nucleation and these are more symmetric when compared to gamma-TuRC.

It is indeed possible that in the absence of GCP4, 5 or 6 in cells, more symmetrical gamma-TuSC structures are formed, but this has never been established in animal cells, so it is also possible that in such cells, partial complexes form and nucleate microtubules. Furthermore, even if ring-shaped gamma-TuSC oligomers are formed in animal cells, they would still not fully match microtubule geometry and would require an activation mechanism, as reported by the Agard lab (Kollman 2015, NSMB). Since our paper does not shed light on this issue, we do not discuss the implications of the loss of GCP subunits.

Authors: Importantly, our approach of combining gamma-TuRC with CAMSAPs in the same assay provides the first direct readout probing the attachment between the gamma-TuRC and the microtubule it nucleates. This assay shows that the mechanisms of gamma-TuRC activation can be fundamentally different, and that these mechanisms have consequences for the subsequent destiny of nucleated microtubules. In fact, not all activator/nucleator-complexes have to work in the same way, by stably capping microtubule ends. We show that, unlike CDK5RAP2, CLASP2 activator/nucleator complex does not tightly cap the microtubule end and allows CAMSAP-binding and subsequent microtubule release. Therefore, depending on the gamma-TuRC activator, microtubule minus ends will either stay attached to gamma-TuRC that nucleated them or will detach and become stabilized by CAMSAPs, allowing the same gamma-TuRC to be reused for another round of microtubule nucleation.

Again, I like the idea, but there is no direct evidence that this occurs in cells. The newly added data quantifying CAMSAP stretches in cells lacking or overexpressing CDK5RAP2 provides at least some indirect evidence. Considering this new data, I suggest to more cautiously phrase some claims in results and discussion. See below.

We agree and, as suggested by the reviewer below, we have now stated that we cannot exclude that CAMSAP2-stabilized microtubule minus ends are generated after gamma-TuRC independent microtubule nucleation.

Regarding the characterizations and quality controls of the gamma-TuRC preps, the authors

have done a good job and now provide a thorough analysis. While variable and depending on the assay used, overall the data confirm my concern that purified “gamma-TuRC” is in fact a mix of gamma-TuRC and significant amounts of smaller, incomplete or sub-complexes.

Specific comments:

1) Considering the presence of significant amounts of incomplete/smaller complexes in the gamma-TuRC preps used in the study, this needs to be clearly stated. The statement “This fits with the fact that complete...” on page 5 needs to acknowledge the presence not only of gamma-TuRC (36%) but also of smaller complexes (37%), most likely gamma-TuSC, which is the second major species in human cell extract, as shown in numerous previous studies.

We agree and we have elaborated on gamma-TuSC population on p.5-6 of the revised manuscript: “...we also characterized the fluorescence intensity of purified complexes and compared it to that of single GFP molecules and GFP-EB3, which is known to be a dimer³⁵. In these measurements, GFP-EB3 was ~1.7x brighter than GFP, whereas GCP3-GFP-containing fluorescent puncta displayed two peaks, with intensities corresponding to 1-2 GFPs (37%) and 4-5 GFPs (36 %) (Fig. 1a and Extended data Fig. 1k). The first peak most likely includes γ -tubulin small complex (γ -TuSC) population that is generally present in γ -TuRC preparations and would be expected to contain one GCP3-GFP subunit, whereas the second peak confirms the presence of complete γ -TuRCs that are expected to contain five GCP3-GFP subunits”.

2) Since the authors acknowledge that only a fraction of the complexes in the “gamma-TuRC” prep are actually gamma-TuRCs (about 36%) and the rest are incomplete or sub-complexes and since this is crucial for interpreting the results, I strongly suggest that the text does not refer to immobilized gamma-TuRCs but rather more generally to “gamma-tubulin complexes” or similar.

We agree that “gamma-tubulin complexes” is an excellent suggestion and have incorporated it throughout the manuscript.

3) P. 6 “This was likely because preincubation greatly increased the percentage of γ -TuRCs colocalizing with CDK5RAP2...”. An alternative explanation that should be considered here would be that pre-incubation promotes formation of more complete rings from subcomplexes, as shown for CM1 containing adapters that can oligomerize gamma-TuSC in yeast (Agard and Schiebel labs).

In our in vitro reconstitutions, we did not observe any significant amounts of complexes that contained seven gamma-TuSCs (Ext Fig. 1k and Ext Fig. 4c). Furthermore, our results suggest that preincubation with CDK5RAP2 did not induce oligomerization of smaller subcomplexes into larger ones. Rather, CDK5RAP2 preferentially activated full gamma-TuRCs. We have elaborated on this on page 10, 2nd paragraph: “It revealed that CDK5RAP2 and CLASP2 by themselves had

no noticeable effect on the size distribution of the γ -TuCs, indicating that they did not induce oligomerization of γ -TuSCs into γ -TuRC (size distributions are similar in Extended data Fig. 4c-e).“

4) P. 9 *“Recent work has shown that also partial/incomplete γ -TuRCs can nucleate microtubules” – “in vitro” should be added to this sentence.*

We have added “in vitro” to the sentence on page 9.

5) Discussion: *“However, given the asymmetric γ -TuRC structure and the fact that not all γ -tubulin subunits within γ -TuRC are essential for microtubule nucleation, it is possible that some protofilaments at the γ -TuRC-capped minus end will be unattached and acquire a flared conformation permissive for CAMSAP binding”.*

I cannot follow the rationale here – how does the fact that not all gamma-TuRC subunits are essential support the idea that gamma-TuRC may not fully cap the minus-end? Please explain better and/or rephrase.

We meant that if not all gamma-TuRC subunits are essential, one could argue that partial gamma-TuRCs that cannot fully cap the minus end could still be nucleating microtubules. However, we fully agree with reviewer’s comments that alternative scenarios are possible in cells (e.g., larger gamma-TuSC assemblies might take over), so we removed this point from the discussion on page 15, as our paper provides no data on this subject.

6) *Apart from the presence of asymmetric gamma-TuRCs at MT minus-ends following nucleation in cells, which remains speculative, the discussion should also consider the possibility that in cells CAMSAPs could also preferentially bind and stabilize MTs that may be nucleated independently of γ -TuRCs. This possibility is still in agreement with the KO and CDK5RAP2 overexpression experiments, which may shift the percentage of gamma-TuRC-dependent vs independent nucleation. Indeed, Tsuchiya and Goshima (JCB, 2021) showed that even in the absence of gamma-tubulin, interphase microtubules still formed and CAMSAP, ch-TOG and CLASP1 were among the factors that were crucial for microtubule generation. This study should also be cited.*

We thank the reviewer for suggesting an excellent point for discussion. We have added this point on page 17 in 1st paragraph. “Still, we cannot exclude that CAMSAP-stabilized minus ends are generated in CDK5RAP2 knockout cells in a γ -tubulin-independent manner, since formation of microtubules dependent on CLASP1, chTOG and CAMSAPs has been described in γ -tubulin-depleted cells ⁶⁸, and future studies would be needed to further dissect different microtubule nucleation pathways.”

7) *The new data with KO and CDK5RAP2 overexpression cell lines is a good addition, but needs to be presented better. Also, Fig. 5a,b and Ext. Fig 6a seem redundant.*

We have altered the figures to make them more clear. We would like to clarify that the quantification shown in Fig. 5b presents values per “square micron area of the cell”, whereas Ext. data Fig. 6a (Ext. data Fig. 6b in the revised version) is the quantification of the cell area, which was used to generate the plot in Fig. 5b. The values “per cell” were shown in Ext. data Fig. 6b (Ext. data Fig. 6c in the revised version) in order to provide a complete overview of the data. Since CDK5RAP2/Myomegalin knockout cells were somewhat smaller in size, calculating the number of CAMSAP2 stretches per square micron area of the cell provides a better quantification, which is included in the main figure.

Presentation issues:

- all figures and panels showing KO/GFP-CDK5RAP2 overexpression experiments: the magnified regions shown in black/white in the insets are supposed to show the GFP-CDK5RAP2 channel that is shown in red in the main image, yet they never match. This may be an exposure/thresholding issue, but it is confusing. They should look the same if they show the same signal.

We agree that the combination of inverted grayscale and red color in merged images was confusing. Therefore, we have changed the color scheme to cyan and red for Fig. 5 and Ext. data Fig. 6. Now, the insets match the signal in the merge images.

- related to the above: there is clearly weak red signal all over the cells in the main images, even in the samples without GFP-CDK5RAP2 expression, but this seems to be absent in the magnified b/w images

This was also the issue of choosing a red color overlay with inverted grayscale. In the new version, CAMSAP2, EB1 and tubulin signals are shown in cyan, and this issue was resolved.

- are the quantifications done in the dashed line boxes (if so, how are they chosen when there is no CDK5RAP2 signal?) or in whole cells? This should be explained.

The dashed boxes mark regions of interest for the insets, and the quantifications were performed not in the dashed boxes but in whole cells. The quantification method was explained in the Methods section previously, and now we have created a separate subheading “Analysis of CAMSAP2 stretches, EB1 and tubulin intensity in fixed cells” on page 33, so that the procedure used for the data analysis in Fig. 5 and Ext. data Fig. 6 would be easier to find.

- what is the expression level of GFP-CDK5RAP2 relative to the endogenous protein? A western blot would be useful to evaluate the level of overexpression.

A Western blot has been added to the revised version (new Ext. Data Fig. 6a). GFP-CDK5RAP2 cell lines in the wild type and AKAP450 background were clonal (Chen et al., eLife 2022, PMID: 35787744), whereas GFP-CDK5RAP2-expressing triple CDK5RAP2/Myomegalin/AKAP450 knockout cells were a mixed cell population with respect to the GFP-CDK5RAP2 transgene, and therefore, the signal was weaker, because cells with very low expression of the transgene were present in the population. Cells with similar GFP intensity, with clearly visible GFP signal at the centrosome (or the centrosome and the Golgi in the wild type cells; Golgi signal of GFP-CDK5RAP2 is absent in AKAP450 knockouts) were selected for the analysis in all three cell lines. In the clonal cell lines, GFP-CDK5RAP2 overexpression was estimated to be 6-8-fold increase compared to the endogenous protein.

Reviewer #2 (Remarks to the Author):

This a revised manuscript by Rai, Jiang, Akhmanova and colleagues. The manuscript presentation has improved. The additional analyses have improved the description of the enormous amount of experiments presented in this manuscript. The authors have added in vivo studies to visualize the impact of CDK5RP regulation of minus end microtubule regulation by CAMSAPs. However there remains some concerns remain that should be addressed before acceptance for publication:

1)The authors present additional characterization of the gamma tubulin ring complex (g-TURC). Although these additional data and analyses have improved the characterization of their g-TURC assemblies, these additional analyses highlight the severe heterogeneity of the purified g-TURC assemblies used in this study in comparison to work published by other groups by other groups using recombinant over-expressed g-TURC. Although incomplete g-TURC maybe remain active and can nucleate microtubules, they will likely be responsible for the incomplete attachment of newly nucleated microtubules, which impacts some important conclusions of the paper-- not all microtubule protofilaments are attached to g-TURC during nucleation and that that CAMSAP binding promotes the release of microtubules by interacting with these flared microtubules minus ends. I believe the authors should put all their data into consideration and quantitate these events and compare them to the proportion of intact G-TURC assemblies (11%). This is also consistent with the fact that very few g-TURC complexes nucleate microtubules even under conditions where MT regulators such as CLASP, chTOG and EB3 enhance microtubule polymerization. Will more purified g-TURC assemblies allow less CAMSAP binding and release of minus ends.

The estimation of 11% intact gamma-TuRC assemblies comes from mass photometry data and includes non-fluorescent contaminants, which are not relevant for any of our quantifications because only microtubule nucleation events occurring at GFP-containing puncta were taken into account throughout the paper. With the addition of regulators, especially CDK5RAP2 and CLASP2, we see an increase in the microtubule nucleation by 20-fold, or up to 35% efficiency when GFP-containing complexes are quantified, which is very significant. We agree that not all

of these active complexes are complete gamma-TuRCs. However, a significant fraction of active complexes are indeed complete gamma-TuRC (based on data shown in Ext. Data Fig.4f, they can be estimated to be ~30% in control, ~56% in CDK5RAP2 and ~49% in CLASP2,). We discuss these subpopulations on page 10, 2nd paragraph. Since events of interest like CAMSAP-binding and microtubule release can only happen when the gamma-tubulin complex is active, comparing the fraction of intact gamma-TuRC within the active population makes more sense.

We further note that the heterogeneity of gamma-tubulin containing complexes in our in vitro microtubule nucleation assays cannot be directly compared to the majority of other studies, because for purification of gamma-TuRC, we used a homozygous GCP3-GFP cell line, while most other published studies used other approaches to purify gamma-TuRC, which do not allow full labeling of a specific gamma-TuSC subunit, and the majority of these studies did not quantify microtubule nucleation events from single fluorescent complexes.

2)The authors present a new approach to study the initial localization of the microtubule regulators, CDK5RAP2, CLASP, chTOG, and EB3 with respect to the g-TURC assemblies during microtubule nucleation (Figure 1). Although this adds an important dimension to the work, the studies presented lack any form of controls. Control studies should be performed where every aspect of the process should be controlled to understand the source of the co-localization. For example, are microtubule nucleation/polymerization events emerging from G-TURC necessary for the initial localization of CLASP, chTOG, EB3. How would this co-localization change if no microtubule polymerization was allowed (-GTP)? if soluble tubulin was omitted from the assays, will the above regulators bind g-TURC assemblies? Also controls showing the lack of aggregation/association of these regulators on treated glass surfaces in the absence of these g-TURC or polymerizing microtubules are crucial. These controls will be crucial to validate the roles of microtubule polymerization, tubulin binding to g-TURC or regulators in the process of regulating microtubule nucleation.

We note that some of the controls suggested by the reviewer regarding the effect of microtubule nucleation/polymerization on initial colocalization of regulators with gamma-tubulin-containing complexes were already included in the manuscript (see conditions without soluble tubulin, to prevent microtubule polymerization, in Fig. 1i,j and Ext. data Fig. 2d). We have now also included in the same panels a control for the effect of protein aggregation on the glass surface by comparing the initial colocalization of regulators premixed with purified GFP in the absence of microtubule polymerization (Fig. 1i,j and Ext. data Fig. 2d). These controls show hardly any colocalization of CDK5RAP2, CLASP2 or chTOG with GFP alone, and very significant colocalization of CDK5RAP2 (38% and 30%), CLASP2 (72%) and chTOG (6% and 28%) with and without free tubulin and thus microtubule polymerization. These data demonstrate that all the nucleation-promoting factors studied here can specifically interact with gamma-tubulin containing protein complexes in a tubulin- and microtubule polymerization independent manner.

3) There are problems with the kymograph /time-lapse-still presentations and videos in the manuscript. Although improved, there are still many kymographs that are messy and despite guide images, it is still quite hard to see what the authors are presenting. There are many multi-image stills/kymographs that are not being presented in single colors (even in the supplementary information) The impact of the manuscript will dramatically improve if all the kymographs, time-lapse stills are presented more thoroughly and clearly.

We have improved the kymographs and still images by enlarging them, adding individual color channels as insets, adjusting the signal contrast, improving schematic images and color overlays in Fig. 1, 2, 3, 4, 5 and Extended data Fig. 2, 3,6. We have now also annotated the videos better.

4)Considering the enormous amount of data in this manuscript in its earlier version, the authors added even more studies of the impact of CDK5RAP2 on CAMSAP localization at microtubule ends in vivo. I really don't think presenting these studies is at all necessary. There are many observations presented in the paper which are superficially analyzed due to the lack of space. For example, the totality of studies in figure 1 can be an independent manuscript if studied in more depth and analyzed carefully. The authors choice to add even more in vivo studies to the manuscript is understood in relation to the remainder of the manuscript, but I am not quite sure that the additional data adds a large dimension to the work. The space used by these studies takes focus away from the careful analyses of the complex in vitro reconstitution studies.

We think that in vivo data are an important addition to the manuscript as they provide physiological relevance to the study by showing that CDK5RAP2 and its paralog Myomegalin are not at all essential for gamma-TuRC-dependent microtubule nucleation in cells but affect the balance of centrosomal and non-centrosomal, CAMSAP2-stabilized microtubules, which fits well with the conclusions of our in vitro experiments. We note that these data addressed an important comment of Reviewer #1, who found them a good addition to the manuscript.

Reviewer #3 (Remarks to the Author):

None

Decision Letter, second revision:

Our ref: NCB-A49143B

22nd November 2023

Dear Dr. Akhmanova,

Thank you for submitting your revised manuscript "CAMSAP-driven microtubule release from γ -TuRC and its regulation by nucleation-promoting factors" (NCB-A49143B). It has now been seen by the original referees and their comments are below. The reviewers find that the paper has improved in revision, and therefore we'll be happy in principle to publish it in Nature Cell Biology, pending minor revisions to satisfy the referees' final requests and to comply with our editorial and formatting guidelines.

The current version of your manuscript is in a PDF format, so please email us a copy of the file in an editable format (Microsoft Word or LaTeX)-- we can not proceed with PDFs at this stage.

Thank you again for your interest in Nature Cell Biology Please do not hesitate to contact me if you have any questions.

Sincerely,

Daryl

Daryl Jason Verzosa David, PhD

Senior Editor, Nature Cell Biology
Nature Portfolio

Heidelberger Platz 3, 14197 Berlin, Germany
Email: daryl.david@nature.com
ORCID: <https://orcid.org/0000-0002-9253-4805>

Reviewer #1 (Remarks to the Author):

The authors have further improved the data presentation and description and have addressed all of my remaining concerns.

Reviewer #2 (Remarks to the Author):

The revised manuscript has addressed all the reviewers concerns and is now ready for publication.

Decision Letter, final checks:

Our ref: NCB-A49143B

5th December 2023

Dear Dr. Akhmanova,

Thank you for your patience as we've prepared the guidelines for final submission of your Nature Cell Biology manuscript, "CAMSAP-driven microtubule release from γ -TuRC and its regulation by nucleation-promoting factors" (NCB-A49143B). Please carefully follow the step-by-step instructions provided in the attached file, and add a response in each row of the table to indicate the changes that you have made. Please also check and comment on any additional marked-up edits we have proposed within the text. Ensuring that each point is addressed will help to ensure that your revised manuscript can be swiftly handed over to our production team.

In recognition of the time and expertise our reviewers provide to Nature Cell Biology's editorial process, we would like to formally acknowledge their contribution to the external peer review of your manuscript entitled "CAMSAP-driven microtubule release from γ -TuRC and its regulation by nucleation-promoting factors". For those reviewers who give their assent, we will be publishing their names alongside the published article.

Nature Cell Biology offers a Transparent Peer Review option for new original research manuscripts submitted after December 1st, 2019. As part of this initiative, we encourage our authors to support increased transparency into the peer review process by agreeing to have the reviewer comments, author rebuttal letters, and editorial decision letters published as a Supplementary item. When you submit your final files please clearly state in your cover letter whether or not you would like to participate in this initiative. Please note that failure to state your preference will result in delays in accepting your manuscript for publication.

Cover suggestions

COVER ARTWORK: We welcome submissions of artwork for consideration for our cover. For more information, please see our guide for cover artwork.

Nature Cell Biology has now transitioned to a unified Rights Collection system which will allow our Author Services team to quickly and easily collect the rights and permissions required to publish your work. Approximately 10 days after your paper is formally accepted, you will receive an email in providing you with a link to complete the grant of rights. If your paper is eligible for Open Access, our Author Services team will also be in touch regarding any additional information that may be required to arrange payment for your article.

Please note that *Nature Cell Biology* is a Transformative Journal (TJ). Authors may publish their research with us through the traditional subscription access route or make their paper immediately open access through payment of an article-processing charge (APC). Authors will not be required to make a final decision about access to their article until it has been accepted. Find out more about Transformative Journals

Please use the following link for uploading these materials:
[Redacted]

Best regards,

Kendra Donahue
Staff
Nature Cell Biology

On behalf of

Daryl Jason Verzosa David, PhD

Senior Editor, Nature Cell Biology
Nature Portfolio

Heidelberger Platz 3, 14197 Berlin, Germany
Email: daryl.david@nature.com
ORCID: <https://orcid.org/0000-0002-9253-4805>

Reviewer #1:

Remarks to the Author:

The authors have further improved the data presentation and description and have addressed all of my remaining concerns.

Reviewer #2:

Remarks to the Author:

The revised manuscript has addressed all the reviewers concerns and is now ready for publication.

Final Decision Letter:

Dear Dr Akhmanova,

I am pleased to inform you that your manuscript, "CAMSAPs and nucleation-promoting factors control microtubule release from γ -TuRC", has now been accepted for publication in Nature Cell Biology.

Please note that *Nature Cell Biology* is a Transformative Journal (TJ). Authors may publish their research with us through the traditional subscription access route or make their paper immediately open access through payment of an article-processing charge (APC). Authors will not be required to make a final decision about access to their article until it has been accepted. Find out more about Transformative Journals

If your paper includes color figures, please be aware that in order to help cover some of the additional cost of four-color reproduction, Nature Portfolio charges our authors a fee for the printing of their color

figures. Please contact our offices for exact pricing and details.

If you have not already done so, we strongly recommend that you upload the step-by-step protocols used in this manuscript to the Protocol Exchange (www.nature.com/protocolexchange), an open online resource established by Nature Protocols that allows researchers to share their detailed experimental know-how. All uploaded protocols are made freely available, assigned DOIs for ease of citation and are fully searchable through nature.com. Protocols and Nature Portfolio journal papers in which they are used can be linked to one another, and this link is clearly and prominently visible in the online versions of both papers. Authors who performed the specific experiments can act as primary authors for the Protocol as they will be best placed to share the methodology details, but the Corresponding Author of the present research paper should be included as one of the authors. By uploading your Protocols to Protocol Exchange, you are enabling researchers to more readily reproduce or adapt the methodology you use, as well as increasing the visibility of your protocols and papers. You can also establish a dedicated page to collect your lab Protocols. Further information can be found at www.nature.com/protocolexchange/about

With kind regards,
Daryl

Daryl Jason Verzosa David, PhD

Senior Editor, Nature Cell Biology
Nature Portfolio
Advisory Editor, npj Biological Physics and Mechanics

Heidelberger Platz 3, 14197 Berlin, Germany
Email: daryl.david@nature.com
ORCID: <https://orcid.org/0000-0002-9253-4805>
